# Timing and trajectory of *BCR::ABL1*-driven chronic myeloid leukaemia

Aleksandra E. Kamizela[1,2,3,8], Daniel Leongamornlert[1,8], Nicholas Williams[1], Xin Wang[1,4], Kudzai Nyamondo[1,2,3], Kevin Dawson[1], Michael Spencer Chapman[1,5], Jing Guo[1], Joe Lee[1,2], Karim Mane[1,2], Kate Milne[3], Anthony R. Green[2,3], Timothy Chevassut[6], Peter J. Campbell[1,2], Patrick T. Ellinor[4,7], Brian J. P. Huntly[2,3], E. Joanna Baxter[3] & Jyoti Nangalia[1,2,3 ✉]

Mutation of some genes drives uncontrolled cell proliferation and cancer. The Philadelphia chromosome in chronic myeloid leukaemia (CML) provided the very first such genetic link to cancer[1,2]. However, little is known about the trajectory to CML, the rate of *BCR::ABL1* clonal expansion and how this affects disease. Using whole-genome sequencing of 1,013 haematopoietic colonies from nine patients with CML aged 22 to 81 years, we reconstruct phylogenetic trees of haematopoiesis. Intronic breaks in *BCR* and *ABL1* were not always observed, and out-of-frame exonic breakpoints in *BCR*, requiring exon skipping to derive *BCR::ABL1*, were also noted. Apart from *ASXL1* and *RUNX1* mutations, extra myeloid gene mutations were mostly present in wild-type cells. We inferred explosive growth attributed to *BCR::ABL1* commencing 3–14 years (confidence interval 2–16 years) before diagnosis, with annual growth rates exceeding 70,000% per year. Mutation accumulation was higher in *BCR::ABL1* cells with shorter telomere lengths, reflecting their excessive cell divisions. Clonal expansion rates inversely correlated with the time to diagnosis. *BCR::ABL1* in the general population mirrored CML incidence, and advanced and/or blast phase CML was characterized by subsequent genomic evolution. These data highlight the oncogenic potency of *BCR::ABL1* fusion and contrast with the slow and sequential clonal trajectories of most cancers.

Chronic myeloid leukaemia (CML) occupies a landmark position in the history of oncology research, marking the first instance in which a genetic anomaly was implicated in the development of cancer. The seminal discovery of the Philadelphia (Ph) chromosome in 1960 by Nowell and Hungerford[1] and *BCR::ABL1* fusion gene in 1973 by Rowley[2], heralded the era of oncogenomics. Targeting *BCR::ABL1* by means of tyrosine kinase inhibition (TKI) has since resulted in uniquely successful patient outcomes in CML, a result not replicated for most other cancers[3].

Cancers emerge from the stepwise accumulation of key genetic mutations critical to cell growth and regulation[4]. Such mutations accumulate over an extended period, commencing decades before clinical presentation, for example, early in life whole-genome duplication in ovarian cancer and chromosome 3p loss in clear cell renal cell carcinoma[5,6]. Cancer evolution may even commence in utero, as demonstrated for *JAK2* mutations in adult-onset polycythaemia vera[7]. By contrast, cancer incidences in Japanese survivors of the atomic bombs showed a peak in CML within 10 years of radiation exposure, raising the possibility that *BCR::ABL1* driven clonal expansion and the trajectory to CML are unlike that of adult malignancies studied so far[8].

Somatic mutations accumulate in haematopoietic cells throughout life in a clock-like fashion[7,9]. The resulting unique mutational profiles of individual cells can be harnessed to reconstruct phylogenetic trees that depict ancestral cellular relationships and evolutionary history. This approach has enabled precise quantification of clonal dynamics in both healthy haematopoiesis and haematological malignancy[7,9,10]. Here, using genome-wide somatic mutations and phylogenetic inference, we characterize the fitness and trajectory of *BCR::ABL1* driven clonal expansion and how these factors affect clinical features of CML.

## Driver mutations in CML colony genomes

We studied nine patients, aged 22–81 years at presentation, with chronic phase CML (Fig. 1a and Supplementary Table 1). Patients harboured a *BCR::ABL1* translocation t(9;22)(q34;q11) with typical fusion transcripts e14a2 or e13a2 involving the major breakpoint cluster region of *BCR*. Patients had varying therapeutic responses to TKI therapy. Three patients were responsive to first-line Imatinib or Dasatinib, six required second line TKI and two patients failed several different TKIs (Supplementary Table 1 and Extended Data Fig. 1). Blood and bone marrow were sampled from diagnosis in six out of nine patients and further during therapy in four patients (Fig. 1a). In three individuals (PD57333, PD57334 and PD57335), sampling time points were only after diagnosis (7 months to 2 years and 9 months, Fig. 1a and Supplementary Table 1).

[1]Wellcome Sanger Institute, Hinxton, UK. [2]Cambridge Stem Cell Institute and Department of Haematology, University of Cambridge, Cambridge, UK. [3]Cambridge University Hospitals NHS Trust, Cambridge, UK. [4]Cardiovascular Disease Initiative, Broad Institute of MIT and Harvard, Cambridge, MA, USA. [5]Centre for Haemato-Oncology, Barts Cancer Institute, Queen Mary University of London, London, UK. [6]Department of Clinical and Experimental Medicine, Brighton and Sussex Medical School, Brighton, UK. [7]Cardiovascular Research Center, Massachusetts General Hospital, Boston, MA, USA. [8]These authors contributed equally: Aleksandra E. Kamizela, Daniel Leongamornlert. ✉e-mail: jn5@sanger.ac.uk

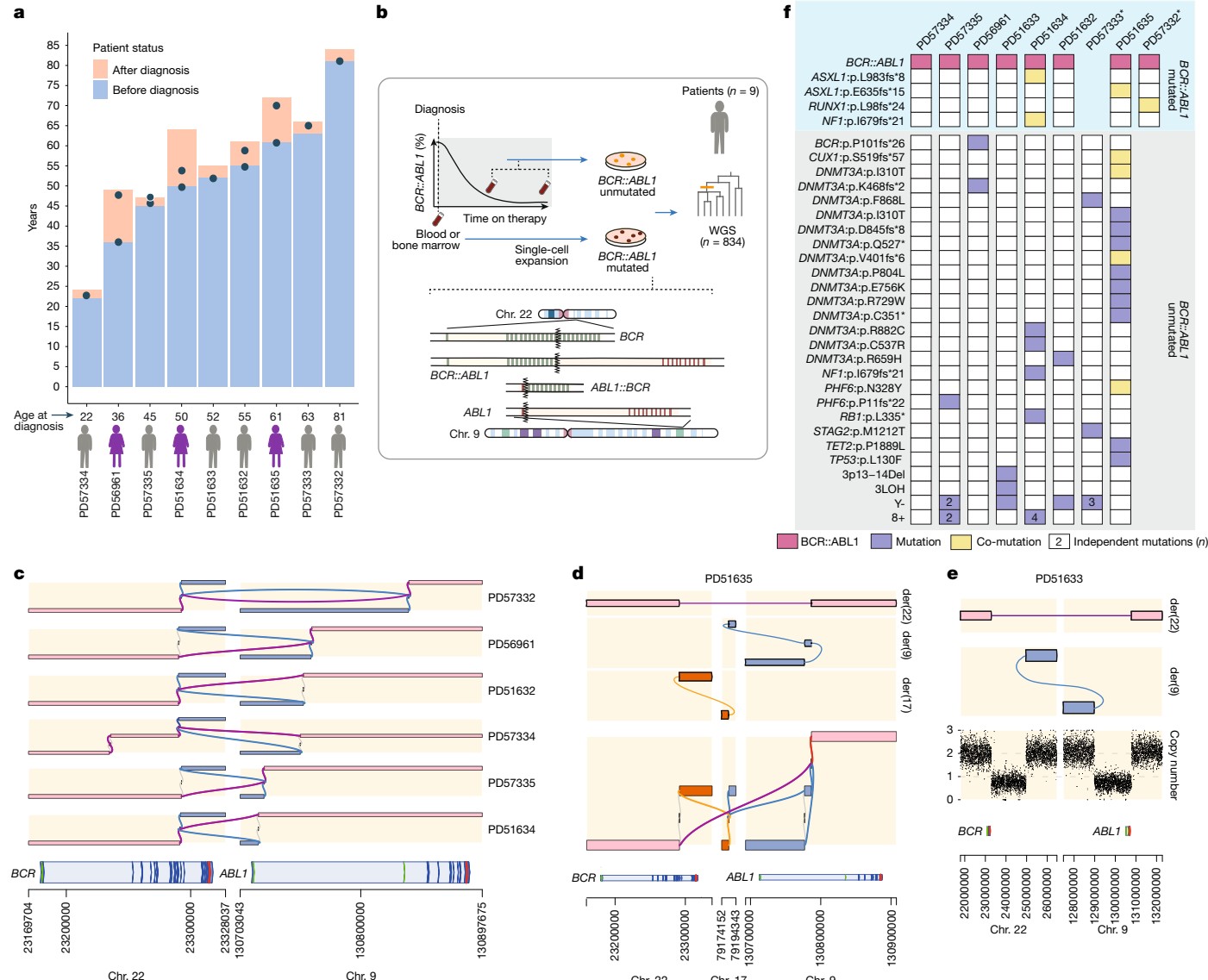

**Fig. 1 | Patient cohort and study design. a**, Patient cohort. Bar lengths correspond to the patient age at last follow-up or death, with orange shading representing disease duration and blue shading representing time before diagnosis. Dots represent sampling time points. **b**, Experimental design with a schematic of t(9:22) translocation that results in *BCR::ABL1* fusion (Philadelphia chromosome or derivative chromosome (chr.) 22, der(22)) and reciprocal *ABL1::BCR* fusion (der(9)). **c**, Structural variant diagram of typical, reciprocal *BCR::ABL1* present in six out of eight patients. The der(22) is shown in pink with a purple 'path' from chr. 9 to chr. 22, whereas der(9) is shown in blue with a blue path from chr. 22 to chr. 9. Any segment without a purple or blue path shows a deletion of the reference genome: such small losses (1–49 bp) occur in five out of six patients. Any segment with both a purple and blue path shows a duplication of the reference genome. **d**, *BCR::ABL1* in patient PD51635 involving

chromosome 17, producing atypical der(9) shown in blue, der(17) shown in orange and der(22) shown in pink. **e**, *BCR::ABL1* in PD51633 involves large deletions on der(9) precluding formation of *ABL1::BCR*. Lower panel shows the copy number changing from two to one for large segments on chr. 22 and chr. 9, representing deleted regions. **f**, Driver mutations and copy number alterations detected in at least one colony for each patient. Pink squares indicate the identified *BCR::ABL1* translocation, purple squares indicate single mutations within a patient and yellow squares indicate that the mutation occurs in a colony that also carried a different mutation. Mutations observed within the CML clone are shown at the top within the blue shaded area, and mutations present in wild-type colonies are shown below within the grey shaded area. Numbers represent the number of independently acquired mutations for a given gene within a patient.

Mononuclear cells (MNCs) were cultured in vitro to provide single-cell derived clonal DNA for whole-genome sequencing (WGS) (Fig. 1b). Following quality control, 179 out of 1,013 colonies were excluded due to low sequencing coverage or reduced clonality. In total, 834 whole genomes were taken forward (mean 93 whole genomes per patient (range 35–163), mean depth of sequencing 15.3 times (range 8.6–46.3 times)). A total of 397,063 autosomal single-nucleotide variants (SNVs) and 13,646 autosomal insertion and deletions were identified.

We detected *BCR::ABL1* in eight out of nine individuals (Fig. 1c–e). In PD57333, sampled at molecularly detectable relapse (*BCR::ABL1/ABL1*

ratio of 3%), we only captured *BCR::ABL1* in whole blood WGS (two DNA read pairs in average read depth 45.9×), but not from the cultured colonies due to the low mutant clonal fraction. WGS provides a unique opportunity to interrogate the *BCR::ABL1* breakpoint due to the ability to reconstruct exact genomic *BCR::ABL1* (derivative, der(22)) and *ABL1::BCR* (der(9)) breakpoints, not otherwise accessible from routine clinical complementary DNA (cDNA) analysis. Two observations were apparent. First, deleted or duplicated DNA regions (length 1–687 base pairs (bp) across der(22) and der(9)) were observed at the point of fusion (Supplementary Table 2). In two patients, this completely

disrupted der(9). In PD51635, there was a complex three-way translocation resulting in der(22), but also t(9;17)(q34.12;q25.3) and t(17;22)(q25.3;q11.23) resulting in an atypical der(9) (Fig. 1d and Supplementary Table 2). In PD51633, we observed large roughly 1 Mb losses deleting both 5′ ABL1 and 3′ BCR along with 55 extra genes (Fig. 1e and Supplementary Table 2). We also observed non-templated inserted sequences at points of fusion (Extended Data Fig. 2b,c). Second, whereas several individuals harboured the expected intronic breakpoints within BCR and ABL1, we observed three cases (PD56961, PD57332 and PD57335) with exonic breakpoints in BCR, an occurrence only rarely reported in the literature[11,12]. All three cases predicted out-of-frame fusions with ABL1, and continuation of the fusion reading frame would result in a stop codon within 1–17 codons (Extended Data Fig. 2a–c). Nevertheless, these patients harboured typical clinically detectable fusion transcripts (Extended Data Fig. 2d). These fusion transcripts would have only been compatible with the DNA derived event if the out-of-frame exon was spliced out. Using SpliceAI, all cases showed a reduced splice acceptor probability for the out-of-frame exon (Δ score PD56961 0.49, PD57332 0.67 and PD57335 0.75), supporting this hypothesis (Extended Data Fig. 2e).

Within BCR::ABL1-positive colonies, we found occasional further mutations in ASXL1 (L983fs*8, E635fs*15) in PD51635 and PD51634. In PD57332, a canonical RUNX1 RUNT domain truncating mutation was present as a dominant subclone after BCR::ABL1 acquisition (Figs. 1f and 2). These driver mutations have been reported to affect patient prognosis, treatment response and disease transformation[13–16]. Further driver mutations were more commonly found in BCR::ABL1-negative colonies, particularly trisomy 8 (six events), DNMT3A mutations (14 events) and loss of Y, but also mutations in TET2, NF1, CUX1, STAG2, PHF6, BCR, RB1 and TP53 (Figs. 1f and 2). Perhaps, it is not surprising that mutations in these genes are not consistently associated with CML features and outcomes[15–17], given that they are not preferentially observed within the CML clone.

## Phylogenetic trees in patients with CML

Phylogenetic trees of haematopoietic cells in nine patients with CML are depicted in Fig. 2, and show the pattern of sharing of genome-wide somatic mutations across individual colonies. A shared branch represents mutations identified in downstream descendant lineages, and mutations within the lowest branches are only found in single colonies. Several general observations can be made. As would be expected, the total number of mutations in individual colonies increases with patient age, in keeping with clock-like mutation acquisition in human haematopoietic stem and progenitor cells (HSPCs)[9,10,18]. All trees have an abundance of coalescences (branching points) near the root, or 'top', of the trees. This reflects the rapid expansion of HSPCs and resulting genomic divergence during embryogenesis, such that for any two normal HSPCs, their most recent common ancestor (MRCA) is near the start of life. Following early expansion, normal haematopoiesis in young patients have a 'comb'-like phylogenetic structure, whereas older patients have many clonal expansions[9]. We observed prominent intra-patient recurrence for specific driver mutations, for example, eight independent DNMT3A mutations in PD51635, three independent loss of chromosome Y events in PD57333, two independent chromosome 3 aberrations in PD51633 and recurrent trisomy 8 in PD51634, strongly suggesting the existence of patient-specific selection landscapes.

A striking feature of these trees is the pattern of branching in BCR::ABL1-positive colonies. These colonies emerge below a long, vertical branch that only carries one identified driver among all somatic mutations assigned to this branch: the BCR::ABL1 translocation. The BCR::ABL1-positive clade in six patients has a rapid burst of coalescences emerging directly beneath this shared branch, akin to that observed at the start of life, indicating rapid division into a large pool of leukaemic HSPCs. In two patients (PD51635 and PD51633), a relatively slower, but still recent clonal expansion pattern is observed. These data indicate that the MRCA and commencement of tumour clonal expansion in patients with CML are recent events, contrasting with patterns observed for other driver mutations in haematopoiesis[5–7,9].

## Mutation accumulation in BCR::ABL1 cells

Mutations accumulated in BCR::ABL1-negative HSPCs at 18 mutations per year (95% bootstrapped confidence interval, $CI_{boot}$, 16.1–20.2) in line with published data[7,9]. BCR::ABL1-positive HSPCs had roughly 90 extra mutations (mean +91.7, $CI_{boot}$ 33.6–151.6, Fig. 3a and Supplementary Note 1). Indeed, in many patients, the mutation burden of BCR::ABL1-positive colonies was equivalent or higher than BCR::ABL1-negative HSPCs despite being sampled at a younger age (PD51634 and PD51632) (Figs. 2 and 3a).

We considered whether these increased mutations were due to new or existing mutational processes. In BCR::ABL1-positive and negative colonies, expected endogenous clock-like mutational processes (single-base substitution (SBS) signatures SBS1 and SBSblood)[9,18,19] were present in addition to SBS18 (Fig. 3b,c). No additional mutational processes were found, contrary to previous reports of BCR::ABL1-induced DNA damage[20]. We found no evidence of TKI-induced mutations, however, recurrent trisomy 8 was observed in BCR::ABL1-negative colonies (Figs. 1f and 2), which could reflect increased proliferation or survival of trisomy 8 HSPCs during CML development, or their subsequent positive selection by TKI.

To explore how existing mutational processes contribute to increased mutation burden in BCR::ABL1-positive cells, we assigned mutational signatures to individual branches of phylogenies (Fig. 3b and Extended Data Figs. 3 and 4), grouping by (1) BCR::ABL1-negative (wild-type) branches, (2) shared branch of the BCR::ABL1 clade representing the pre-CML lineage, (3) early shared branches of the BCR::ABL1 clade, representing historical early CML clonal expansion, (4) all branches of the BCR::ABL1 clade, including recent mutations and (5) early-in-life branches. An increased proportion of SBS1 mutations was observed in the BCR::ABL1 clade (Fig. 3c and Extended Data Fig. 3, +13.6–30.9%, $P = 4.0 \times 10^{-11}$), mirrored by an increased proportion of C>T transitions at CpGs (Fig. 3d and Extended Data Fig. 4, +16.6–31.3%, $P = 2.9 \times 10^{-11}$). This reflects an increased spontaneous deamination of methylated cytosines resulting from increased cell divisions[21] (Fig. 3e). A similar pattern was observed for SBS18, which has also been observed in the rapid growth contexts of fetal haematopoiesis[19] and placental tissue[22]. Overall, our data show that increased mutations in CML cells simply reflect their higher rate of cell division.

We observed no differences in mutational patterns between the pre-CML lineage (shared branch of the BCR::ABL1 clade) and wild-type HSPCs (Fig. 3e). This confirms that the CML cell-of-origin acquired mutations similarly to normal HSPCs before its transformation by BCR::ABL1, which probably occurred at the end of the long shared branch.

It was unusual to sample BCR::ABL1-positive colonies years after diagnosis, as most patients were in molecular remission. PD51635 had clinically stable and low BCR::ABL1/ABL1 ratios (0.5%) on TKI (Extended Data Fig. 1), and their two sampled BCR::ABL1-positive colonies (Fig. 2) had a lower proportion of C>T at CpGs (mean 0.18) when compared with BCR::ABL1-positive genomes at diagnosis (mean 0.26, Fig. 3f) suggesting that these lineages were dividing at normal rates despite harbouring BCR::ABL1. We confirmed that the BCR::ABL1 translocation events in these colonies and the diagnostic sample were identical. This unusual observation raises the possibilities that TKI therapy resulted in preferential clearance of only more rapidly dividing BCR::ABL1-mutated cells, that TKI therapy slows cell division in some BCR::ABL1-positive cells or that the downstream consequences of BCR::ABL1 have been epigenetically silenced in this patient. Competing clonal haematopoiesis

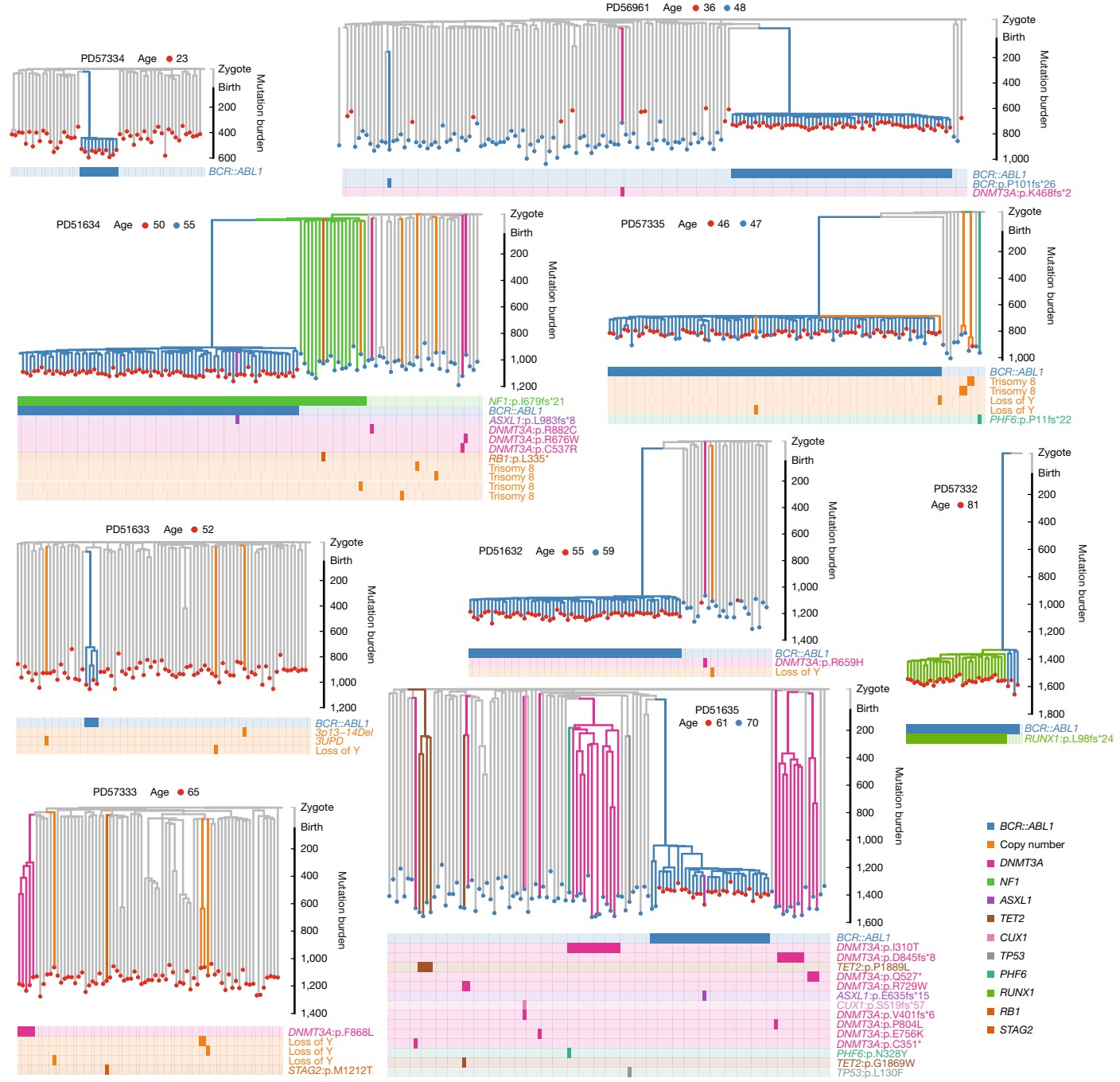

**Fig. 2 | Phylogenetic trees of chronic phase CML.** Each tree shows the pattern of sharing of somatic mutations in sampled colonies. The vertical axis on the right shows the somatic mutation burden at any point of the tree. Private branches represent those mutations present in only a single colony, and shared branches represent mutations present in one or more colonies. A branching point (coalescence) in the tree represents an ancestral haematopoietic stem cell that underwent a symmetrical self-renewing division wherein progeny of each of the daughter cells was sampled as a colony, and represents the MRCA of the downstream colonies, or the 'clade' that share this MRCA. The dots at the bottom represent individual sequenced colonies, with their colours corresponding to sampling time points. Highlighted branches are coloured by the driver mutation and copy number change that they carry.

or immune regulation may also have constrained cell division rates of remaining *BCR::ABL1* cells.

## Telomere lengths in CML

Telomeres, the repetitive DNA sequences at chromosome ends that shorten with cell division, are a direct read out of cell division history. In healthy HSPCs, telomere attrition has been shown to be roughly 30 bp per year[9], similar to that observed here in wild-type cells ($n$ = 469, −30.8 bp per year, $CI_{boot}$ −12.6–48.2 bp). The model intercept, which provides an estimated telomere length at birth, was 6,180 bp ($CI_{boot}$ 5,216–7,086 bp) in line with previous estimates[9]. *BCR::ABL1*-positive HSPCs ($n$ = 365) had a marked decrease in telomere length despite being sampled at a younger age, with an extra 556.9 bp of loss independent of age (CI −86.0–993.2 bp, Supplementary Note 2 and Fig. 3g). We did note the unusually short telomeres in PD51635 *BCR::ABL1*-negative HSPCs, which could reflect the increased cell division history of clonal haematopoiesis within the wild-type compartment (Fig. 3h).

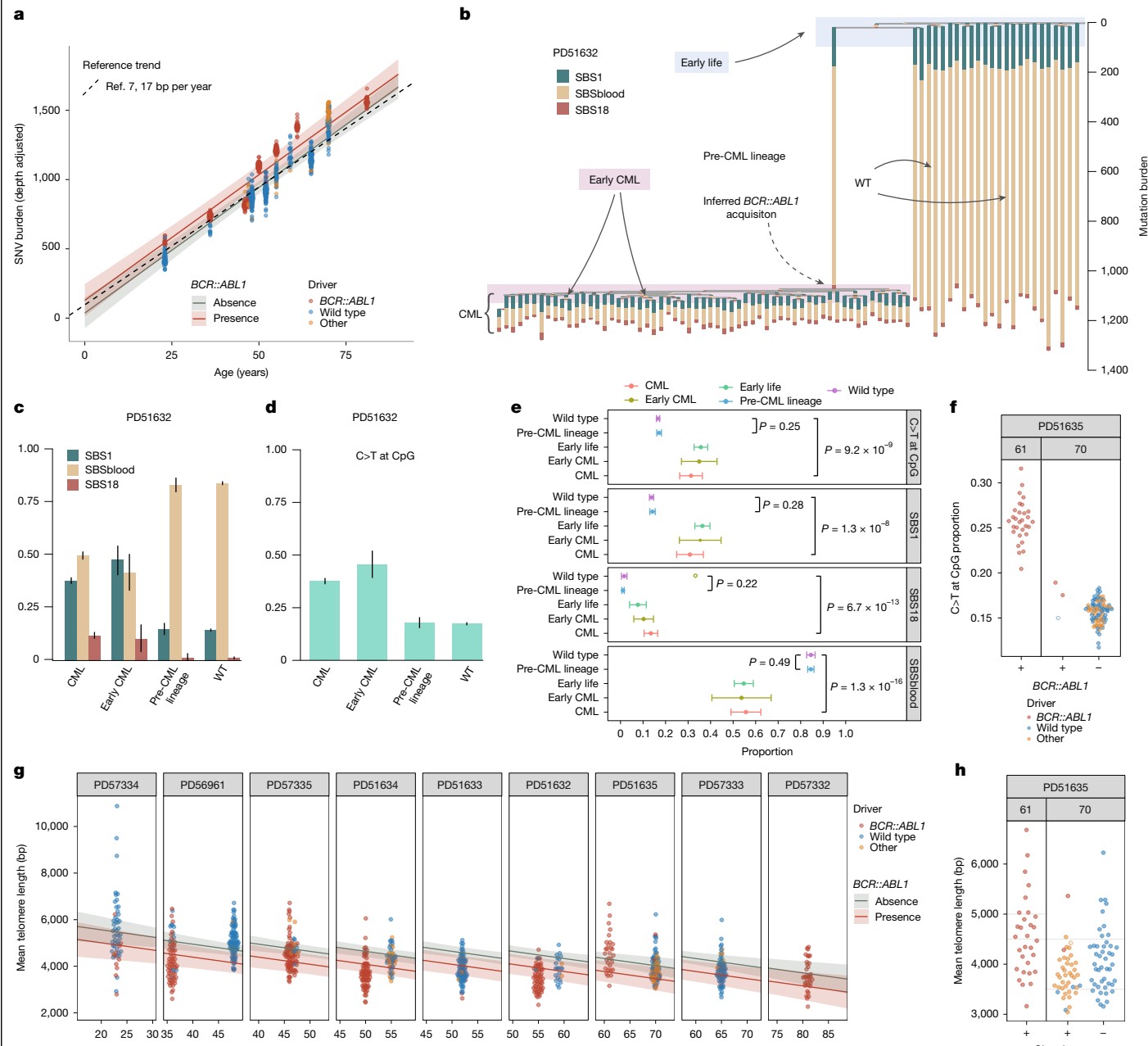

**Fig. 3 | Mutation accumulation in *BCR::ABL1* cells. a**, Number of somatic SNVs carried by colonies (*n* = 834), coloured by driver status (*BCR::ABL1*, red; other driver mutation, orange; wild-type (WT), blue) and relationship with age, with mixed effects model regression lines and 95% confidence intervals stratified by *BCR::ABL1* status (red, *BCR::ABL1*-positive; grey, *BCR::ABL1*-negative, model excludes PD57332 due to lack of wild-type, *n* = 799). **b**, A phylogenetic tree for an example patient PD51632 with the estimated proportion of SBS signature SBS1, SBSblood and SBS18 contributing to SNVs for each of the branches, categorized into five groups: (1) branches of the *BCR::ABL1*-positive clonal expansion ('CML'), (2) ancestral branch of *BCR::ABL1* colonies representing the lineage of CML origin ('pre-CML lineage'), (3) early mutations within the CML (that is, shared branches within the clonal expansion, 'early CML', lower shaded purple box), (4) branches of *BCR::ABL1*-negative colonies (wild-type, WT) and (5) early-in-life mutations representing the top 100 mutations of the phylogenetic tree, upper shaded blue box. **c**, Bar plot of SBS1, SBSblood and SBS18 proportions contributing to SNV spectra for PD51632, with the height of the bar showing the posterior mean and the error bars representing 95% credibility intervals. **d**, Bar plot represents the proportion of C>T at CpG changes

for PD51632. The height of the bar depicts the observed proportions and the error bars mark the 95% binomial proportion confidence intervals. **e**, Cohort-wide plots of proportions of C>T at CpGs, SBS1, SBS18 and SBSblood, excluding PD57333 and PD57332 because of absence of either WT or mutant colonies in these patients. The dots represent the mixed effect meta-analysis estimated proportion and the bars 95% confidence intervals. The *P* values are for a two-sided test of the null hypothesis that the difference between proportions is zero. The *P* values are not adjusted for multiple testing. **f**, Per sample dot plot for PD51635, showing C>T at CpG proportion (excluding *BCR::ABL1* clade trunk in *BCR::ABL1*-positive colonies) by sample *BCR::ABL1* status, faceted by time of sampling and annotated by per sample driver status. **g**, Per-patient estimated mean telomere length (bp) of colonies (*n* = 834) by age of sampling, annotated by driver status and mixed effects model regression lines and 95% confidence intervals stratified by *BCR::ABL1* status. **h**, Per sample dot plot describing mean telomere length (bp) by clonal status (*BCR::ABL1*, an alternative driver mutation or 'driverless' clonal expansion) for PD51635, faceted by time of sampling and annotated by per sample driver status.

## Timing and rate of *BCR::ABL1* expansion

We calculated the timing of the start of *BCR::ABL1* clonal expansion by modelling mutation accumulation at a constant rate until the commencement of clonal expansion, following which there was a higher rate of mutation acquisition, as shown above. The timing of the coalescences is inferred using our Markov chain Monte Carlo method rtreefit[23] adapted to account for the transformation event occurring at the end of the long 'trunk' preceding the clonal expansion (Supplementary Note 3). With these 'time-based' trees, we next estimated growth rates using our phylofit[7] tool, which assumes a sigmoidal clonal outgrowth curve. The upper end of the prior range of exponential phase growth rate, $s$ was increased until the posterior estimate of $s$ was insensitive to further increases (Extended Data Fig. 5a). We report annualized growth rate, $S = e^s - 1$, as a percentage, alongside instantaneous per year growth rate, $s$, where in a small time period, $\delta t$, measured in years, the fractional increase in the clone size is roughly $s\delta t$.

In the two youngest patients (PD57334, 22 years; PD56961, 36 years), the time from onset of clonal expansion to clinical diagnosis was only 3.2 years (CI 2.4–4.2 years) and 3.9 years (CI 3–5.3 years, Fig. 4a and Supplementary Tables 3 and 5). In keeping with recent tumour origins, estimated growth rates were very high at 99,000,000% per year (CI 906,000–13,500,000,000% per year, $s = 13.81$, CI 9.11–18.72) and 26,800% per year (CI 11,800–63,000% per year, $s = 5.59$, CI 4.78–6.44, Fig. 4b and Supplementary Tables 3 and 5). These translate to the mutant clade doubling in size every 18 to 45 days, respectively. Three middle-aged patients (45–55 years, PD51632, PD51634, PD57335) also had recent and rapid *BCR::ABL1* clonal expansion to diagnosis, ranging from 4.4 years (CI 3.4–5.8 years) with a growth rate of 73,500% per year (CI 24,000–240,000% per year, $s = 6.60$ CI 5.49–7.78, PD57335) to 6.3 year (CI 5.0–8 years) with a growth rate of 2,700% per year (CI 1,700–4,300% per year, $s = 3.33$ CI 2.89–3.79, PD51634) (Fig. 4a,b and Supplementary Tables 3 and 5).

By contrast, in three patients aged 52 years (PD51633), 61 years (PD51635) and 81 years (PD57332), we observed slightly longer clonal trajectories of 13.1 years (CI 10.5–16 years), 14.0 years (CI 12.0–16.2 years) and 10.3 years (CI 7.9–13.4 years), respectively. These patients had intermediate rates of clonal expansion, nonetheless, achieving growth at 114–245% per year ($s = 0.76–1.24$, Fig. 4a,b). Of note, the growth rate in PD51635 seemed initially to be slower (as judged by longer inter-coalescent intervals) and then more rapid within a subclone (Fig. 2 and Supplementary Tables 3 and 5).

Given such extremely rapid inferred growth rates, we benchmarked phylofit to assess whether large growth rates could be accurately inferred (Supplementary Note 3). Orthogonal estimates of growth rate using cloneRate[24] ('maxLikelihood' and 'birthDeathMCMC') gave similar growth estimates (Extended Data Fig. 5b and Supplementary Table 4), providing extra support for the very high estimated growth rates in CML. A notable exception is PD57335 in which late coalescences in the mutant clade have a noticeable impact on cloneRate estimates but not when using phylofit (Supplementary Table 4). As previously reported[24], phylofit credibility intervals were less conservative than those from cloneRate.

Acknowledging the small number of patients, there seemed to be a plausible trend of younger age of onset correlating with more explosive growth together with a shorter duration between the beginning of *BCR::ABL1* clonal expansion and diagnosis (Fig. 4a,b), and older individuals demonstrating relatively less explosive clonal expansion. PD51632 was a slight outlier to these trends, having had treatment for colorectal cancer with chemotherapy in the preceding 8 months before CML diagnosis, which may have enhanced selection on *BCR::ABL1*-positive HSPCs.

We explored the clinical significance of differences in CML growth rates and clinical response. All three of the fastest growing CMLs in the cohort failed to achieve optimal molecular remission (defined as *BCR-ABL1/ABL1* ratio less than 1% by international standards) at 12 months due to TKI-induced cytopenias and slow reduction in *BCR::ABL1* levels (Extended Data Fig. 1). By contrast, two of the slowest growing CML clones in the oldest patients achieved rapid major molecular remissions. These observations raise the possibility that growth rates in CML may affect early therapeutic response.

Despite the varying clinical presentation patterns within this small cohort, we observed high correlation between the inferred growth rates of CML and the time to clinical presentation (Fig. 4c,d). Given that these two parameters were inferred from distinct features of the tree, namely, patient age at the start of the clonal expansion versus the pattern of coalescences within the clonal expansion (Fig. 2 and Methods), this pattern was striking. This observation provides evidence for the predictability of clinical presentation based on tumour growth rates and suggests that there is a limit to the population size of CML leukaemic stem cells that can be tolerated before inevitable clinical presentation (Supplementary Note 4).

## *BCR:ABL1* and CML in the All of Us cohort

The relative timing of mutations within a branch of a phylogenetic tree cannot be disentangled. Although we can accurately estimate the timing of commencement of *BCR::ABL1* clonal expansion, we cannot pinpoint exactly when *BCR::ABL1* occurs along the preceding long shared branch. Circumstantial evidence strongly suggests that *BCR::ABL1* occurs at the end of this branch, triggering clonal expansion: (1) mutational patterns in the *BCR::ABL1* branch are identical to wild-type HSPCs, suggesting absence of *BCR::ABL1* until the end of the branch; (2) *BCR::ABL1* is sufficient to induce a CML-like phenotype in mice[25], indicating that its acquisition triggers transformation; (3) no additional recurrent genetic driver mutations were found on shared branches harbouring *BCR::ABL1*; (4) pre-existing clonal haematopoiesis was not observed upstream of *BCR::ABL1* and (5) acquiring *BCR::ABL1* earlier in the branch would necessitate additional non-genetic events to consistently trigger clonal expansion in every patient, making this explanation less parsimonious and violating Occam's razor.

Nevertheless, by only studying patients with CML, any possibility that *BCR::ABL1* may also drive slower, decades long expansion, such as CH, may be overlooked. Whereas *BCR::ABL1* has not been reported in CH, such studies have generally only focussed on point mutations, insertions or deletions or chromosomal copy number changes[26,27]. Furthermore, studies of genomic translocations in healthy blood have typically used whole blood cDNA and not assessed for the presence of stable HSPC clones[28]. To explore this, we examined the US-based All of Us cohort[29] of 206,173 whole-genome sequenced participants with linked electronic health data for evidence of *BCR::ABL1* CH. We identified 39 individuals (0.019%) with two or more supporting reads of *BCR::ABL1* or *ABL1::BCR*, most of whom had a diagnosis of CML, haematological features suspicious of CML (for example, splenomegaly or basophilia) or unspecified haematological issues (Extended Data Fig. 6a). Although 14 individuals did not have a CML-relevant status reported, of these, three had blood sampling after their most recent clinical record and three had no clinical information beyond 1 year after sampling (Extended Data Fig. 6b). All those with follow-up data exceeding 2.5 years had a CML-relevant status reported. These findings suggest that a readily detectable *BCR::ABL1* clone in the healthy population, in the absence of a CML diagnosis, is genuinely rare. In fact, the prevalence of *BCR::ABL1* in the All of Us (0.019%) closely matches the prevalence of CML in the United States (0.02%)[30], indicating that *BCR::ABL1* does not also drive asymptomatic CH.

We also identified 40 individuals (0.019%) with a single read of *BCR::ABL1* or *ABL1::BCR*. The confidence in these cases is lower, and more than half lacked features of CML. We hypothesized that this group could include genuine patients with CML (for example, prediagnosis or treated), *BCR::ABL1* artefacts, or *BCR::ABL1* in non-HSPCs. To address

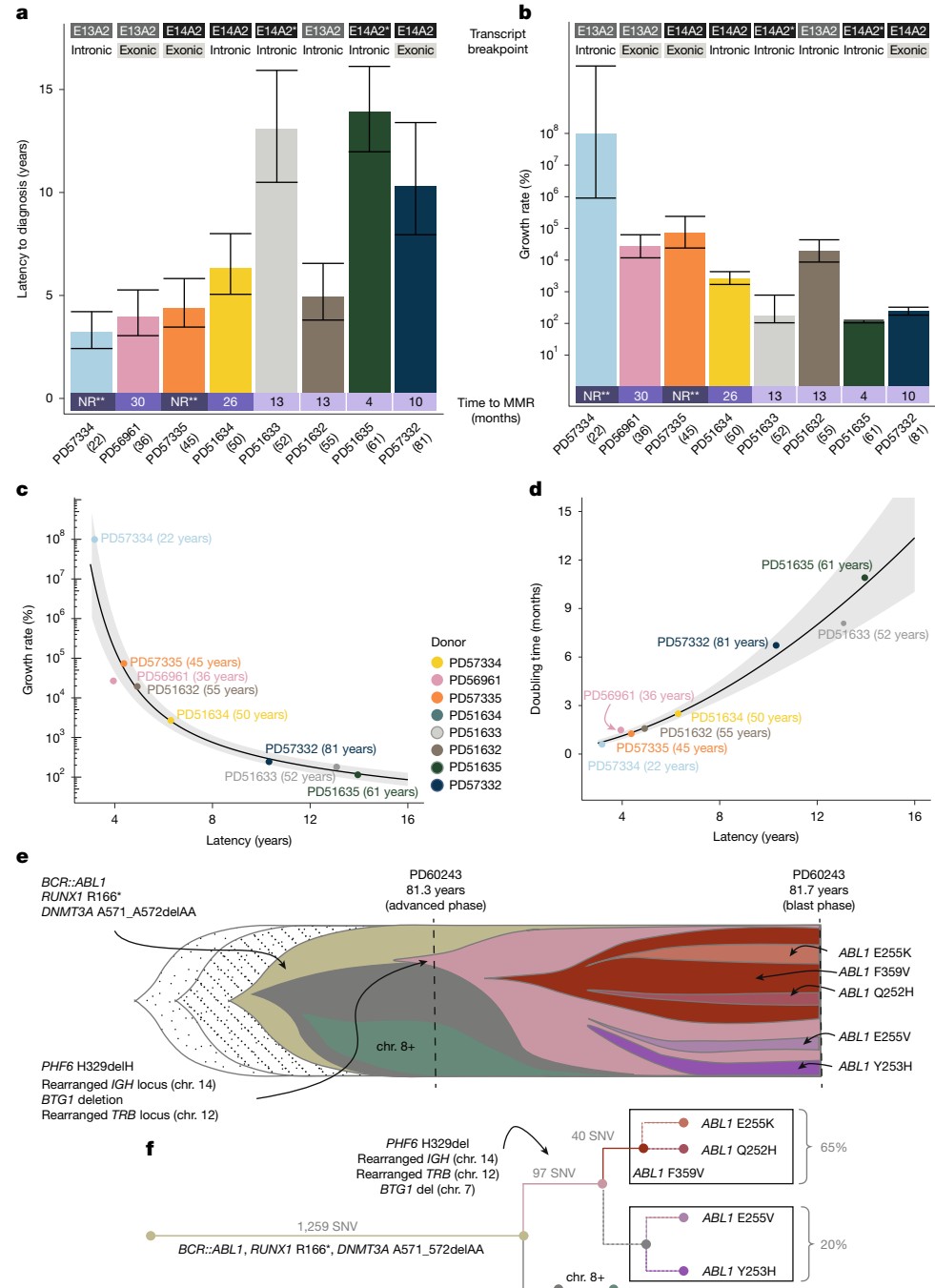

**Fig. 4 | Timing and fitness of *BCR::ABL1*-driven clonal expansion. a,b,** Bar plots show rtreefit-based latency in years (**a**) and phylofit-based (Methods) annualized growth rate as a percentage (**b**), with patients ordered by age at diagnosis. The height of the bars and vertical lines indicate the posterior median and equal-tailed 95% credibility intervals, respectively. Mutant transcript type and intronic versus exonic out-of-frame *BCR* and *ABL1* breakpoints are depicted along top strips. An asterisk * indicates non-reciprocal *BCR::ABL1* translocation with der(9) loss. Time (in months) to major molecular remission (MMR) (*BCR::ABL1/ABL1* ratio less than 0.1% by International Standards) is shown underneath. NR, not reached MMR. **Follow-up times for PD57334 and PD57335 were 23 and 21 months, respectively. **c,** Relationship between inferred *BCR::ABL1* acquisition (years) and annualized growth rate (%) per year for the *BCR::ABL1*-positive clone. The dots are based on the posterior median estimates of *S* and latency and the fitted line and 95% confidence interval (grey band) is derived from the linear model log(log(1 + *S*)) ~ log(latency). **d,** Estimated doubling time $\left( = \frac{1}{\log_2(1 + S)} \right)$ versus time from *BCR::ABL1*-induced clonal expansion

to diagnosis ('latency'). The fitted line and 95% confidence interval (grey band) is derived from the linear model used in **c. e,f,** Fish (**e**) and tree plot (**f**) describing the broad architecture of clones found at advanced phase (81.3 years) and blast phase (81.7 years) in PD60243. *ABL1* mutant clones were confirmed to be in different cells using read based phasing. Tree reconstructions predict a median of 2 (range 1–3) *ABL1* mutant clones nested within the *ABL1* p.F359V clone, an example of which is shown. The fishplot (**e**) *y* axis reflects the inferred cancer cell fraction of the clones at each measured time point, and the shown emergence points of clones are for illustrative purposes. Grey shaded components at the left of the fishplot (dotted and lined) represent historical mutant clones with single (grey dotted) and two (grey dashed) driver mutations from *BCR::ABL1*, *RUNX1* and *DNMT3A*: the ordering of these events is unknown. The tree plot (**f**) shows the same tree solution as **e**, with annotation of autosomal SNV burden for respective subclones. Dashed lines represent a possible branching structure. Boxes are labelled with respective clonal fractions for blast phase sample PD60243d.

the possibility of sequencing artefacts, we screened roughly 10,000 HSPC whole genomes from single-cell derived haematopoietic colonies, previously sequenced at the Sanger Institute[7,9,19,31,32], for *BCR::ABL1*. Given the clonality of these genomes, bona fide mutations should have a variant allele frequency (VAF) of 50%. WGS coverage was 10–15× and no donors were known to have CML. Across roughly 10,000 colonies, only three reads, each from a different colony, were translocated between *BCR* and *ABL1* target regions, but none were in the correct orientation for *BCR::ABL1* or *ABL1::BCR*. This suggests that single *BCR::ABL1* reads in All of Us are less likely to be sequencing artefacts, raising the possibility that some of these cases represent mutated, differentiated blood cells that cannot maintain clones over time. If this were the case, we would not expect such clones to increase in frequency with age. Indeed, single read *BCR::ABL1* carriers showed a more uniform incidence across age, whereas carriers with two or more reads, as well as patients with CML, showed increased incidence with age (Extended Data Fig. 6c). Overall, our analyses demonstrate that *BCR::ABL1* does not commonly cause CH, supporting the one-hit model of CML model depicted by our phylogenetic data.

### Disease progression in CML

Progression to blast phase CML is associated with extra mutations in *RUNX1*, *BCOR*, *TP53*, *ABL1*, +8 and isochromosome 17q (i(17q))[17,33,34]. Paired-exome and targeted gene sequencing before and after CML progression have also shown increased mutation burden[15,35]. Using WGS of peripheral blood MNC or granulocyte DNA, we analysed four patients (aged 38–81 years) with advanced CML, characterized by elevated circulating blast cells (10–19%). In all cases, we observed further driver mutations within the *BCR::ABL1* clone, such as, *RUNX1* p.Arg166*, *ASXL1* p.Tyr59fs, *ASXL1*.p.G646fs*12 and *BCOR* p.Leu532fs, along with copy number alterations such as +1q, −16q, +8 and i(17q) (Supplementary Table 6). *BCR::ABL1* with additional drivers expanded to dominate the CML clone, leaving little evidence of the historical *BCR::ABL1* 'only' clone. This contrasts with chronic phase CML in this study, in which additional drivers within the *BCR::ABL1* clone were either absent, or at low levels, in most cases (six out of eight). In two chronic phase cases, we did observe high clonal burden for additional driver mutations: in utero *NF1* frameshift (PD51634), and *RUNX1* mutation (PD57332). PD51634 subsequently transformed to blast phase CML, whereas PD57332 died soon after treatment initiation, with uncertainty on the phase of the underlying disease. These data confirm that advanced CML is largely genomically driven, characterized by *BCR::ABL1* clones that acquire additional driver mutations that offer further selection advantage.

One patient with advanced-phase CML (PD60243) developed blast crisis. The 81-year-old patient had elevated blasts (11% CD7⁺/CD13⁺/CD33⁺/CD117⁺/HLA-DR⁺ cells) at presentation with *RUNX1* p.Arg166* and *DNMT3A* p.Ala571_Ala572del mutations in addition to *BCR::ABL1* (Fig. 4e,f). This clone shared at least 1,259 mutations, not dissimilar to the expected number of mutations for normal HSPCs at this age, suggesting that the MRCA of this clone arose relatively recently (probably in the seventh decade of life). Two subclones were detected: (1) trisomy 8 (56% of *BCR::ABL1/RUNX1/DNMT3A*-positive cells) and (2) in-frame *PHF6* p.H329delH (14% of *BCR::ABL1/RUNX1/DNMT3A* cells) (Fig. 4e,f). A few months after treatment with Nilotinib, the patient developed roughly 20% blasts in the peripheral blood, at which point the *PHF6*-mutated subclone was clonally dominant. This clone showed further focal events—*BTG1* deletion (chromosome 12), rearranged *TRB* (chromosome 7) and rearranged *IgH* (chromosome 14)—typical changes of B cells consistent with lymphoid blast crisis. This confirms that lymphoid identity and differentiation emerged after *BCR::ABL1* acquisition by means of RAG-mediated genomic evolution. There were five different *ABL1*-mutated subclones within this highly mutated *BCR::ABL1* clone, with *ABL1*.p.Phe359Val as the dominant mutation (65% of cells). Other *ABL1* mutations included p.Q252H, p.Y253H, p.E255K and p.E255V.

Our tree reconstructions suggest that 1–3 *ABL1*-mutated clones may have occurred in cells already harbouring *ABL1*.p.Phe359Val, consistent with the notion that more than one *ABL1* mutation can coexist within the same clone. Aggregated *ABL1*-mutated clones constituted 70–99% of cells at blast phase (Fig. 4e,f). Subclones showed several hundred extra genome-wide mutations, inconsistent with the short time that had passed (0.4 years), indicating earlier *ABL1* mutation acquisition, rapid clonal outgrowth and/or genomic instability during blast phase.

## Discussion

Most cancers follow a multi-step trajectory, with clones progressively acquiring genetic and epigenetic alterations that lead to a malignant tumour. This process commences decades, frequently 30–50 years, before cancer presentation[5–7]. Against both these patterns, CML is an unusual neoplasm. Cancers driven by a single acquired genetic alteration are rare and we did not identify concurrent genomic alterations required for clonal expansion beyond *BCR::ABL1*. 'One-hit' cancers can be observed in the paediatric setting, with infantile *MLL*-rearranged acute lymphoblastic leukaemia[36] and paediatric ependymomas also driven by single genetic fusions[37,38]. These tumours presumably occur during a susceptible developmental window. Beyond the paediatric setting, haematological cancers driven by single genetic events, such as mutant-*JAK2* driven myeloproliferative neoplasms, are also characterized by slow clonal outgrowth over decades[7,9,32,39]. The cancer trajectory to adult CML bucks this trend, with explosive and rapid tumour growth, reaching tumour doubling times of 2–3 weeks and clinical disease presenting as soon as 3–4 years later. In keeping with this, our data highlight an absence of *BCR::ABL1* clonal haematopoiesis in the population. Subsequent genomic evolution following *BCR::ABL1* drives disease progression, in line with previous studies[15,17,40], but also determines cell identity during blast phase.

Our estimates of the duration from the start of *BCR::ABL1* clonal expansion to disease presentation fall within ranges from radiobiological studies of cancer incidences in Japanese survivors of the atomic bombs[8,41]. This suggests that ionizing radiation increases CML incidence by means of increased initiation of *BCR::ABL1*, and that subsequent CML clonal trajectories disregard the initial mode of *BCR::ABL1* acquisition. Indeed, such a short period from *BCR::ABL1* acquisition to clinical presentation can only be explained by extremely rapid outgrowth. Our estimates of duration and rates of growth are inferred from independent parameters of the phylogenetic trees; specifically, the length of the shared branch harbouring *BCR::ABL1* and the pattern of coalescent intervals during clonal expansion, respectively. We validate our estimates using orthogonal mathematical modelling and observe independent evidence of rapid growth through accelerated mutation acquisition and telomere attrition. Thus, the data presented offer robust inferences of CML clonal dynamics.

Within our small cohort, we observe variation in growth rates, with explosive growth (more than 10,000–1,000,000% per year) in three young patients and growth rates of 114 and 275% per year in two patients aged more than 60 years. Our cohort is too small to make definitive correlations but one may speculate as to possible reasons for this. Age could be a contributing factor through cell extrinsic factors, for example, competing clones of clonal haematopoiesis (PD51635) may hamper *BCR::ABL1* clonal expansion in older patients. Cell intrinsic causes, such as shortened telomeres, age-related HSPC changes or atypical t(9;22) events (for example, loss of reciprocal *ABL1::BCR*, observed in PD51633 and PD51635)[42] may also affect *BCR::ABL1* potency. Variation in growth rates could affect disease response, as four out of the five individuals with the highest growth rates failed to achieve optimal therapeutic targets, in contrast to more slowly growing CML. The intricate link between the rate of decline of *BCR::ABL1* levels post-TKI and the likelihood of achieving long-term remissions has long been recognized[43,44]; however, the mechanistic basis underlying patient-to-patient variability

in *BCR::ABL1* kinetics has been lacking. Our data indicate that the rate of growth of the tumour may be an important factor. One in five patients with CML still fail two lines of TKI therapy[45] and early treatment failure is associated with a poor prognosis. Thus, a better understanding of the role of CML cancer trajectories could enable earlier personalization of therapy to enhance tumour remission.

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

## Methods

### Patients and sample acquisition

Peripheral blood and bone marrow samples were obtained from patients with chronic phase CML. We selected patients who responded to first-line TKI (PD51633, PD51632, PD51635 and PD57332), those that did not respond to first-line TKI (PD57334, PD57335 and PD57333) and those with slow response (PD56961 and PD51634). Patients with advanced-phase CML were also included. Patients provided informed, written consent for the use of their samples for research. Patients were selected to include a wide range of ages at diagnosis and variable treatment outcomes. The study was covered under NHS Research Ethics Committee approval numbers 05/MRE/44 and 18/EE/0199.

### In vitro expansion of single-cell-derived blood colonies

MNCs were isolated from peripheral blood or bone marrow samples using Lymphoprep TM (Stem Cell Technologies). Single-cell suspensions of MNCs were grown in semisolid methylcellulose-based medium MethoCultTM H4034 Optimum (Stem Cell Technologies) for 10–14 days as previously described[7]. Colonies were individually picked and lysed in 45 µl of RLT buffer (Qiagen).

### RT–PCR for *BCR::ABL1* transcript

Transcript types from patients PD56961, PD57332 and PD57335 were determined using a standardized PCR with reverse transcription (RT–PCR) protocol[46]. Primers used were BCR-e1-A: GACTGCAGCTCC AATGAGAAC (*BCR* exon 1), BCR-b1-A: GAAGTGTTTCAGAAGCTTCTCC (*BCR* exon 12–13) and ABL-a3-B: GTTTGGGCTTCACACCATTCC (*ABL1* exon 3). This yielded a 344-bp PCR product for transcript e13a2 (6 bp (e12) + 106 bp (e13) + 175 bp (a2) + 57 bp (a3)) and a 419 bp PCR product for transcript e14a2 (6 bp (e12) + 106 (e13) + 75 (e14) + 175 (a2) + 57 (a3)).

### DNA library preparation, sequencing and read alignment

10–20 µl of lysed colony suspensions underwent WGS library preparation using the 'laser capture micro-dissected biopsy' pipeline[47] with eight cycles of PCR. This pipeline enables the generation of high complexity WGS libraries from an input of 150–200 cells. Samples with more than 2 ng µl$^{-1}$ of generated library DNA were used for paired-end, 150-bp reads WGS with Illumina NovaSeq 6000 machines. Reads were aligned to the human reference genome (GRCh38, NCBI) using the BWA-MEM (Burrows–Wheeler Aligner) algorithm.

### Somatic mutation identification and filtering

SNV were identified using CaVEMan[48] for each colony by comparison to an in silico unmatched sample (PD38Is_wgs). CaVEMan was run with the 'normal contamination of tumour' parameter set to zero, and the tumour or normal copy numbers set to five or two. Reads supporting an SNV had to have a median BWA-MEM alignment score greater than or equal to 140 and median number of soft clipped bases of 0. Further filtering designed for this bespoke pipeline was applied (https://github.com/MathijsSanders/SangerLCMFiltering). The use of the unmatched normal meant that this process called both somatic and germline SNVs. The removal of germline SNVs and artefacts of sequencing required further filtering. As published in ref. 7, we used pooled information across colonies and read counts from a matched germline WGS buccal sample to ensure that genuine somatic variants that may be present in the germline sample, either as embryonic variants or due to tumour-in-normal contamination, were also identified. Short insertions and deletions (indels) were called using cgpPindel[49] with the standard WGS cgpPindel VCF filters applied, except the F018 Pindel filter was disabled as it excludes loci of depth of less than ten. Copy number aberrations (CNA) were identified using ASCAT[50] by comparison to a matched normal sample or a wild-type colony from the same individual. The union of colony SNVs and indels was then taken, and reads counted across all samples belonging to the individual (colonies and buccal samples) using VAFCorrect. This allows mutations detected in any one or more colonies to be identified across all other colonies to fully capture the pattern of sharing of mutations across the colonies from an individual patient. This generates a data matrix of the number of reads supporting every mutation, depth of the sequencing of that site, and the variant allele fraction across every colony for a patient. Brass and GRIDSS pipeline[51] was used for calling structural variants. Both cgp-Pindel and Brass used PDv38is_wgs as an unmatched normal sample.

### Creating a genotype matrix

The genotype at every locus within each sample was 1 (present), 0 (absent) or NA (unknown). We inferred the genotype in a depth sensitive manner. We assumed the observed mutant read count for a colony at a given site was MTR ~ binomial($n$ = depth, $P$ = expected VAF), if the site was mutant, and MTR ~ binomial($n$ = depth, $P$ = 0.01), if the site was wild-type. The genotype was set to the most likely of the two possible states provided one of the states was at least 20 times more likely than the other. Otherwise, the genotype was set to missing (NA). The expected VAF was 0.5 for autosomal sites, but for chromosome X, Y and CNA sites, it was set to 1/ploidy. For loss-of-heterozygosity sites, genotype was overridden and set to missing if it was originally 0.

### Phylogenetic tree topology

We constructed phylogenetic tree topologies using maximum parsimony with MPBoot[52]. This method minimizes the number of changes required to reach the set of mutations assigned to each sample. The inputs for MPBoot were the binary genotype matrices with missing values per individual. These were exported as a multiple alignment fasta file with one line per colony with mutant represented as A, wild-type as T and missing as ?. The command line used was: mpboot -f <fasta> -bb 1,000.

### Donor with no wild-type colonies

No wild-type colonies were captured for PD57332. The tree was constructed as above, which resulted in just the mutant clade being present. Approximate features of the mutant 'truncal' branch were inferred as follows: (1) estimating the duration of the mutant clade as the average root to tip SNV burden divided by the cohort average *BCR::ABL1* SNV mutation rate, (2) the duration of the trunk was inferred (age at sample-height of the mutant clade) and (3) the length of the trunk in molecular time was then taken as the (duration of trunk) × (cohort wild-type SNV mutation rate) + (expected mutation burden at birth).

### Driver annotation

As each branch on a phylogenetic tree was assigned with SNVs and indels, it was possible to screen all branches for the presence of potential driver mutations. For this purpose, a previously[7] composed list of genes ($n$ = 35), common in clonal haematopoiesis and myeloproliferative disorders, was used: *ASXL1, BCOR, CALR, CBL, CSF3R, CUX1, DNMT3A, EZH2, GATA2, GNAS, GNB1, IDH1, IDH2, JAK2, KIT, KRAS, MLL3, MPL, NF1, NFE2, NRAS, PHF6, PPM1D, PTPN11, RB1, RUNX1, SETBP1, SF3B1, SRSF2, SH2B3, STAG2, TET2, TP53, U2AF1* and *ZRSR2*, with the addition of *ABL1* and *BCR*, totalling 37 genes. Branches with identified drivers were highlighted in colour on the phylogenetic trees. Annotation for *BCR::ABL1* fusion was added manually to the branch on the basis of its presence or absence in individual colonies from Brass and GRIDSS results.

### Timing branches

Given the linear accumulation of somatic mutations with age, we can infer the time point in life when driver mutations in phylogenetic trees had occurred. Branches at the top of a tree comprise mutations acquired at a young age, with branches lower down representing mutations arising later in life. We used our previously developed method rtreefit (https://github.com/nangalialab/rtreefit) for converting trees in

which branch lengths are expressed in molecular time (that is, number of mutations) into trees in which the branch lengths are expressed in units of time (years)[7]. In brief, the method jointly fits separate wild-type and mutant constant mutation rates (that is, number of SNVs accumulated per year) and absolute time branch lengths using a Bayesian per individual tree-based model under the assumption that the number of observed mutations assigned to a branch is Poisson distributed with mean = branch duration × sensitivity × mutation rate, and subject to the constraint that the root to tip duration is equal to the age at sampling. Furthermore, the method accounts for an elevated mutation rate during embryogenesis by assuming an excess mutation rate through development. In running rtreefit, the mutant clade was defined as not including the trunk, so the method assumed that BCR-ABL1 was acquired at the end of the trunk. The donor PD57332 had no wild-type samples so an in silico wild-type outgroup with a long branch (equal to 1,000 × cohort wild-type mutation rate) was added. This enabled PD57332 to be processed by rtreefit in a similar way to the other donors. The rtreefit algorithm was run with four chains and 20,000 iterations per chain. Mutations were assigned to the tree in a depth sensitive manner using treemut with mutations being hard-assigned to the highest probability branch (https://github.com/nangalialab/treemut). Branch lengths were adjusted for the branch-specific SNV detection sensitivity[7], in which the sensitivity of detection of fully clonal SNV variants was directly estimated from the per colony sensitivity for detecting germline heterozygous SNVs together with a multiplicative correction for the clonality (VAF) of the colonies. In calculating mutation burden and branch lengths, CNAs present in any colony in an individual were uniformly masked in all colonies for that individual and then the overall mutation burden was scaled back up by the reciprocal of 1 − expected number of mutations in the masked region. In addition to SNVs, indels and the BCR-ABL1 fusion, colonies showed a variety of CNA events. These events were curated as being present or absent in each of the colonies giving an event genotype vector like that obtained for SNVs and indels. Once the tree topology was inferred using the SNV genotypes, the branches that exactly matched the event genotype were identified and the event assigned to the corresponding branch.

### Quality control of a phylogenetic tree topology

The initial quality assessment step of phylogenetic trees included the removal of colonies for which the sensitivity of CaVEMan somatic mutation detection was below 60%. Colonies that might have been contaminated with cells from another colony were also excluded. If the colony was not clonal, then the mean VAF of SNVs assigned to its private branch would be lower than 50%. To ensure that samples were clonal, colonies were excluded if the VAF of SNVs that mapped to their private branches was significantly lower than 0.4. If the VAF of mutations assigned to a colony was not 0 in non-ancestral branches, then that colony was also removed because of this indicating that cells from this colony were contaminating other colonies.

### Per sample BCR::ABL1 mutation status

To ascertain BCR::ABL1 mutation status for each sample, we undertook a 'joint' genotyping methodology using GRIDSS. For each patient, Brass structural variant calls were reviewed for all events of interest pertaining to the BCR::ABL1 fusion event that included a core 'target' region consisting of the gene regions of ABL1 and BCR. For PD51635, we added RBFOX3 (chromosome 17), and for PD51633, we incorporated the large deletions present on der(9). For each patient, all samples were analysed together over the target regions to jointly identify evidence for structural variants. Per-patient results were reviewed and visualized with gGnome (https://github.com/mskilab-org/gGnome), which allowed the reconstruction of derivative chromosomes in patients with complex events such as PD51635. To ascertain the consequence of translocations, GRASS (Gene Rearrangement Analysis System, https://github.com/cancerit/grass) was used. In PD51635, a complex structural variant

t(9;17) event (inversion 'chr. 9 130776970GG]chr9:130786796' and translocation 'chr. 9 130777016T [chr17:79184121[ATT') was simplified to an equivalent single translocation event (chr. 9: +:130777016, chr. 17: +:79184121, AT) for annotation (Fig. 1d and Supplementary Table 2).

### Prediction of consequence of exonic BCR::ABL1 breakpoints

In three patients (PD56961, PD57332 and PD57335) we identified breakpoints in BCR exons (exon 14, 15 and 15, respectively). To identify stop codons, we continued the reading frame from the breakpoint, adjusting for any inserted bases within the translocation event. We used SpliceAI[53] to predict the splicing probabilities for each respective fusion sequence. Fusion sequences were reconstructed using GRCh38 reference sequence for BCR (chr. 22: 23170509–23328037) and ABL1 (chr. 9: 130825254–130897675) incorporating a 10-kb flanking sequence and any extra bases from the BCR::ABL1 translocation event (Extended Data Fig. 2a). Reconstructed fusion and reference sequences were used as input in the 'custom sequence' script (https://github.com/Illumina/SpliceAI), and 'raw' splice acceptor and donor probabilities from SpliceAI were converted to bedGraph format and reviewed on IGV[54].

### Mutational signature analysis

De novo mutation signature extraction was performed using HDP (https://github.com/nicolaroberts/hdp) with the mutations assigned to individual branches being treated as samples. Branches with fewer than 50 mutations were grouped into two per donor groups; those short branches that end before 100 mutations molecular time were pooled in the 'early life' group and the rest of the short branches were into the 'late life' group. These groups were also treated as samples. HDP extraction was then run across four chains with the following parameters for hdp_posterior: burnin=10,000, n=500, spacing=250. SBSblood signature[55] was downloaded and collated with the Pan-Cancer Analysis of Whole Genomes signatures (https://cog.sanger.ac.uk/cosmic-signatures-production/documents/COSMIC_v3.3.1_SBS_GRCh38.txt). De novo extraction identified three signatures SBSblood, SBS1 and SBS18, which showed the following cosine similarities to their respective published Pan-Cancer Analysis of Whole Genomes and/or SBSblood signatures: 0.927, 0.942 and 0.884. We refitted per-donor branch groupings and individual branches against the above published versions of SBSblood, SBS1 and SBS18. This signature attribution was carried out for each of these per-donor categories or branches using sigfit::fit_to_signature with the default 'multinomial' model. The per-branch attributions were then carried out by (1) assigning a per-mutation signature membership probability and then (2) summing these signature membership probabilities over all SNVs assigned to a branch to obtain a branch-level signature attribution proportion. The per-mutation signature membership probability was calculated using:

$$P(\text{mutation } \epsilon \text{ Sig}) = \frac{P(\text{mutation }|\text{Sig})P_0(\text{Sig})}{\sum_{\text{Sig}' \epsilon \{\text{SBS1,SBSblood,SBS18}\}} P(\text{mutation }|\text{Sig}')P_0(\text{Sig}')}$$

where the prior probability, $P_0$(Sig), is given by the mean Sigfit attribution probability of the specified signature, Sig, for the category that the mutation belongs to. The cohort level analysis of mutational signatures and C>T at CpG representation for branch categories was carried out using a random effects meta-analysis using the rma function in the 'metafor' R package.

### Growth rate estimation

The growth rate of BCR-ABL1 clones was estimated using the previously described phylofit approach[7,9]. In brief, phylofit is a Bayesian approach that estimates the growth rate by directly fitting a three-parameter logistic growth curve trajectory using the joint probability density of coalescence times given the population size trajectory. The three parameters estimated by this method are the saturation population

size $N$, the exponential phase growth rate $s$ and the midpoint of the curve $t^{(m)}$. Given that patients with CML present with a high *BCR::ABL1* fraction we set the upper bound of the prior for the midpoint to be the age at diagnosis and the lower bound to be the age of the MRCA of the mutant clade. We adopted uniform priors on the growth rate, $s$, and on the log scale saturation population size. Now, the annualized growth rate is given by $S = \exp(s) - 1$ and the uniform priors for $s$, midpoint $t^{(m)}$ and population size $N$, are:

$$s \sim \text{Uniform}(0.001, 30)$$
$$t^{(m)} \sim \text{Uniform}(\text{age of MRCA, age at diagnosis})$$
$$\log_{10}(N) \sim \text{Uniform}(4, 7)$$

The choice of upper bound for $t^{(m)}$ was motivated by the observation that patients with CML generally show a high *BCR-ABL1* burden at diagnosis. The input data for phylofit was the time-based trees obtained using rtreefit as described above. Note that the branch lengths of the input trees were chosen to be the mean of the posterior branch lengths. The trees were restricted to the earliest sampling point. These time points are all at diagnosis or after diagnosis. Comparison of estimates were checked using an alternative recently published method clone-Rate[24] as detailed in Supplementary Note 3.

### Telomere analysis
Telomerecat (v.4.0.2, https://github.com/cancerit/telomerecat) was used to estimate mean telomere length (bp) with (-t 75) to ameliorate the impact of NovaSeq sequencing artefacts (further details in Supplementary Note 2).

### Mixed models
Linear mixed models used for SNV burden and telomere analysis were implemented in the R package lme4 to estimate the impact of age and mutant status. Age (age_at_sample_exact) was defined as the count of completed years from birth at sampling and sample mutant status (*BCR_ABL1*) was defined as *BCR::ABL1* positive (Mt) or negative (Wt). For each response variable ($y$), we first tested the significance of mutant status (model 0 versus model 1) and then further terms were added to see whether this improved the model compared to the base model (model 1).

model_0 = $y \sim 1 + \text{age\_at\_sample\_exact} + (1|\text{patient})$
model_1 = $y \sim 1 + \text{age\_at\_sample\_exact} + BCR\_ABL1 + (1|\text{patient})$
model_2 = $y \sim 1 + \text{age\_at\_sample\_exact} + BCR\_ABL1$
$\qquad + (1 + \text{age\_at\_sample\_exact}|\text{patient})$
model_3 = $y \sim 1 + \text{age\_at\_sample\_exact} + BCR\_ABL1$
$\qquad + (1 + BCR\_ABL1|\text{patient})$

Models were fitted with default lme4 parameters. If a model did not converge, lme4::allFit() was used to refit the model to all available optimizers (provide by the lme4, optimx and dfoptim R packages), the best optimizer was selected from non-singular and converged refits with the highest negative log-likelihood. Only non-singular and converged models were considered for model selection using the Bayesian information criterion (BIC). For the final selected model, 95% confidence intervals (percentile bootstrap intervals) were calculated for each fixed effect, using confint(type='perc') from the first 1,000 converged and non-singular parametric bootstrapped models generated using the bootstrap() function from the R package lmeresampler, using a seed (1234) for reproducibility.

### SNV burden models
We first removed all samples ($n = 35$) from PD57332 as we were only able to grow *BCR::ABL1* colonies and therefore our estimations of SNV mutation burden (nsub_adj) were expected to be biased; this left 799

samples across eight patients. The final model used is shown below and was found to have the lowest BIC value (8,645.52), as detailed in Supplementary Note 1.

SNV model 3 = nsub_adj $\sim$ age_at_sample_exact + *BCR_ABL*1
$\qquad + (1 + BCR\_ABL1|\text{patient})$

### Telomere length models
To account for reported issues with telomere estimation using Telomerecat and NovaSeq sequenced data[9], we explored any batch effect of library preparation (library.cluster) and sequencing run (run_id.uniq) within wild-type samples ($n = 469$) on mean telomere length (length). The addition of sequencing run as an extra random effect (1| run_id.uniq) resulted in the lowest BIC model (7,483.33). This term was added to a series of models tested on the full dataset ($n = 834$), with the final model below having the lowest BIC value (13,265.15), as detailed further in Supplementary Note 2.

Telomere model 3 = length $\sim$ age_at_sample_exact + *BCR_ABL*1
$\qquad + (1 + BCR\_ABL1|\text{patient}) + (1|\text{run\_id.uniq})$

### Bulk DNA library preparation, sequencing and read alignment
DNA extracted from peripheral blood was subjected to WGS library preparation and sequenced paired-end, with 150-bp reads on Illumina NovaSeq 6000 machines. The reads were aligned to the human reference genome (GRCh38, NCBI) using the BWA-MEM algorithm.

### Bulk unmatched somatic mutation identification and filtering
SNVs were identified using CaVEMan[47] for each bulk sample by comparison to an in silico unmatched sample (PD38Is_wgs). CaVEMan was run with the 'normal contamination of tumour' parameter set to zero, and the tumour and normal copy numbers set to five and two, respectively. To increase sensitivity, SNVs only flagged as seen in a panel of normal samples ('VUM') were rescued. All SNVs were required to have less than half of supporting reads clipped (CLPM = 0) and a median BWA-MEM alignment score greater than or equal to 140 (ASMD ≥ 140). Short insertions and deletions (indels) were called using cgpPindel[48] with standard WGS cgpPindel VCF filters applied, except the F010 Pindel filter as it excludes variants seen in a panel of normal samples. Driver candidate variants were restricted to the 37 genes described above. To filter germline variants, we retained SNVs and indels with a gnomAD v.3.1.2 (ref. 56) popmax allele frequency less than 0.01 (annotated using echtvar v.0.2.0, ref. 57). The union of SNVs and indels was taken and reads counted across all samples belonging to the individual using VAFCorrect as above. Sequencing read depth was reviewed in IGV[54] to identify arm-level copy number events. GRIDSS v.2.13.1 was used for structural variant calling, and to estimate the VAF of *BCR::ABL1*.

### Bulk phylogeny reconstruction of PD60243
We used DPClust[58,59] to infer mutational clusters using SNVs (CaVEMan) and copy number and/or sample purity (Battenberg) called for each sample of PD60243a (81.3 years granulocytes, PD60243c 81.7 years mononuclear cells, PD60243d 81.7 years granulocytes) using a matched buccal sample (PD60243e). Manual inspection of the initial clusters identified 19 outlier poor quality variants that were subsequently removed for a second DPClust run with the addition of all *ABL1* mutations identified in the unmatched analysis. All four mutually exclusive *ABL1* mutations (as determined by read phasing) were grouped in the same cluster meaning we could not discern the exact subclonal phylogeny of *ABL1* mutations (which is expected with bulk reconstruction given the low VAF and 80× coverage). We inferred the total *ABL1* mutation burden at blast phase by adding the mutually exclusive *ABL1* mutations as individual subclones and removing their assigned cluster. Using ctree (https://github.com/caravagnalab/ctree)[60], we performed an exhaustive tree search (sspace.cutoff of 100,000) and retained the

top scoring trees that maintained *ABL1* mutation mutual exclusivity. We calculated the sum of *ABL1* cancer cell fractions (CCF), accounting for nesting, to obtain the CCF median and range.

## All of Us cohort analysis

The All of Us Research Program[29] is a US population-based cohort that links electronic health record data to WGS data for 206,173 participants, with an average genome-wide coverage of 30×. We searched CRAM files (Data repository v.7) to identify sequencing reads supporting a canonical *BCR::ABL1* (or reciprocal *ABL1::BCR*) variant between *BCR* (chr. 22: 23289313–23292813) and *ABL1* (chr. 9: 130674613–130874613). Reads that were incorrectly oriented or had insufficient mapping quality (score less than or equal to 6) were filtered. GRASS (https://github.com/cancerit/grass) was used to annotate sequencing reads that passed filters. GRASS takes pairs of genomic coordinates representing potential rearrangement events and predicts the fusion consequences along with their associated genes. We required the fusion event to be specifically between *BCR* and *ABL1*. After identifying *BCR::ABL1* carriers in the cohort, we extracted electronic health record data and searched for International Classification of Diseases (ICD) codes related to cancer, blood or immune diseases. In addition to searching for 'chronic myeloid leukaemia' (CML), we included conditions such as 'abnormal white cell count', 'basophilia', 'splenomegaly', 'unspecified cancer' or 'anaemia'. On the basis of these disease entries and domain knowledge, we categorized carriers into four groups: (1) CML mentioned, (2) abnormal white count (for example, leucocytosis), basophilia or splenomegaly mentioned, (3) other haematological issues (for example, anaemia) or unspecified cancer and (4) no relevant disease. Blood sampling date and the date of the most recent ICD code was used to define the time from sample collection to last follow-up. When calculating the incidence of CML and *BCR::ABL1* carrier status across age groups, we defined age at diagnosis for CML and age at blood sampling for *BCR::ABL1* carrier status. For each age group, summary statistics were calculated by averaging annual statistics from 2018 to 2022, assuming population structure remained stable during this period.

## Reporting summary

Further information on research design is available in the Nature Portfolio Reporting Summary linked to this article.

## Data availability

Sequencing files are available through the EGA (Dataset EGAD00001015353) in line with Wellcome Sanger Institute data sharing, and all somatic mutation .vcf files have been uploaded to Mendeley (https://doi.org/10.17632/yg29vx2f35.1). Use of individual-level data in the All of Us Program is available to researchers across the world through the Researcher Workbench, a cloud-based computing platform (https://www.researchallofus.org/register/). Summary-level data are available to the public through a data browser provided by the research programme (https://databrowser.researchallofus.org/). Source data are provided with this paper.

## Code availability

Code and software created for analyses are available at GitHub (https://github.com/nangalialab/CML).

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

**Acknowledgements** We thank the Cambridge Stem Cell Biobank for their invaluable support with sample collection and colony growth. We acknowledge the contribution of the Cancer, Ageing and Somatic Mutation laboratory and IT teams, and the Sanger core facility for their support with next generation sequencing. We thank J. Jones, H. Langoallen, the haemato-oncology diagnostics service and the East Genomics Laboratory Hub, Cambridge University Hospitals for diagnostic services. The Nangalia laboratory is supported by Cancer Research UK, Wellcome core funding at Wellcome Sanger Institute, Alborada Trust, the Leukaemia Lymphoma Society and Blood Cancer UK. We thank the patients for their participation in this study. We gratefully acknowledge All of Us participants for their contributions, without whom this research would not have been possible. We thank the National Institutes of Health's All of Us Research Program for making available the participant data examined in this study.

**Author contributions** The following authors contributed equally: A.E.K. and D.L.; N.W. and X.W. A.E.K., N.W. and D.L. performed all genomic analyses. K.D. assisted with modelling. J.L., K.N. and K. Milne assisted with patient selection, and sequencing. E.J.B. directed colony growth. K. Mane provided computational support. T.C. provided mentorship to A.E.K. A.R.G., P.J.C. and B.J.P.H. provided scientific advice. X.W., J.G. and P.T.E. analysed All of Us data. M.S.C. performed *BCR:ABL1* screening in colonies. A.E.K., N.W., D.L., X.W. and J.N. prepared the manuscript. J.N. conceived the study and directed the research.

**Competing interests** The authors declare no competing interests.

**Additional information**
**Correspondence and requests for materials** should be addressed to Jyoti Nangalia.

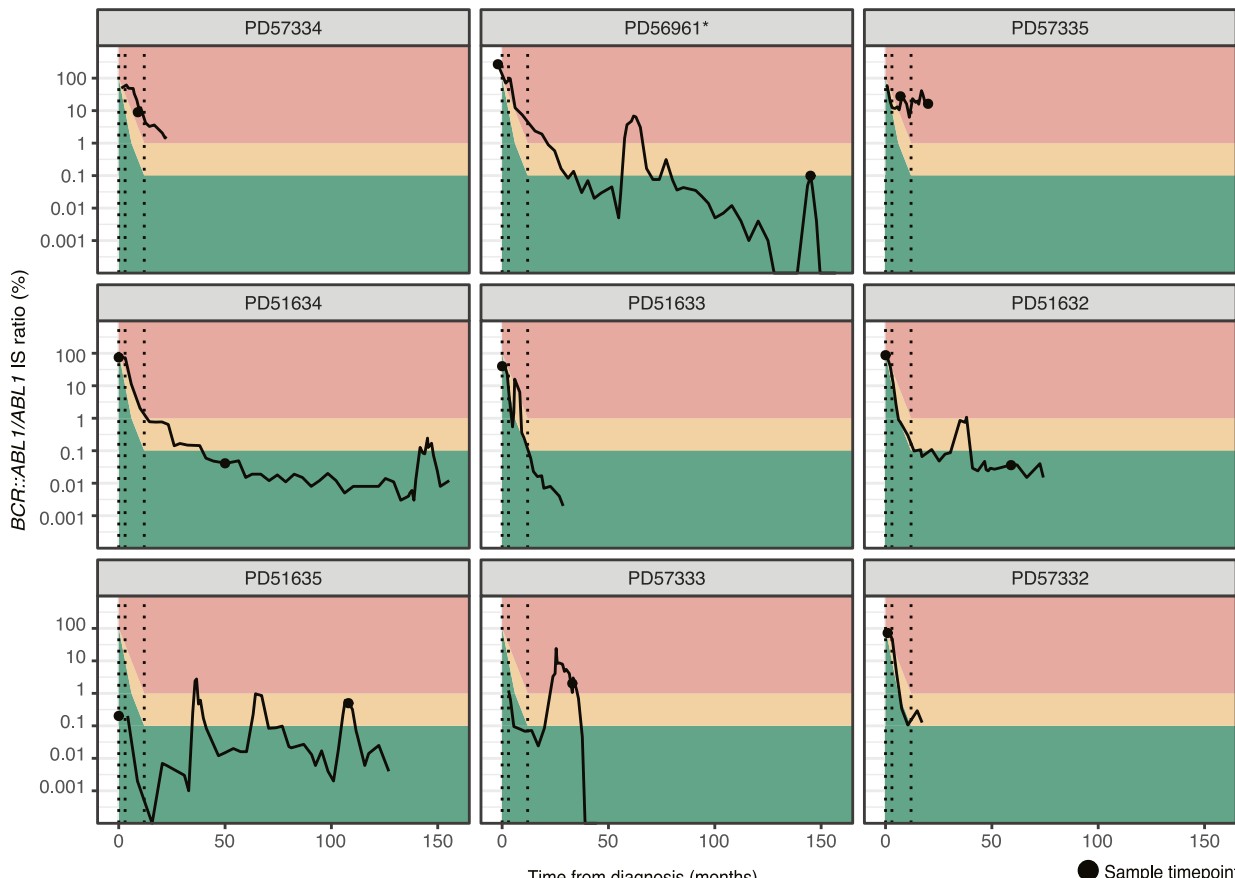

**Extended Data Fig. 1 | Patient cohort details with *BCR::ABL1/ABL1* ratios during treatment.** Plots showing the clinical *BCR::ABL1/ABL1* ratio (international standard levels) measured using RT-qPCR from diagnosis until last follow-up. The colours on the plot indicate an optimal (green), warning (yellow) or failure (red) of response according to European Leukemia Net recommendations, with vertical dotted lines showing the diagnosis timepoint and key post therapy timepoints (3 and 12 months) in assessing optimal response to tyrosine kinase inhibitor treatment. Black dots represent the sampling points for whole genome sequencing.

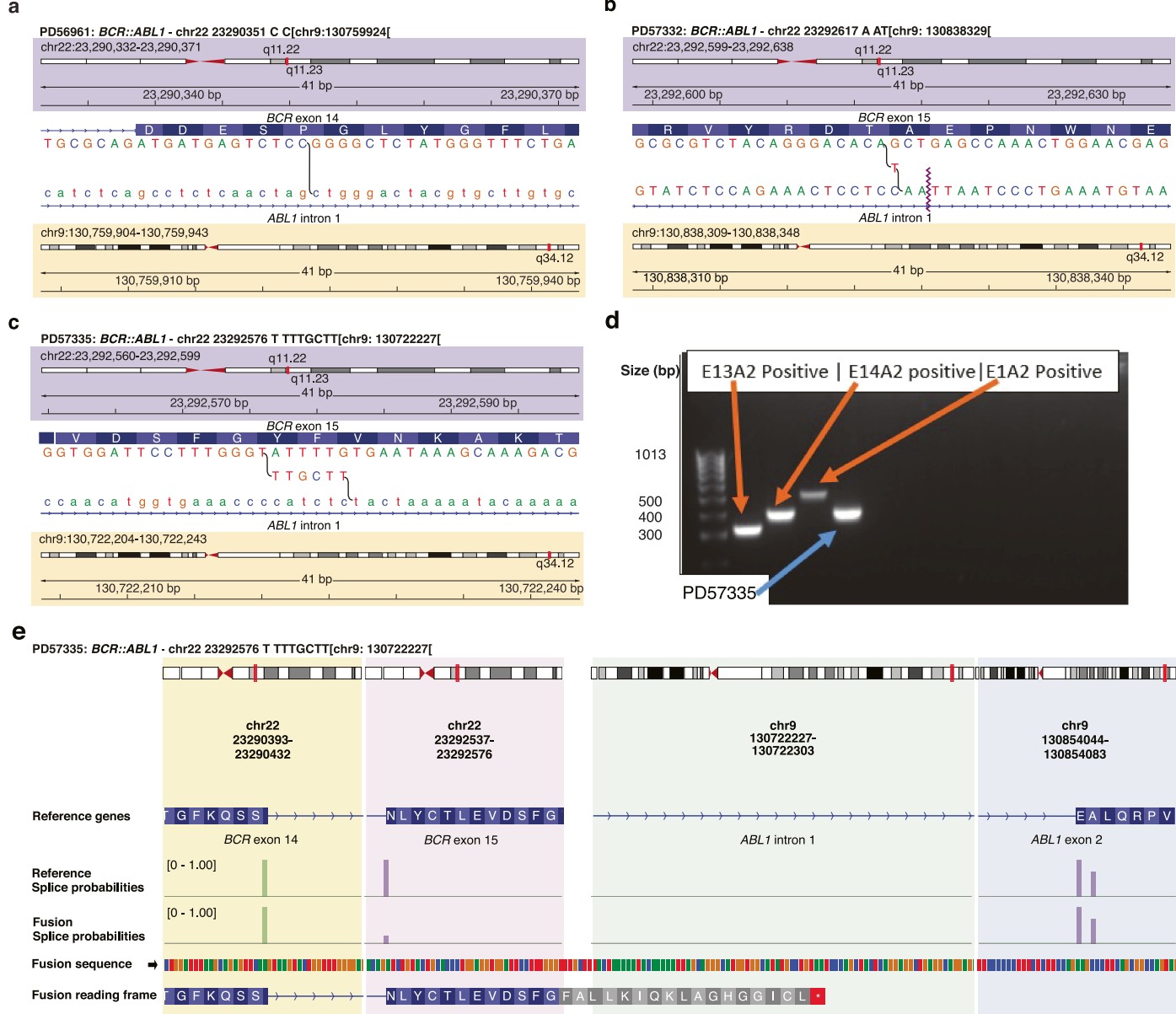

**Extended Data Fig. 2 | BCR::ABL1 event summary and exonic breakpoints exploration.** Panels **a-c**, show base pair resolution of the 3 fusions with exonic breakpoints: **a**. PD56961 **b**. PD57332 with an insertion of one nucleotide. Purple zigzag denotes the end of a fusion induced stop codon (TAA). **c**. PD57335 with an insertion of 6 nucleotides. **d**. Gel image of PD57335 BCR::ABL1 PCR product (well 5), showing that despite the breakpoint on chromosome 22 falling within exon 15, a standard e14a2 transcript was detected at the RNA level. Well 1 contains HyperLadder™ 100 bp (Meridian Bioscience) with positive control PCR products in wells 2–4 (E13A2, E14A2 and E1A2 respectively). The experiment was performed once. **e**. Splicing probabilities for PD57335 BCR::ABL1 fusion using SpliceAI. The vertical panels define 4 regions: the preceding exon to the chr22 breakpoint (BCR exon 14), the breakpoint within BCR exon 15, the intronic breakpoint within ABL1 intron1 and the following ABL1 exon 2. The rows describe: reference genes, reference splice probabilities between 0-1 for reference sequence with donor in green and acceptor in purple, fusion splice probabilities between 0-1 for reconstructed fusion sequence with donor in green and acceptor in purple, fusion sequence including 5 bp non-templated insertion, and fusion reading frame showing the consequence of the fusion sequence in grey, with the stop codon shown in red.

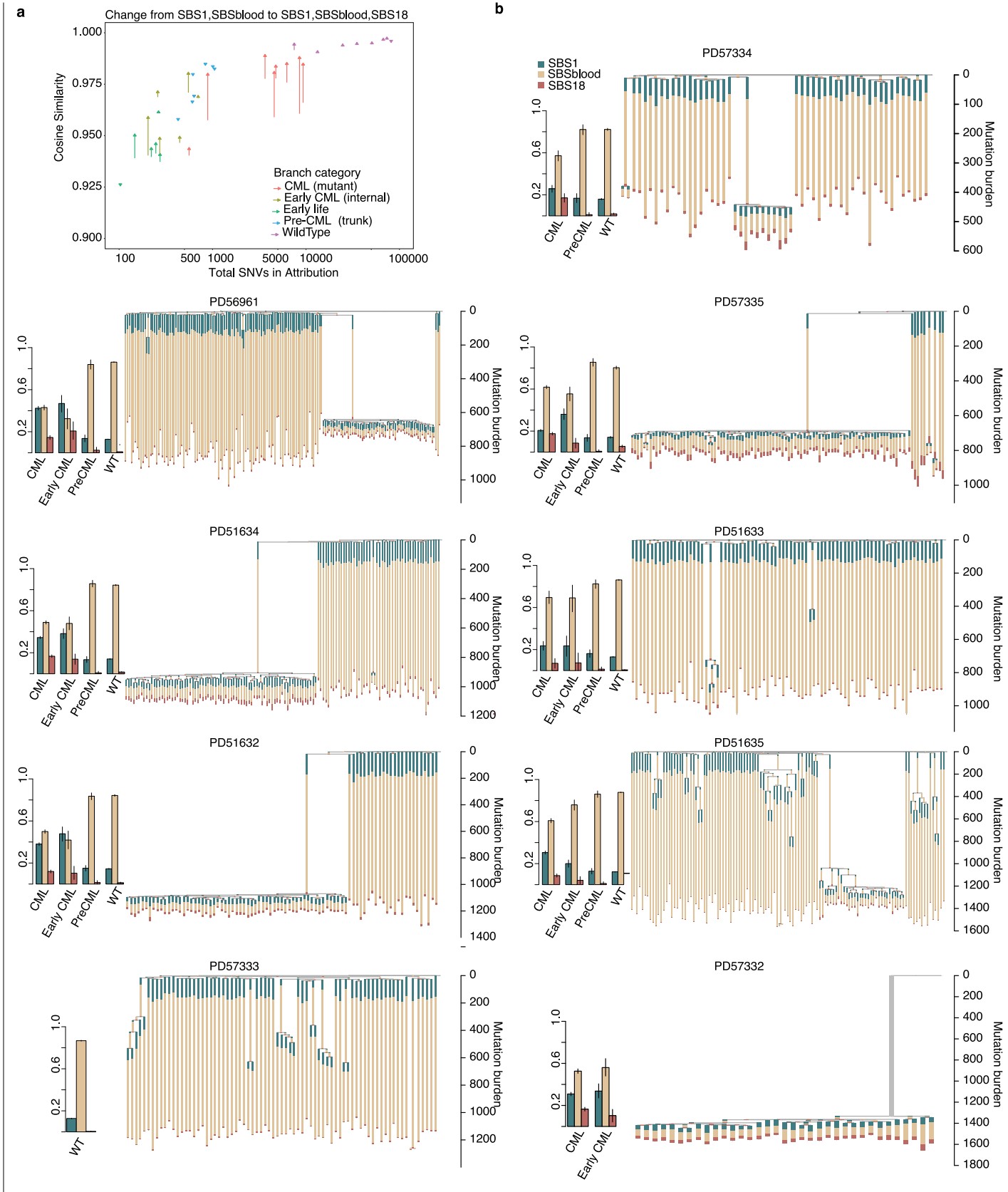

**Extended Data Fig. 3** | See next page for caption.

**Extended Data Fig. 3 | Phylogenetic trees with branch-specific mutational signature profiles. a**. The graph shows the cosine similarity for single nucleotide variant (SNV) attribution to a combination of mutational signatures "SBS1", "SBSblood" and "SBS18", compared to attribution to SBS1 and SBSblood only. The change in the cosine similarity between the *sigfit* based posterior mean reconstructed signature and the observed signature is indicated by the directionality of the arrows, showing that for branches within the CML clade, particularly, early branches of the clonal expansion, as well as for early-in-life mutations, the cosine similarity improves with the inclusion of attribution to "SBS18". Of note, there is little improvement in cosine similarity with the addition of SBS18 attribution for wildtype colonies or the trunk of the CML clade, ie., the pre-CML lineage, suggesting that SBS18 is not active outside of rapid expansion. Branches are categorised into five groups (i) branches of the *BCR::ABL1*-positive clonal expansion ("CML (mutant)"), (ii) ancestral branch of *BCR::ABL1*-colonies representing the lineage of origin of the CML ("Pre-CML (trunk)"), (iii) earliest mutations within the CML clade (ie., the shared branches within the clonal expansion, "Early CML (internal)"), (iv) branches of *BCR::ABL1*-negative colonies ("WildType, WT"), and (v) early-in-life branches ("Early life"). **b**. Phylogenetic trees of 9 patients with proportions of mutational signatures annotated to their branches. Bar plots show the proportion of SBS1, SBSblood and SBS18 contributing to SNVs spectra of different branch types. Bars represent posterior means with 95% credible intervals as error bars. The Y-axis shows the time in years since conception. The branches are categorised into four groups (i) branches of the *BCR::ABL1*-positive clonal expansion ("CML"), (ii) ancestral branch of *BCR::ABL1*-colonies representing the lineage of origin of the CML ("Pre CML"), (iii) the earliest mutations within the CML (ie., the shared branches within the clonal expansion, "Early CML"), and (iv) the branches of *BCR::ABL1*-negative colonies (wild-type, WT).

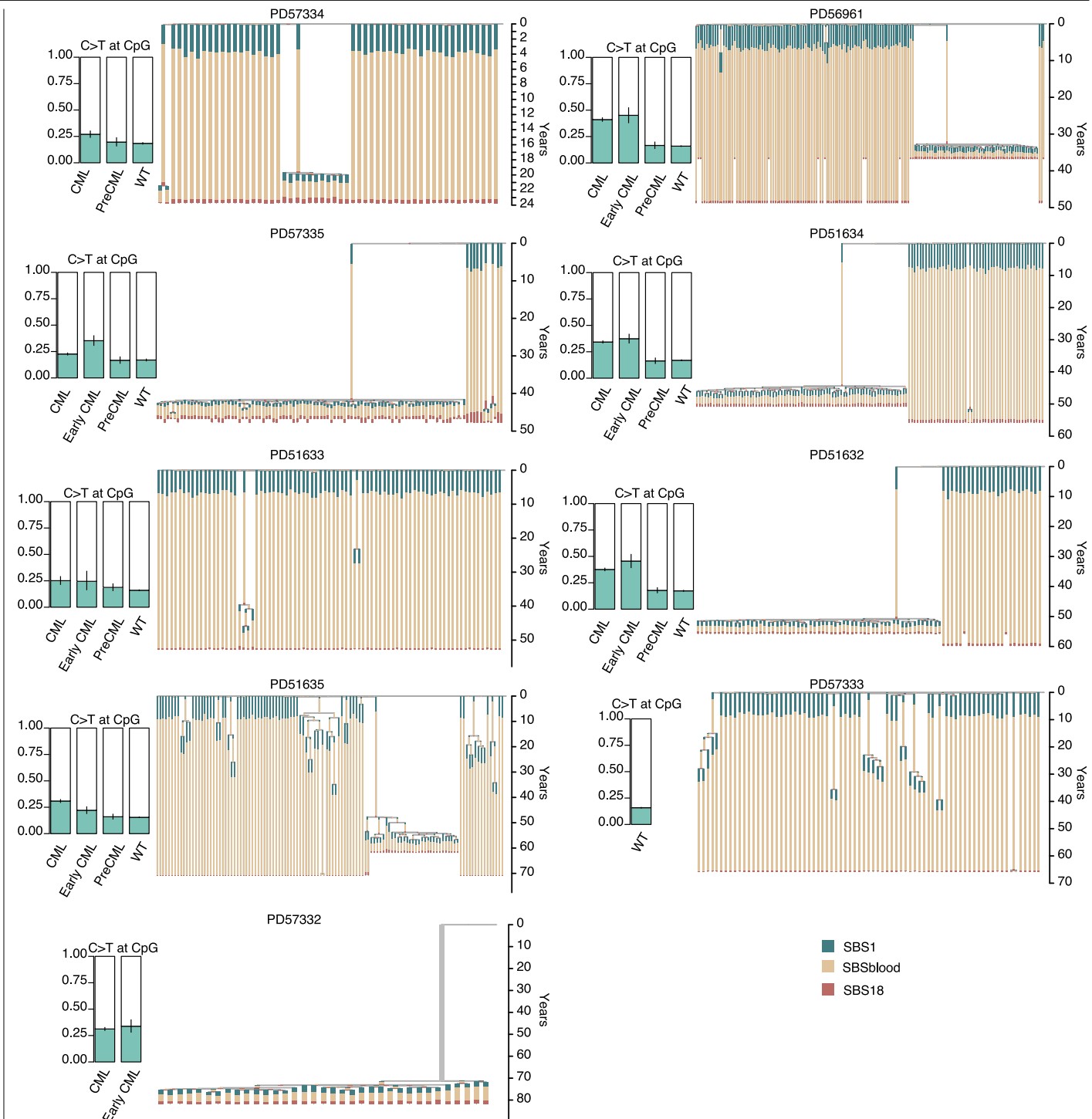

**Extended Data Fig. 4 | Molecular time phylogenetic trees with branch-specific C > T at CpG proportions.** Phylogenetic trees of 9 patients with proportions of mutational signatures annotated to their branches. Bar plots show the proportion of C > T at CpGs contributing to SNVs of different branch types. The Y-axis shows the somatic mutation burden since conception.

The branches are categorised into four groups (i) branches of the *BCR::ABL1*-positive clonal expansion ("CML"), (ii) the ancestral branch of *BCR::ABL1*-colonies representing the lineage of origin of the CML ("pre-CML"), (iii) the earliest mutations within the CML (ie., the shared branches within the clonal expansion, "early CML"), and (iv) the branches of *BCR::ABL1*-negative colonies (wild-type, WT).

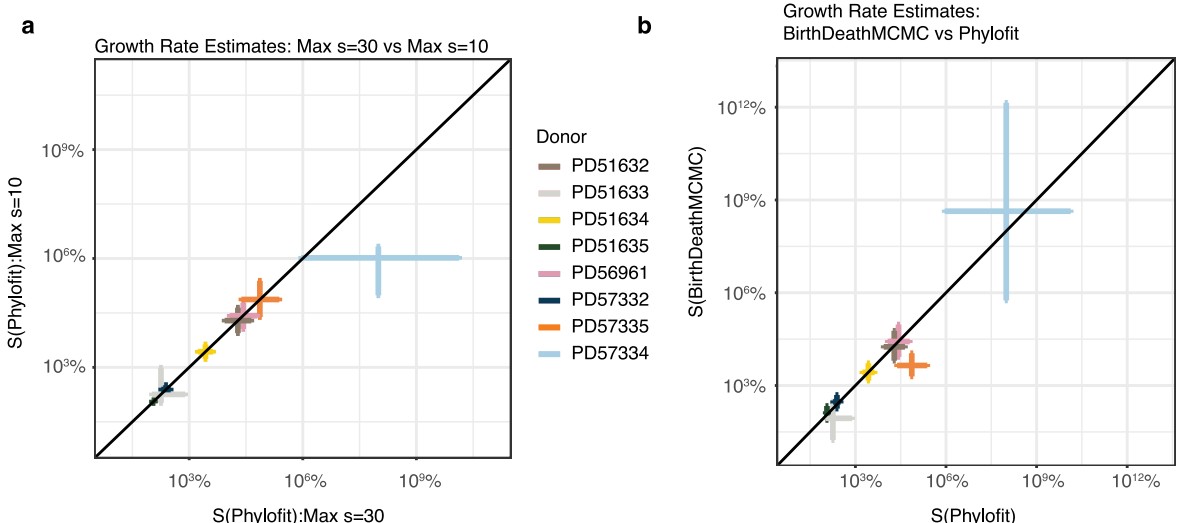

**Extended Data Fig. 5 | Validation of growth rates estimates. a**. Comparison of growth rates estimated using *phylofit* with two different maximum s values. **b**. Comparison of growth rates estimated using *phylofit* and the BirthDeathMCMC. Both methods are broadly consistent.

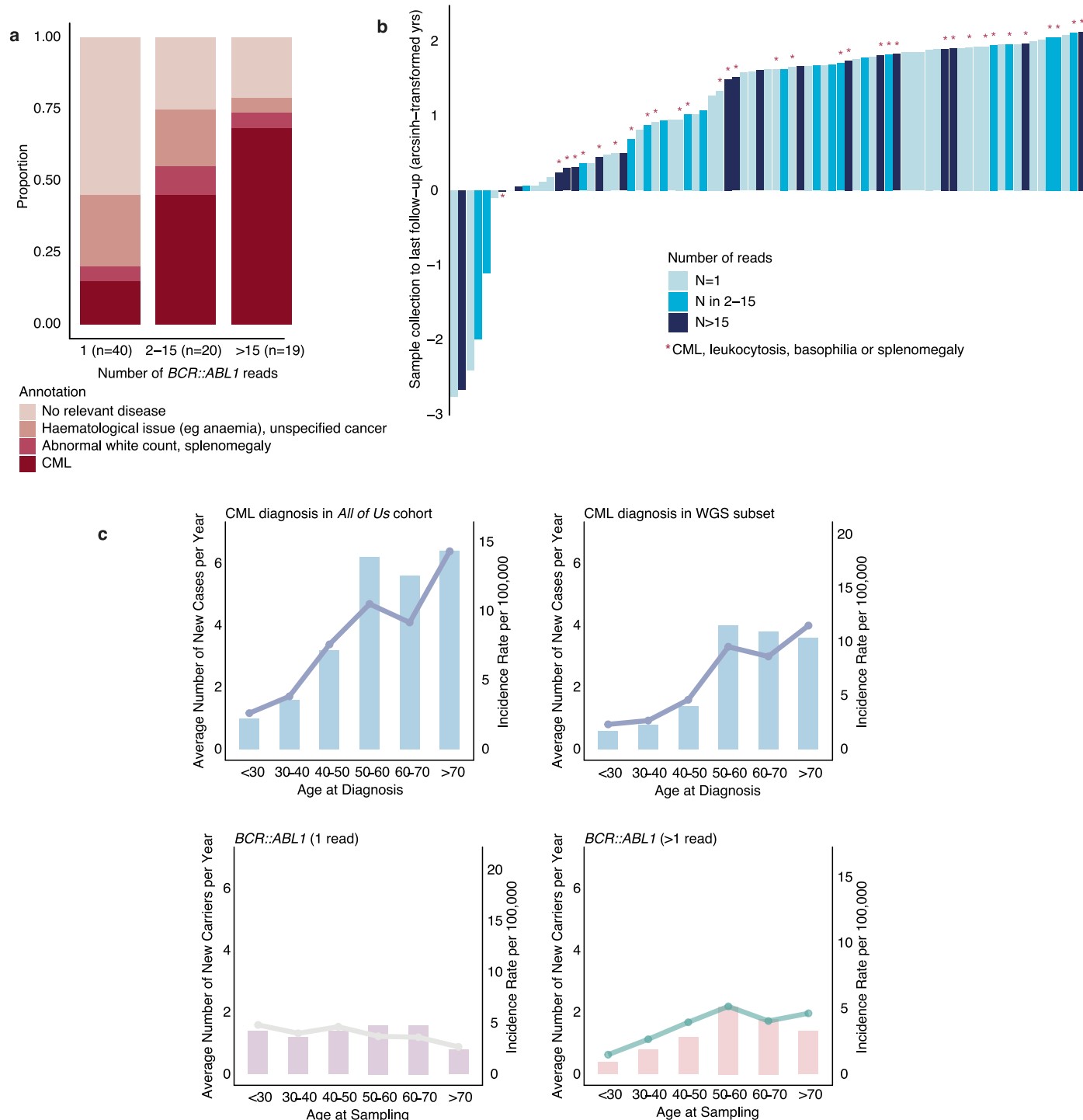

**Extended Data Fig. 6 | *BCR::ABL1* and CML in the *All Of Us* cohort. a**. CML diagnosis increases with *BCR::ABL1* supporting read count. **b**. Each bar in the plot represents a *BCR::ABL1* carrier identified, with the bar colour indicating the number of reads supporting their carrier status. Individuals diagnosed with CML or a pre-CML state during the follow-up period are marked with an asterisk. The Y-axis shows follow-up time, transformed using an arcsinh transformation (which approximates the natural logarithm while retaining zero-valued observations) for visualisation purposes. Exception received to the Data and Statistics Dissemination Policy from the All of Us Resource Access Board to display these data. **c**. CML incidence and *BCR::ABL1* carrier status across age groups. In each panel, the bars represent the average number of new CML cases or *BCR::ABL1* carriers per year, aligned with the left y-axis. The dots indicate the incidence rate per 100,000 population, corresponding to the right y-axis. For each age group, summary statistics were calculated by averaging annual statistics from 2018 to 2022, under the assumption that the population structure remained relatively stable during this period. WGS, whole genome sequencing subset of *All of Us* cohort.

# Reporting Summary

## Statistics

For all statistical analyses, confirm that the following items are present in the figure legend, table legend, main text, or Methods section.

| n/a | Confirmed | |
|---|---|---|
| ☐ | ☒ | The exact sample size (*n*) for each experimental group/condition, given as a discrete number and unit of measurement |
| ☐ | ☒ | A statement on whether measurements were taken from distinct samples or whether the same sample was measured repeatedly |
| ☐ | ☒ | The statistical test(s) used AND whether they are one- or two-sided *Only common tests should be described solely by name; describe more complex techniques in the Methods section.* |
| ☐ | ☒ | A description of all covariates tested |
| ☐ | ☒ | A description of any assumptions or corrections, such as tests of normality and adjustment for multiple comparisons |
| ☐ | ☒ | A full description of the statistical parameters including central tendency (e.g. means) or other basic estimates (e.g. regression coefficient) AND variation (e.g. standard deviation) or associated estimates of uncertainty (e.g. confidence intervals) |
| ☐ | ☒ | For null hypothesis testing, the test statistic (e.g. *F*, *t*, *r*) with confidence intervals, effect sizes, degrees of freedom and *P* value noted *Give P values as exact values whenever suitable.* |
| ☐ | ☒ | For Bayesian analysis, information on the choice of priors and Markov chain Monte Carlo settings |
| ☐ | ☒ | For hierarchical and complex designs, identification of the appropriate level for tests and full reporting of outcomes |
| ☒ | ☐ | Estimates of effect sizes (e.g. Cohen's *d*, Pearson's *r*), indicating how they were calculated |

*Our web collection on statistics for biologists contains articles on many of the points above.*

## Software and code

Policy information about availability of computer code

| | |
|---|---|
| Data collection | FASTQ files were generated by Novaseq sequencing machines. |
| Data analysis | Single-nucleotide substitutions (SNV) were called using the CaVEMan (Cancer Variants through Expectation Maximization) algorithm, version 1.15.2 (https://github.com/cancerit/CaVEMan). Filtering designed for quality control following processing through the Sanger low-input sequencing pipeline was also applied (https://github.com/MathijsSanders/SangerLCMFiltering). Small insertions and deletions were called using the Pindel algorithm as implemented in the cgpPindel workflow, version 3.10.0 (https://github.com/cancerit/cgpPindel). Copy number variants were called using the ASCAT algorithm as implemented in the ascatNgs workflow, version 4.5.0 ( https://github.com/cancerit/ascatNgs). Brass and GRIDSS pipeline (https://github.com/cancerit/BRASS and Cameron at al. 2017) was used for calling structural variants. Per patient structural variant results were reviewed and visualised with gGnome (https://github.com/mskilab-org/gGnome) this allowed the reconstruction of derivative chromosomes in patients with complex events. To ascertain the consequence of translocations grass (https://github.com/cancerit/grass) was employed. To further investigate the consequence of exonic breakpoints we used SpliceAI (https://github.com/Illumina/SpliceAI), to predict the splicing probabilities for each respective fusion sequence. Reconstructed fusion and reference sequence was used as input in the "custom sequence" script (https://github.com/Illumina/SpliceAI), "raw" splice acceptor and donor probabilities from SpliceAI were converted to bedGraph format and reviewed on IGV v2.17.4 (https://igv.org/doc/desktop/) release 2.17.4. De novo mutation signature extraction was performed using HDP (https://github.com/nicolaroberts/hdp). The SBSblood signature50 was downloaded and collated with the PCAWG signatures ( https://cog.sanger.ac.uk/cosmic-signatures-production/documents/ COSMIC_v3.3.1_SBS_GRCh38.txt). Allele counts at SNV and Indel sites were carried out using vafCorrect (https://github.com/cancerit/ vafCorrect). Mutations were mapped to phylogenetic branches using treemut ( https://github.com/nangalialab/treemut). Temporal branch lengths and per driver mutation rates were inferred using rtreefit (https://github.com/nangalialab/rtreefit). The growth rate of BCR-ABL1 clones was estimated using the previously described PhyloFit approach (Williams et al. 2022). Telomerecat (version 4.0.2, https://github.com/ cancerit/telomerecat) was used to estimate mean telomere length (bp). Unmatched somatic mutation identification and filtering: was |

performed with CaVEMan in addition to standard filters, SNVs flagged with "VUM" (seen in panel of normals) were rescued, all SNV were required to have a CLPM=0 and ASMD >=140. Short insertions and deletions (indels) were called using cgpPindel with the standard WGS cgpPindel VCF filters applied, except the F010 Pindel filter was disabled as it excludes variants seen in panel of normals. Driver candidate variants were restricted to the 37 gene set described. To filter germline variants we retained only SNVs and indels with a gnomAD v3.1.2 (Chen et al. 2024 ) popmax allele frequency < 0.01 from (annotated using echtvar v0.2.0 Pedersen & Ridder 2023). The union of SNVs and indels was then taken and reads counted across all samples belonging to the individual using VAFCorrect. Bulk phylogeny reconstruction: we used DPClust (https://github.com/Wedge-lab/dpclust) algorithm to infer mutational clusters using SNVs (CaVEMan filtered for proximity to indels called by PINDEL) and copy-number/sample purity (Battenberg https://github.com/cancerit/cgpBattenberg) called for each sample using a matched normal sample. Ctree (https://github.com/caravagnalab/ctree) was used to perform an exhaustive tree search. Random effects meta-analysis: rma function in the "metafor" R package. Code for analyses can be found at  https://github.com/nangalialab/CML.

For manuscripts utilizing custom algorithms or software that are central to the research but not yet described in published literature, software must be made available to editors and reviewers. We strongly encourage code deposition in a community repository (e.g. GitHub). See the Nature Portfolio guidelines for submitting code & software for further information.

# Data

Policy information about availability of data

All manuscripts must include a data availability statement. This statement should provide the following information, where applicable:
- Accession codes, unique identifiers, or web links for publicly available datasets
- A description of any restrictions on data availability
- For clinical datasets or third party data, please ensure that the statement adheres to our policy

Genome assembly GrCH38 is available at https://www.ncbi.nlm.nih.gov/datasets/genome/GCF_000001405.26/. PCAWG signatures are available at  https://cog.sanger.ac.uk/cosmic-signatures-production/documents/COSMIC_v3.3.l_SBS_GRCh38.txt. Sequencing files have been deposited in the European Genome-Phenome Archive (https://www.ebi.ac.uk/ega/home), under accession number EGAD00001015473 in line with Wellcome Sanger Institute data sharing, and all somatic mutation .vcf files will be uploaded to Mendeley  doi: 10.17632/yg29vx2f35.1 for publication. Use of individual-level data in the All of Us program is available to researchers across the world through the Researcher Workbench, a cloud-based computing platform (https://www.researchallofus.org/register/). Summary-level data is available to the public through a data browser provided by the research program (https://databrowser.researchallofus.org/).

# Research involving human participants, their data, or biological material

Policy information about studies with human participants or human data. See also policy information about sex, gender (identity/presentation), and sexual orientation and race, ethnicity and racism.

| Reporting on sex and gender | The sex of all participants is detailed in the study. |
| --- | --- |
| Reporting on race, ethnicity, or other socially relevant groupings | No variables on race, ethnicity or other socially relevant groups |
| Population characteristics | Samples were obtained from patients with chronic phase of chronic myeloid leukaemia treated at Cambridge Universities NHS Trust. Patients were selected to include a wide range of ages at diagnosis and variable treatment outcomes and were aged 22 to 81 years of age. 6 males and 3 females were studied. |
| Recruitment | All participants  were enrolled in the study "Causes of Clonal Disorders Study" following fully informed and written consent, and in line with the Declaration of Helsinki. Selection of patients was opportunistic based on hospital  attendance and sample availability, and therefore,  was biased towards attenders during sample collection, and those with samples already banked. The limitations and impact of sampling strategy is discussed in the paper. |
| Ethics oversight | The study was covered under NHS Research Ethics Committee approval 05/MRE/44 and 18/EE/0199. |

Note that full information on the approval of the study protocol must also be provided in the manuscript.

# Field-specific reporting

Please select the one below that is the best fit for your research. If you are not sure, read the appropriate sections before making your selection.

☒ Life sciences ☐ Behavioural & social sciences ☐ Ecological, evolutionary & environmental sciences

For a reference copy of the document with all sections, see nature.com/documents/nr-reporting-summary-flat.pdf

# Life sciences study design

All studies must disclose on these points even when the disclosure is negative.

| Sample size | No specific protocol was employed to establish the number of single-cell-derived colonies per patient prior to sample collection. We opted for more than 50 colonies per patient to ensure the inclusion of a significant number of both mutant and wild-type colonies for the calculation of |
| --- | --- |

mutation burden and tree-based HSC population growth parameters. 9 patients diagnosed with CML were chosen to represent different age groups at diagnosis and different treatment outcomes achieved.

| | |
|---|---|
| Data exclusions | Some single-cell-derived colonies were excluded from the analysis for quality control reasons or because there was evidence indicating that they originated from the same cell (technical replicate). |
| Replication | We performed a validation of growth rates estimates of the BCR::ABL1-mutated clone using different parameters for Phylofit and the maximum likelihood approach from Johnson et al. 2023. Replication of sequencing data involved sequencing multiple colonies from the same individual as detailed in phylogenetic trees, to increase confidence in parameter estimation. No wet lab experiments were repeated. |
| Randomization | N/A- this is a descriptive study with no test versus control groups. |
| Blinding | No blinding was undertaken in this descriptive study. Interpretation was of high throughput unbiased data generation. No subjective assessment was involved that required blinding. |

# Reporting for specific materials, systems and methods

We require information from authors about some types of materials, experimental systems and methods used in many studies. Here, indicate whether each material, system or method listed is relevant to your study. If you are not sure if a list item applies to your research, read the appropriate section before selecting a response.

## Materials & experimental systems

| n/a | Involved in the study |
|---|---|
| ☒ | Antibodies |
| ☒ | Eukaryotic cell lines |
| ☒ | Palaeontology and archaeology |
| ☒ | Animals and other organisms |
| ☒ | Clinical data |
| ☒ | Dual use research of concern |
| ☒ | Plants |

## Methods

| n/a | Involved in the study |
|---|---|
| ☒ | ChIP-seq |
| ☒ | Flow cytometry |
| ☒ | MRI-based neuroimaging |

## Plants

| | |
|---|---|
| Seed stocks | N/A |
| Novel plant genotypes | N/A |
| Authentication | N/A |

