## [Peer Review File · Nature]

Timing and trajectory of BCR::ABL1 driven chronic myeloid leukaemia

Corresponding Author: Dr Jyoti Nangalia

Version 0:

Reviewer comments:

Referee #1

(Remarks to the Author)

Thus us a very well written description of an extensive piece of work, using whole genome sequencing of single cell derived colonies from patients with chronic myeloid leukemia, to investigate the impact of the acquisition of the BCR::ABL1 oncogene on the proliferative capacity of affected cells. In contrast to previous work investigating other hematological malignancies and normal hemopoiesis in which driver mutations are acquired many years before the onset of disease, the authors identify rapid proliferation of the malignant clone and confirm the 'single-hit' origin of the disease.

This is an excellent study that provides new insights into a disease long-considered to be due to a single genetic event, and adds to the debate regarding the role of other acquired somatic mutations. The small number of patients is obviously a limitation and the absence of paired samples for chronic phase and progression to blast crisis is regretful, as the events leading to progression are still poorly understood. However the work is an important addition to the literature, and the authors had controlled their studies with currently available technology.

The authors looked at 9 patients of varied age and disease responsiveness but it is unclear how patient selection was made. Of the nine, only 3 responded to firstline therapy which is unusual in a disease in which fewer than 20% of patients are resistant to imatinib and less than 10% to second generation drugs. Since some of the patients were sampled only after treatment was initiated and the presence of BCR::ABL1 colonies was essential for the studies, was the cohort deliberately skewed towards poor responders? If so, how does affect their interpretation of their finding.

It would appear that some of the patients experienced loss of a previous response: were there any obvious explanations for this such as dose cessation for side effects or obvious non-compliance. If not, did the authors find any evidence of kinase domain mutations. If so, could they track their origin and selection?

(Remarks on code availability)

This is beyond my expertise and that of my current group

Referee #2

(Remarks to the Author)

This interesting study by Kamizela et al. investigates the genetic underpinnings of chronic myeloid leukemia (CML) by analyzing whole genome sequencing (WGS) data from single-cell-derived colonies from 9 patients. The study includes some unexpected findings such as variations in the expected gene breakpoints in BCR and ABL1 suggesting complex mechanisms of fusion transcript generation. The concept of patient specific selection landscapes in CML is very nice. Using phylogenetic analysis and mathematical models of clone growth, the study reveals that clonal expansion driven by BCR::ABL1 fusion begins often less than a decade before CML diagnosis, with explosive early growth rates. These findings are in line with CML incidence data from Japanese survivors of atomic bombs but contrast with typical cancer trajectories, indicating the unique oncogenic potency of BCR::ABL1 fusion in driving rapid clonal outgrowth. The authors also find that nearly all of the mutational signature contributions in these phylogenetic trees can be attributed to 2 endogenous clock-like signatures in blood. The data and methods are clearly described and the modeling work is rigorous. The work provides solid evidence for the aggressiveness of this 'one-hit' cancer, and underlines the clinical significance of dynamics like initial

growth rates on time to diagnosis and potentially response to TKI therapy. I have mostly minor comments and suggestions to be addressed focusing on the modeling applications, listed below.

Minor comments:

1. As shown in Figure S4, the upper end of the prior distribution for the higher estimates of clone growth rates is set to $S=10^8$. The corresponding s value for an S value of 10^8 is simply $s=13.8$, based on $\log(1+10^8/100)$. The highest simulated ground truth is $s=11.5$. In expected estimates derived from simulations, this prior is too restrictive for small n . Why do the authors not then make the prior for s uniform on $(0, 50)$ or similar? I would expect this to work just as well for the MCMC in practice, without bringing up the issue of potentially artificially inflating the performance of the methods (it is not clear what else might be causing the saturation of RMSE in final panel of Figure S4 for Phylofit at small n). This is an example that the reporting of annualized percentage growth 'S' for the clone growth rate throughout the paper is a bit confusing to interpret with how large the % becomes in some clones in the applications to CML. This distorts the estimate of instantaneous per cell per year growth rate, 's', driving the initial exponential clonal expansion in their model. We suggest at least in addition reporting 's' the per cell per year growth rate throughout, which is commonly reported and is easier to interpret for the analyses and results.
2. Supplementary Note 3 mentions that simulations indicate the expected time between acquisition and the MRCA of a large mutant clade is approximately exponentially distributed with a mean of $1/s$. There is actually mathematical justification for this when assuming a birth-death process which would be nice to include here along with simulations. Namely, the time that a lineage persists until it splits on a Yule tree is exponentially distributed with mean = $1/s$ (e.g., see Mooers et al. Syst Biol 2012).
3. The panel captions for Extended Figure 5 a and b are swapped.
4. Extended data 3 are not ultrametric trees as described in the caption title, and panel A is not explained in the caption.
5. Bottom of page 38 "See Figure S1 for the per donor clade specific mutation rates." Figure S1 currently does not appear to include the per donor clade specific mutation rates.
6. To be fixed - page 26 still says '(ref)'
7. There are 2 different Github links currently provided for rtreefit so it is not clear which was used for these results (NickWilliamsSanger and NangaliaLab). Please clarify.
8. Some repetition of ideas in last and third to last paragraphs of results. In the last paragraph, the results on doubling time are essentially the same as those for growth rates just explained in a different way. Further, the explanations in Supp Note 4 are somewhat confusing and long-winded. After providing Figure S5 with various assumed values for N , the authors show essentially in the log-log analysis that, regardless of the N values, the estimated stem cell number at diagnosis will not be the same across patients. It seems plausible that growth is not logistic with a carrying capacity equal to the total number of stem cells; it appears to saturate below a fraction of 1. Therefore, the idea that the latency is simply a function of growth rate may be incorrect and it is likely a function that also includes carrying capacity. The authors could clarify these observations in Supp Note 4.
9. Notes on patient-specific growth estimates:
 - PD57335: The estimates for both ages at sampling should be considered, as there is nothing in the tree to suggest that treatment significantly affects the mutation rate over the given year (I also think there should be 2 dots for this patient in Figure 1A?). This will allow for orthogonal estimates on the same clone with the second timepoint.
 - PD51633: This clone has 4 sampled cells which the authors show is too small for both methods' estimates. I suggest using the birth-death MCMC method from the cloneRate package for this sampled number. This small n also exacerbates the issue that confidence intervals for estimates for Phylofit are too narrow as the authors do note.
 - PD57334: Again here, I suggest exploring output using the birth-death MCMC method in cloneRate for $n=11$ sampled cells, as it is shown in Supp Note 3 that Maximum likelihood method can have a bias for small n sampled cells. Do estimates from the birth-death MCMC method fit closer with the Phylofit method?
 - PD57332: There is evidence of imbalanced selection (due to a subclone) that leads to different results with the lengths based method and maxLikelihood method using cloneRate (shown in Fig S2). I suggest that the RUNX1 subclone be measured separately. Measuring BCR::ABL1 clone ($n=5$, including one coalescence time from RUNX1 ancestral branch) will be noisy, but estimates can be made using Phylofit and birth-death MCMC in cloneRate (suggest this over maxLikelihood or Lengths based methods for $n=5$). General note of interest to mention: RUNX1 mutation in PD57332 allows the clone to reach full stem cell domination, not present in any other of the clones/patients.
 - PD51635: From the tree structure for the clone with BCR::ABL1, there appears to be subclonal expansions (but without known drivers) and then Tx drives a refractory clone without those subclonal expansions.
10. It would be nice to also include comparisons using cloneRate and Phylofit for clones other than BCR::ABL1 such as NF1 embryonic mutation PD51634, RUNX1 subclonal mutation in PD57332, and multiple clones in PD51635. This excellent dataset can also advance knowledge of somatic mosaicism/CHIP in general! Also, making the processed data easily accessible (namely, the mutation and ultrametric trees for each patient) upon publication will benefit the research community and future studies.

(Remarks on code availability)

The link <https://github.com/nangalia/CML> does not seem to work currently.

Referee #3

(Remarks to the Author)

In this manuscript, the evolutionary genomic dynamics of HSC lineages in Philadelphia translocation-positive (PhT+) CML patients have been characterized in nine individuals. The new data generated through WGS of single-cell colonies are valuable for exploring the mutation rate in PhT+ CHIPs and the timing of PhT acquisition, but they seem to have lower

sequencing depths than some recent studies (<https://doi.org/10.1038/s41586-022-04786-y> and <https://doi.org/10.1038/s41586-021-04312-6>). Anyway, the investigation is biomedically important, as the initiation of CH (leading to CML) by PhT acquisition would suggest that it could serve as a unique driver event, enabling more precise disease prediction and indication of imminent blood cancer.

The manuscript claims that the PhT acquisition, mapped on the shared branch of large clonal hematopoiesis (CH) in every patient, occurred at or near its most recent common ancestor (MRCA) or coalescence. Given the considerable length of these shared branches in the CHs, with numerous mutations also mapping to them, the authors are faced with the challenge of demonstrating that PhT acquisition was the critical event precipitating CH formation. This necessitates an ordering of polymorphisms along a lengthy shared branch—a task rendered impossible by the statistically indistinguishable frequencies of these variants. The current manuscript does not solve this problem. It has not convincingly mapped PhT acquisition onto MRCA, meaning that MRCA's timing only places the latest time bound on PhT acquisition rather than providing information on PhT's actual acquisition timing.

Additional considerations cause me to be skeptical.

Selecting CML patients with PhT for WGS sequencing means the universal presence of PhT across the samples does not substantiate the claim that its acquisition directly led to the CH event. Such a conclusion would require further evidence even with an unbiased sample of PhT+ and PhT-CMLs. For instance, should PhT emerge as the sole driver in PhT+ HSC lineages, it could be argued as the primary cause of CH initiation, but still, the longitudinal sampling of patients before they develop would be needed to be sure.

The finding that PhT+ CHs in CML patients mutate at an accelerated rate fails to associate PhT with the MRCA node specifically. Such a rate change is previously documented, with both healthy (in older age) and diseased (across all ages) individuals exhibiting more CHs and a faster accumulation of mutations (<https://doi.org/10.1093/molbev/msad279>). Therefore, faster mutation rates are linked with the presence of CHs in general, not uniquely with PhT+ CHs. Anyway, such rate changes cannot preferentially place PhT onto the MRCA node, as there is no way to exclude many other variants mapped to long shared branches.

While the patterns of telomere decay observed during CH-HSC diversification are reasonable, they do not aid in precisely mapping PhT to the MRCA node in the HSC phylogeny. Similarly, the identification of unique mutational signatures (SBS18) in expanding CML clades, although potentially diagnostic of PhT+ CML, does not preferentially map PhT to the MRCA, as compared to other mutations. It's noteworthy that a common SBS9 signature is reported in older age MPN patients in another study (<https://doi.org/10.1093/molbev/msad279>), suggesting that disease-related CHs can exhibit new signatures and such findings are not exclusive to PhT+ CML. Anyway, the same three mutational signatures are seen across every lineage in all patients' phylogenies (the aging markers SBSblood and SBS1, and the oxidative damage marker SBS18). The higher proportion of SBS18 observed in CML branches may simply be due to their recent coalescence, meaning that the same number of mutations would have been acquired under this regime in PhT- and PhT+ lineages, but they simply represent a smaller proportion of the total mutational burden in the longer, PhT+ lineages.

Ultimately, the detection of a PhT event in an individual has not been shown to be THE event causing CML CH, so the current results cannot claim to establish "the oncogenic potency of a single BCR::ABL1 fusion and its capacity to drive fast clonal outgrowth to cancer" without further evidence, such as longitudinal sampling of individuals who ultimately develop CML.

(Remarks on code availability)

The reproducibility of results and the adequacy of computational scripts provided could not be assessed because the somatic HSC variation data were not provided. In the past, authors of Nature manuscripts have provided such information freely for somatic variation datasets (as it is not germline).

Version 1:

Reviewer comments:

Referee #1

(Remarks to the Author)

I thank the authors for the additional work done in response to my comments and also for their expanded explanations of the limitations and interpretations of their results. I understand the limitations and feel that the authors have done as much as they can with available material and resources to improve the manuscript. I have no further suggestions

(Remarks on code availability)

Referee #2

(Remarks to the Author)

The authors have done an excellent job of addressing the Reviewers' comments and have conducted appropriate further analyses, some of which are now included in the paper and Supplementary. As a result, the manuscript is further

strengthened. This is a well conducted study that will be an important contribution to the field. I have a couple final minor comments for the authors based on their changes to address for clarity.

1. The authors now include thorough benchmarking of the different methods for growth rate estimation in the Supplementary, and there is one point of interest that they could expand upon to strengthen the paper. Figures S5c and S5d show that late coalescences will always cause cloneRate estimates to not align well with estimates from Phylofit. The authors state correctly that this is expected, and indeed removing the late coalescences when applying cloneRate recovers the correct growth estimate. In the real data presented however, estimates were typically found to be quite close using the 3 methods in 7 of 8 clones (only PD57335 does not have overlapping confidence intervals as the authors explain). This implies that consequential late coalescences are actually rare in their dataset (as also seen in trees in Figure 2). Alternatively, the authors simulate the test data in revised Figures S3 and S4 for a longer time period (until saturation with probability 99.99%), which I suspect causes late coalescences to occur with very high probability when using the fixed birth/death rates equal to 1 in the simulations. This scenario may not be reflective of most of the clone data in the paper, as observed above. Thus, these observations likely influence the validity of the following statement that the authors make in the rebuttal: "To better reflect the likely CML clonal dynamics, we have simulated the scenario that the clone growth continues beyond the saturation of the population (Rebuttal Figure 2 P=0.9999), in contrast to previous simulations where the clone growth generally halted towards the outset of population saturation (Rebuttal Figure 2 P=0.9)." If the authors could please expand on these observations with respect to saturation and turnover, it would help tease out plausible biological implications for the clone growth dynamics.

2. Thank you for adding the Supplementary Tables S3 and S4. It would be helpful to define what method was used for column G "estimate_cloneRateMoltime" in a Table S4 caption.

(Remarks on code availability)

I confirm that this is a usable resource, I downloaded and ran the code successfully.

Referee #3

(Remarks to the Author)

The authors' response confirms the previous concern that, using the reported dataset, it is impossible to precisely locate the timing of the BCR::ABL1 fusion (formation of the PhT) and the onset of CML. This is because all the SNVs occurring along the phylogenetic stem branch of an expanding HSC clade would bear a similar frequency, so their ordering could not be deconvoluted. The PhT may have happened at any point, not necessarily near the phylogenetic crown immediately preceding the expansion.

However, additional circumstantial evidence in favor of the rapid onset of CML after PhT acquisition is now provided in the rebuttal document. They surveyed data from the All of Us resource for the presence of PhT and CML symptoms/markers. They reported that most individuals with PhT (multi-read detections) show evidence of CML or associated symptoms, which could reasonably be interpreted as evidence that once the BCR::ABL1 fusion occurs, CML follows shortly after. This is a valuable result, and it lends some credence to the primary conclusion of this study.

However, it was surprising that the revised manuscript did not directly discuss the original caveats (articulated in the first paragraph above). Also, the new All of Us analyses are neither presented nor discussed in the main text and the Methods section (primary versus supplementary). (The only mention of "All of Us" appears as an author contribution attributed to the two newly added authors.) This needs to be remedied before the manuscript can be considered further. They are very important to critically evaluate the support for the authors' most impactful conclusions and assumptions in modeling.

In particular, there is a need to discuss population-level frequencies of PhT and CML in the main text and mention All of Us in the Methods to support their claims about timing and the assumptions in their modeling. The figures, detailed methodology, and results in the authors' rebuttal letter should be added in the main text, with details provided in the new Supplementary Note or Extended Data Figure.

(Remarks on code availability)

Version 2:

Reviewer comments:

Referee #2

(Remarks to the Author)

I thank the authors for their thoughtful Responses. Based on these, my last small suggestion is to please explicitly state in the text the chosen birth rate of wild type stem cells (1) or the starting death rate (1) for the rsimpop simulation described in Williams et al. to aid in reader comprehension of turnover that was simulated at clone saturation. It is a little ambiguous as it is currently written at the bottom of page 42, and this is important for downstream use cases of estimates of N and N effective.

I have no further suggestions and again I believe the revised paper is a very valuable and exciting contribution!

(Remarks on code availability)

As before, the code is reusable.

20th November 2024

Dear Michelle,

Many thanks for giving us the opportunity to present a revised manuscript and our apologies for the delay in returning this to you. The delay was caused by the acquisition and sequencing of additional samples (to address a comment for Referee 1), together with arranging institutional access to population cohort data (to address a comment for Referee 3). Please find below our point-by-point responses. Referee comments are in blue text. Our responses are in black text, with changes that we have made in red text. Text included from the manuscript is highlighted, with any changes we have made to the manuscript text in red text.

Referee expertise:

Referee #1: haematology; Referee #2: cancer evolution; Referee #3: phylogenetics

Referees' comments:

Referee #1 (Remarks to the Author):

-This is a very well written description of an extensive piece of work, using whole genome sequencing of single cell derived colonies from patients with chronic myeloid leukemia, to investigate the impact of the acquisition of the BCR::ABL1 oncogene on the proliferative capacity of affected cells. In contrast to previous work investigating other hematological malignancies and normal hemopoiesis in which driver mutations are acquired many years before the onset of disease, the authors identify rapid proliferation of the malignant clone and confirm the 'single-hit' origin of the disease.

We thank the Reviewer for their kind comments. We agree that these data for CML are in contrast with the slower trajectories to clonal haematopoiesis and other haematological malignancies, including other 'one-hit' blood cancers. We also agree that these data confirm the single-hit origin of this disease.

-This is an excellent study that provides new insights into a disease long-considered to be due to a single genetic event, and adds to the debate regarding the role of other acquired somatic mutations. The small number of patients is obviously a limitation and the absence of paired samples for chronic phase and progression to blast crisis is regretful, as the events leading to progression are still poorly understood. However the work is an important addition to the literature, and the authors had controlled their studies with currently available technology.

We appreciate the Reviewer's positive feedback. The Reviewer raises two points (1) low patient numbers and (2) lack of progression samples. We address these in turn.

(1) We fully accept that the current study has a small number of individual patients and that is a limitation. We have acknowledged this in the discussion section (page 11, para 3): However, our study is limited by the small number of individuals studied and the relationship between CML growth trajectories and clinical disease response merits validation at a larger scale. We have also acknowledged the small number of patients in two sections of the Results when (i) making inferences about growth rates and age at presentation, and (ii) growth rates and treatment response (page 8 para 4 and 6).

As we were attempting to estimate timing and growth rates in CML for the first time since earlier radiobiological estimates¹, we chose to study fewer patients and invest in studying each patient deeply. Therefore, we sequenced, on average, 93 whole genomes/patient (range 35-163 genomes/patient) to ensure confidence in our results on the estimated tumour growth trajectories and the clonal landscape of haematopoiesis for these patients. This scale of WGS limited the total number of patients we could study (>1000 whole genomes already), but we wish to assure the Referee that this number of patients, while small, is similar to recently published studies of clonal trajectories to haematological diseases or clonal haematopoiesis, which have generally studied 2-10 individuals²⁻⁶.

We fully agree that a larger study would be beneficial. We hope that information from the current study will be invaluable in informing the design of future studies so that we can measure CML timing, as well as speed of growth, at a larger scale. For example, given the single-hit, sudden and rapid clonal expansion of the CML clone, one can now potentially model the clonal trajectory from bulk WGS of blood at diagnosis, by looking at the rate of decline of the VAFs of subclonal variants in order to infer the growth rate of the tumour^{7,8}. A rapid decline in VAFs of mutations would suggest rapid CML growth, while a slower drop off in VAFs would imply slower clonal outgrowth generating many additional mutations with intermediate VAFs. Alternatively, if one sticks to the experimental approach of growing colonies, then one could grow far fewer colonies from diagnosis in the future, and build new mathematical approaches to estimate growth rates without the requirement of matched wildtype colonies, and with fewer mutant colonies. Lastly, given the telomere attrition in *BCR::ABL1*+ colonies, one could potentially estimate the increase in cell divisions beyond that expected for age through telomere measurements, and use this to infer growth rates and latency using the telomere estimation model that we have built in this study (Supplementary note 2) or from other works^{9,10}. We would be interested in collaborating on such a larger study in the future, the purpose of which would be to correlate tumour trajectories to clinical characteristics and therapeutic response, ideally using clinical trial cohorts from international studies. Such information could help tease out which patients with CML are more or less likely to respond to initial TKI therapy, in order to optimise their therapy at an earlier stage. We would also be interested in extending these insights into CML in different ethnic groups.

In the discussion, we have noted these methodologies that could enable larger scale future studies as follows (page 11, para 3): This could be achieved through the approach adopted here, sequencing a small number of single-cell derived *BCR::ABL1* genomes from CML patients sufficient to infer an accurate tumour growth rate, from telomere length^{43,44}, or from bulk WGS with inference of growth rates from the distribution of subclonal VAFs in the tumour^{45,46}.

(2) We note the comments from the Reviewer regarding the absence of samples at progression. Whilst the Referee did not ask us to include such samples in this study in a revision, we have nonetheless endeavoured to do so.

We have now undertaken whole genome sequencing (WGS) in four advanced CML cases (characterised by elevated blast cells between 10-19%). In all four cases, we observe the presence of additional driver mutations within the *BCR::ABL1*+ clone that have reached clonal dominance. Mutations observed are in *RUNX1*p.Arg166*, *ASXL1*p.Tyr591fs, *ASXL1*.p.G646fs*12 and *BCOR*p.Leu532fs (**Rebuttal Table 1**), and also include copy number events, such as, +1q, -16q, +8, isolated i(17q) (**Rebuttal Table 2**). Additional genomic events at advanced phase CML have been described and we see the previously observed pattern of *ASXL1* co-mutations occurring with additional clonal mutations^{11,12}. These additional drivers come to dominate the CML clone, with little evidence of the historical *BCR::ABL1* “only” clone, in keeping with previous observations. Our findings are in contrast to chronic phase CML within this study, where in the majority (6/8) of cases additional drivers within the *BCR::ABL1* +ve clone are either not present, or at low levels/subclonal (eg. in at most only 2 colonies per *BCR::ABL1* +ve clone). In the remaining two chronic phase cases, we did observe high clonal burden additional mutations; an *in utero* NF1 frameshift in PD51634, and a *RUNX1* frameshift mutation in PD57332. Intriguingly, PD51634 subsequently transformed to blast phase CML, and PD57332 died soon after initiating therapy with uncertainty clinically as to whether there was disease transformation due to absence of diagnostic tests. In summary, our additional data, albeit in only 4 individuals, uses whole genome sequencing to confirm that advanced phase CML is (i) largely genomically driven, and (ii) characterised by the presence of *BCR::ABL1* clones that have acquired additional driver mutations that confer a strong further selective advantage.

Rebuttal Table 1: Summary of driver mutations in four cases of advanced phase CML.

Patient	Driver event	Event	VAF [†]
PD60243a	BCR::ABL1	SV	0.396
PD60243a	DNMT3A.p.A571_A572delAA	DEL	0.5630
PD60243a	RUNX1.p.R166*	SNV	0.5075
PD60254c	BCR::ABL1	SV	0.415
PD60254c	ASXL1.p.G646fs*12	INS	0.5000
PD60254c	KIT.p.A616V	SNV	0.4607
PD60265a ^A	BCR::ABL1	SV	0.371
PD60265a ^A	BCOR.p.L532fs*57	DEL	0.8095 [#]
PD60277c	BCR::ABL1	SV	0.393
PD60277c	CUX1.p.A672V	SNV	0.4505
PD60277c	ASXL1.p.Y591fs*1	INS	0.3786

BCR::ABL1 VAF was calculated by GRIDSS and driver co-mutations VAF calculated using cgpVAF (methods). SV, structural variant; DEL, deletion; INS, insertion; SNV, single nucleotide variant. VAF is reported from granulocyte DNA where available, except for PD60265 where only a WBC sample was available at Dx. †The calculation of VAF derived from SVs such as *BCR::ABL1* incurs an increased reference bias due to increased uncertainty of defining true reference reads. This leads to lower VAFs compared to that calculated for SNVs and INDELS. #Note the variant is on chrX and this patient is male. Shaded rows of grey delineate different donors.

Rebuttal table 2: Chromosome arm-level events summary of advanced cases (n=4)

Patient	Chromosome arm-level events*
PD60243a	+8
PD60254c	+1q & -16q
PD60265a	-
PD60277c	+8, isolated i(17q)

*From manual review of WGS read depth data. Shaded rows of grey delineate different donors

We were able to obtain greater resolution into the origin and timing of the blast phase disease in one patient (PD60243, **Rebuttal Figure 1**). By performing WGS sequencing from matched buccal DNA to successfully remove germline mutations, as well as WGS of a follow-up sample at a blast crisis timepoint, we could interrogate the evolution of transformation. Bulk level WGS may not have the phylogenetic resolution that can be garnered from single-cell derived colonies, however, to its advantage, bulk WGS allows us to interrogate for the presence of very low burden pathogenic subclones (~1% VAF) due to the higher depth of sequencing. This resolution is not possible in our phylogenetic trees, where the number of colonies sampled, especially within the *BCR::ABL1* clade, is lower. Furthermore, bulk WGS allows surveying of clones that may not successfully grow *in vitro* to produce colonies, which is particularly relevant for blasts in advanced CML.

At the time of presentation with advanced phase CML (81.3yrs), PD60243 had evidence of elevated blasts (11% primitive cells expressing CD7, CD13, CD33, CD117 and HLA-DR). The *BCR::ABL1* clone harboured additional *RUNX1* p.Arg166* and *DNMT3A* p.Ala571_Ala572del, in line with the expected genomic evolution of the CML clone at the time of advanced phase presentation (**Rebuttal Figure 1**). This clone with three driver mutations shared at least 1259 mutations. Given the age of the patient (81yrs), this suggests that the most recent common ancestor of the clone (i.e., the most recent clonal sweep following acquisition of the 3rd driver mutation) arose relatively recently (e.g. the 7th decade of life).

There were two additional subclones detected within this multiply mutated CML clone: (i) one subclone with trisomy 8 (56% of *BCR::ABL1/RUNX1/DNMT3A* positive cells) and the other with an in-frame *PHF6* p.H329delH (14% of *BCR::ABL1/RUNX1/DNMT3A* cells) (**Rebuttal Figure 1**). Within a few months of treatment with Nilotinib, the patient developed ~20% immature blasts in the peripheral blood and at this timepoint (81.7yrs), the *PHF6*-mutated subclone was now clonally dominant, revealing further focal events; *BTG1* deletion (chr12), rearranged *TRB* locus (chr 7), and rearranged *IgH* on chr 14 - all typical changes of B-cells, in keeping with lymphoid blast crisis. This confirms that in this individual, lymphoid identity and differentiation arose after *BCR::ABL1* acquisition via RAG-mediated genomic evolution. There were 5 different *ABL1*-mutated subclones within this highly mutated *BCR::ABL1* clone at blast phase, with a dominant *ABL1*.p.Phe359Val clone (53% of cells) and 4 remaining subclones (*ABL1*.p.Q252H/p.Y253H/p.E255K/p.E255V) which we can define as independent clones (using read-pair evidence and therefore accounting for a total of 42% of cells). Our tree reconstructions suggest that 1-3 *ABL1*-mutated clones may have occurred in cells that already harboured *ABL1*.p.Phe359Val, consistent with the notion that cells may harbour more than one *ABL1* mutation. Overall, we estimate that *ABL1*-mutated clones totalled 57-90% of cells at blast phase (**Rebuttal Figure 1**). These additional subclones with *ABL1* mutations had several hundred additional genome-wide mutations (n=204), this is incompatible with the time that had passed (0.4yrs), suggesting either acquisition prior to advanced phase, or rapid clonal outgrowth and genomic instability during blast phase progression.

Rebuttal Figure 1 (new Fig.4e/f): Fish and tree plot describing the broad architecture of clones found at Advanced phase (81.3yrs) and Blast phase (81.7yrs) in PD60243. All *ABL1* mutant clones with mutations affecting amino acids 252-253 (n=4) are confirmed to be in different cells using read phasing. Tree reconstructions predict a median of 2 (range 1-3) of these *ABL1* mutant clones nested within the *ABL1* p.F359V clone, which we here show

an example of. Note that while the Fish plot (e.) y-axis reflects the inferred cancer cell fraction (CCF) of the clones at each measured time-point, the displayed emergence points of clones are for illustrative purposes. Grey shaded components at the left of the fishplot (dotted and lined) represent historical mutant clones with a single (grey dotted clone) and two (grey dashed clone) driver mutations from BCR::ABL1, RUNX1 and DNMT3A - the ordering of these events is unknown. The tree plot (f.) shows the same tree solution as e. but allows the annotation of autosomal SNV burden for the respective subclones. Dashed lines represent a possible branching structure. Boxes are labelled with respective clonal fractions.

We now include the figure and legend above from PD60243 in Figure 4e/f, showing the clonal evolution to advanced and blast phase disease. We have added these recent results as a new section called “Disease progression of advanced phase CML” that commences on page 9, and the drivers detected in Supplementary table 6.

We acknowledge that the number of advanced phase patients that we have studied is very small, but our findings are largely in line with previously reported studies. Further studies of advanced and chronic phase CML using single cell derived sequencing is a longer term effort and outside of the scope of the current manuscript.

The authors looked at 9 patients of varied age and disease responsiveness but it is unclear how patient selection was made. Of the nine, only 3 responded to firstline therapy which is unusual in a disease in which fewer than 20% of patients are resistant to imatinib and less than 10% to second generation drugs. Since some of the patients were sampled only after treatment was initiated and the presence of BCR::ABL1 colonies was essential for the studies, was the cohort deliberately slewed towards poor responders? If so, how does affect their interpretation of their finding.

We thank the Reviewer for raising this important point. The patients were not selected at random but primarily to include a broad range of ages of individuals with CML and responses to therapy, thus capturing the full clinical spectrum of the disease. We first focussed on four patients who at diagnosis and in remission had responded to the first-line TKI (PD51632, PD51633, PD51634, PD51635). Only out of necessity, were remission samples required in order to capture BCR-ABL1-negative colonies if these were not identified at the diagnostic timepoint. As the patterns emerging of explosive clonal outgrowth appeared similar for these first few patients, we then sought to identify more unusual patients, e.g. those that did not respond to first-line therapy, responded more slowly, or developed side effects on the first-line TKI (PD57334, PD57335 & PD57333, PD56961), in order to capture a wider spectrum of possible growth and genomic evolution trajectories. In some of these patients (PD57334, PD51633), given that both BCR-ABL1 positive and negative colonies were identified from a single time-point, we did not seek further samples. Lastly, we added an elderly patient PD57332, despite not having wildtype colonies, to encompass the full age range. We have also added details of how patients were selected to the methods section (patients and sample acquisition) as follows (Methods, page 24): “We selected patients who responded to the first-line TKI (PD51633, PD51632, PD51635, PD57332) and patients who had a more unusual treatment response (non-responders to the first line TKI, patients who developed side effects on the first line medication (PD57334, PD57335, PD57333), and slow responders (PD56961, PD51634).”

Given that several of our non-responders happened to be aged <50 years, it is possible that this study might inadvertently give the impression that young patients are more likely to be non-responders. Whilst this may or may not be true, our study is not sufficiently powered to address the question of whether younger patients are more likely to be non-responders. Similarly, our data raise the possibility that younger patients have faster growth rates. Therefore, we have adjusted the text to make clearer that the number of patients is too small to make any definitive correlations between age, growth rates and response to TKI pending characterisation in a larger study (page 11, para 2 and 3).

-It would appear that some of the patients experienced loss of a previous response: were there any obvious explanations for this such as dose cessation for side effects or obvious non-compliance. If not, did the authors find any evidence of kinase domain mutations. If so, could they track their origin and selection?

The reviewer raises an important point. **We have now incorporated the context of the loss of previous MMR in Supplementary Table 1.** For patients PD51632 and PD57333 there was no clear explanation for the loss of MMR, hence, we have labelled it as an idiopathic loss of response. For the remaining patients the culprit of the loss of MMR were: a trial of dose de-escalation (PD56961, PD51635), treatment cessation during pregnancy (PD56961), and TKI related side effects limiting dosage (PD51634, PD51635).

We did not identify kinase domain mutations in these patients, but we did not sample the tumour clone post loss of MMR when we would have been most likely to identify such mutations. Indeed, capturing *BCR::ABL1* positive colonies when MMR has only just been lost (*BCR::ABL1/ABL1* ratio >0.1% limit of MMR loss) would have been very challenging due to the rarity of *BCR::ABL1* positive cells. Therefore, the absence of such mutations in the diagnostic sample is not particularly informative given the lack of sensitivity we would have to identify rare mutations (eg 1 in <150 HSC frequency) in the diagnostic sample using our phylogenetic approach which only sampled 4-75 mutant *BCR::ABL1* cells. Of note, we do find multiple kinase domain mutations in high depth WGS sequencing of a blast phase patient, as discussed above, suggesting that such clones are abundant at a very low level, potentially even before therapy commences¹³, and that they are selected for by TKI therapy to drive clonal evolution characteristic of more advanced disease.

Referee #1 (Remarks on code availability):

This is beyond my expertise and that of my current group

Referee #2 (Remarks to the Author):

This interesting study by Kamizela et al. investigates the genetic underpinnings of chronic myeloid leukemia (CML) by analyzing whole genome sequencing (WGS) data from single-cell-derived colonies from 9 patients. The study includes some unexpected findings such as variations in the expected gene breakpoints in BCR and ABL1 suggesting complex mechanisms of fusion transcript generation. The concept of patient specific selection landscapes in CML is very nice. Using phylogenetic analysis and mathematical models of clone growth, the study reveals that clonal expansion driven by *BCR::ABL1* fusion begins often less than a decade before CML diagnosis, with explosive early growth rates. These findings are in line with CML incidence data from Japanese survivors of atomic bombs but contrast with typical cancer trajectories, indicating the unique oncogenic potency of *BCR::ABL1* fusion in driving rapid clonal outgrowth. The authors also find that nearly all of the mutational signature contributions in these phylogenetic trees can be attributed to 2 endogenous clock-like signatures in blood. The data and methods are clearly described and the modeling work is rigorous. The work provides solid evidence for the aggressiveness of this 'one-hit' cancer, and underlines the clinical significance of dynamics like initial growth rates on time to diagnosis and potentially response to TKI therapy. I have mostly minor comments and suggestions to be addressed focusing on the modeling applications, listed below.

We thank the Reviewer for their kind comments on the study. We particularly appreciate their detailed review of our modelling strategy and their helpful comments which have led to an improved manuscript. We address the points as detailed below.

Minor comments:

1. As shown in Figure S4, the upper end of the prior distribution for the higher estimates of clone growth rates is set to $S = 10^8$. The corresponding s value for an S value of 10^8 is simply $s = 13.8$, based on $\log(1 + 10^8/100)$. The highest simulated ground truth is $s = 11.5$. In expected estimates derived from simulations, this prior is too restrictive for small n . Why do the authors not then make the prior for s uniform on $(0, 50)$ or similar? I would expect this to work just as well for the MCMC in practice, without bringing up the issue of potentially artificially inflating the performance of the methods (it is not clear what else might be causing the saturation of RMSE in final panel of Figure S4 for Phylofit at small n). This is an example that the reporting of annualized percentage growth 'S' for the clone growth rate throughout the paper is a bit confusing to interpret with how large the % becomes in some clones in the applications

to CML. This distorts the estimate of instantaneous per cell per year growth rate, 's', driving the initial exponential clonal expansion in their model. We suggest at least in addition reporting 's' the per cell per year growth rate throughout, which is commonly reported and is easier to interpret for the analyses and results.

We thank the Reviewer for their comment. We agree with the Reviewer's diagnosis of the reason for the small n saturation of RMSE in Figure S4. The upper end of the range is $S=10^8$, but this corresponds to $s=\log(1+10^8)=18.4$ as opposed to $\log(1+10^8/100)=13.8$. We take the reviewer's point that expressing S as a percentage is potentially confusing, and so have now added s as well as S throughout the revised manuscript when referring to growth rates of clones. We have increased the upper end of the prior range to $s=30$ ($S=10^{13}$) for Phylofit inferences, both for simulations and for the inferences of *BCR::ABL1* clones based on the donor experimental data (**Rebuttal Figure 3**, left panel). In addition we noticed that Phylofit was treating polytomies as a single coalescence, and since we are estimating very rapidly growing clones we thought it necessary to update the algorithm to separately include each coalescence of the corresponding bifurcating tree (*ape::multi2di*). We take this opportunity to also include the *cloneRate::birthDeathMCMC* in our benchmarking with the same prior range for s. To better reflect the likely CML clonal dynamics, we have simulated the scenario that the clone growth continues beyond the saturation of the population (**Rebuttal Figure 2** $P=0.9999$), in contrast to previous simulations where the clone growth generally halted towards the outset of population saturation (**Rebuttal Figure 2** $P=0.9$).

Rebuttal Figure 2. Ten simulated trajectories when the simulation end point is set such that there is (left panel) 90% chance a constant birth-death process has reached $N=100,000$ and (right panel) 99.99% probability it has reached 100,000. The simulated growth rates were fixed at $S=10,000$ ($s=9.2$).

We have updated Extended Data 5, where we now compare *phylofit* to *birthDeathMCMC* rather than to *cloneRate::maxLikelihood*, as shown below, and we also compare the growth rate estimates for Max $s=30$ and Max $s=10$ (**Rebuttal Figure 3**).

Extended figure 5

Rebuttal Figure 3 (updated Extended Figure 5a,b) a. Comparison of growth rates estimated using Phylofit with two different maximum S values. **b.** Comparison of growth rates estimated using Phylofit and BirthDeath MCMC maximum likelihood approach from Johnson et al³¹. The methods are broadly consistent, but with very different estimates for PD57334 where both methods infer extremely rapid growth rates but nonetheless differ by an order of magnitude.

We also update Figure S3 and S4 to include birthDeathMCMC, as shown below:

Rebuttal Figure 4 (updated Figure S3) Root Mean Square Error in Growth Rate Estimation. This plot shows that the cloneRate methods and Phylofit are similarly accurate in their central estimate of growth. For very low values of n (4 and 11) none of the methods performs well, but birthDeathMCMC and Phylofit are more accurate.

Rebuttal Figure 5 (new Figure S4) Mean Error in Growth Rate Estimation. This shows that for saturated simulated trajectories the cloneRate methods tend to underestimate “ s ” whilst Phylofit overestimates “ s ” but tends towards the correct estimate for large n .

Generally we find that the two cloneRate methods (birthDeathMCMC and maxLikelihood) tend to give similar estimates and are well calibrated for making inferences about phylogenies sampled from a pure exponential growth process. However, they sometimes fare less well in the presence of perturbations to the population around sampling either due to population saturation, or because of TKI treatment. It is noticeable that the *Phylofit* method tends to give estimates that are less dependent on the occasional late coalescence in the tree (see updated Supplementary note 3 which is also appended to the end of this rebuttal for ease).

2. Supplementary Note 3 mentions that simulations indicate the expected time between acquisition and the MRCA of a large mutant clade is approximately exponentially distributed with a mean of $1/s$. There is actually mathematical justification for this when assuming a birth-death process which would be nice to include here along with simulations. Namely, the time that a lineage persists until it splits on a Yule tree is exponentially distributed with mean = $1/s$ (e.g., see Mooers et al. Syst Biol 2012).

Thank you for this suggestion. We have included this reference (page 41, highlighted, ref 66) and have modified the text to note that this is a known result.

3. The panel captions for Extended Figure 5 a and b are swapped.

Our apologies for this oversight which we have now corrected on page 23 (highlighted).

4. Extended data 3 are not ultrametric trees as described in the caption title, and panel A is not explained in the caption.

Our apologies for this oversight. We have removed 'ultrametric' from the figure heading and have inserted the missing 3a legend on page 22.

5. Bottom of page 38 "See Figure S1 for the per donor clade specific mutation rates." Figure S1 currently does not appear to include the per donor clade specific mutation rates.

Our apologies for this oversight as it was in an earlier version of the manuscript. We have now removed reference to per donor clades specific mutation rates.

6. To be fixed - page 26 still says '(ref)'

This has now been corrected. Many thanks for spotting this. We have checked and there are no further "(ref)" remaining in the manuscript.

7. There are 2 different Github links currently provided for rtreefit so it is not clear which was used for these results (NickWilliamsSanger and NangaliaLab). Please clarify.

We can confirm that both links are the same and that there is now a notice on the NickWilliamsSanger repository redirecting users to nangaliaLab. For the code to accompany the manuscript, please use the link at NangaliaLab as stated in the Data Availability code.

8. Some repetition of ideas in last and third to last paragraphs of results. In the last paragraph, the results on doubling time are essentially the same as those for growth rates just explained in a different way. Further, the explanations in Supp Note 4 are somewhat confusing and long-winded. After providing Figure S5 with various assumed values for N, the authors show essentially in the log-log analysis that, regardless of the N values, the estimated stem cell number at diagnosis will not be the same across patients. It seems plausible that growth is not logistic with a carrying capacity equal to the total number of stem cells; it appears to saturate below a fraction of 1. Therefore, the idea that the latency is simply a function of growth rate may be incorrect and it is likely a function that also includes carrying capacity. The authors could clarify these observations in Supp Note

We accept that there is some repetition of ideas between the stated paragraphs. We have now shortened the last paragraph of this section of the Results and removed the duplication of the message using doubling times (page 8). We will further shorten the manuscript as required depending on Journal requirements.

With respect to Supplementary note 4, we have simplified and shortened this note. We retain the work on the estimated number of N for different patients (Figure S5), the log-log analysis (Figure S6) that shows that while there is a clear relationship between growth rates and latency, the slope is not what is observed in patients. We then hypothesise the various possibilities to explain this observation as follows (page 53): "In summary, whilst there is a clear correlation between growth rate and latency to diagnosis in the cohort, our modelling suggests that patients do not simply present with disease when they reach a certain number of mutant HSCs following a period of clonal expansion that follows either exponential or logistic growth. Indeed, the latency to diagnosis of initially very rapidly growing clones is greater in patients than expected under our simple model. It is possible that some patients (eg., younger patients) have increased carrying capacity of mutant HSCs, or increased tolerance of disease burden, before presenting with symptoms. Our observation could also be explained by there being a maximum number of leukemic HSCs that are able to be sustained in the bone marrow, thus more dramatically limiting population growth, particularly at extremely high s as observed in some of patients".

9. Notes on patient-specific growth estimates:

- PD57335: The estimates for both ages at sampling should be considered, as there is nothing in the tree to suggest that treatment significantly affects the mutation rate over the given year (I also think there should be 2 dots for this patient in Figure 1A?). This will allow for orthogonal estimates on the same clone with the second timepoint.

Thank you. We have added the missing timepoint to Fig.1A - our apologies for the omission. We have now conducted additional analyses for both timepoints using three methods: birth-death MCMC, cloneRate and *phylofit*. Interestingly, the second time point for PD57335 provides a much higher estimate of “s” than the first time point for all methods, with the results from the second timepoint being most similar to the estimates derived using *phylofit* on the first time point (**Rebuttal Figure 6**).

Rebuttal Figure 6. The estimated growth rates for two PD57335 timepoints. The median and 95% CIs of the growth (“s”) are shown.

We have retained our mention of the growth rates for the first time-point, but both timepoints are included in a new Supplementary Table 3 that includes all the growth rates for *phylofit*, as well as Supplementary Table 4 that includes the comparison of the different methods for the clades.

- PD51633: This clone has 4 sampled cells which the authors show is too small for both methods’ estimates. I suggest using the birth-death MCMC method from the cloneRate package for this sampled number. This small n also exacerbates the issue that confidence intervals for estimates for Phylofit are too narrow as the authors do note.

Thank you for this suggestion. We have now applied the birth-death MCMC method alongside the other methods. The inferred “s” is similar to that estimated using cloneRate::maxLikelihood, but is lower than the headline *phylofit* estimate. It is interesting that the *phylofit* posterior distribution of “s” is so skewed (median is quite close to lower CI) (**Rebuttal Figure 7a**). Investigation of the rstan “pairs” plot for *phylofit* indicates this is because the upper end of prior is on the midpoint bite (**Rebuttal Figure 7b**). Nonetheless, we believe that our choice of prior range for the logistic growth curve mid-point, with the upper bound being given by the age at diagnosis, is consistent with clinical results. Note that increasing the upper bound for the mid-point to 60 years (i.e. 8 years after diagnosis) restores comparability of the growth rate estimate with the other methods ($s_{\text{phylofit}}=0.75$ (0.46-1.60)). Interestingly, this patient presented incidentally with an unusually low white cell count at diagnosis and with subclonal *BCR::ABL1* levels (VAF<0.5) suggesting that symptomatic CML presentation may well have occurred after a longer period of latency.

Rebuttal Figure 7. a. The median and 95% CIs of the growth (“s”) estimates for PD51633. b. The rstan pairs plot of the phylofit inference.

As before, a new Supplementary Table 3 includes all the growth rates for *phylofit*, as well as Supplementary Table 4 that includes the comparison of the different methods for the clades.

- PD57334: Again here, I suggest exploring output using the birth-death MCMC method in cloneRate for n=11 sampled cells, as it is shown in Supp Note 3 that Maximum likelihood method can have a bias for small n sampled cells. Do estimates from the birth-death MCMC method fit closer with the Phylofit method?

Yes, the birth-death MCMC method gives similar results to *phylofit* (**Rebuttal Figure 8**). See, the revised Extended Data 5b (**Rebuttal Figure 3 above, right panel**) showing that generally the two MCMC methods give similar results except for PD57335 and to a lesser extent PD51633.

Rebuttal Figure 8 The estimated growth rates for PD57334. The median and 95% CIs of the growth (“s”) are shown.

The birth-death MCMC model performed better than *phylofit* in terms of the Root mean square error for smaller clades. We have updated Supplementary note 3 to reflect this. The results from MCMC were more in keeping with

estimates derived using *phylofit* rather than the cloneRate model. Therefore, when we model the growth rates using MCMC the results are similar for all patients except for the BCR::ABL1 clade of PD57334 which reaches higher growth rates, PD57335 (depending which time point is used, can be the 4th fastest or the 5th fastest growing clade in our cohort vs 2nd fastest using phylofit) and PD51633 and PD51635 which swap places. **Comparison of growth rates for the different methods is provided in supplementary Table 4.**

- PD57332: There is evidence of imbalanced selection (due to a subclone) that leads to different results with the lengths based method and maxLikelihood method using cloneRate (shown in Fig S2). I suggest that the RUNX1 subclone be measured separately. Measuring BCR::ABL1 clone (n=5, including one coalescence time from RUNX1 ancestral branch) will be noisy, but estimates can be made using Phylofit and birth-death MCMC in cloneRate (suggest this over maxLikelihood or Lengths based methods for n=5). General note of interest to mention: RUNX1 mutation in PD57332 allows the clone to reach full stem cell domination, not present in any other of the clones/patients.

Treating either the *RUNX1* mutation separately (**Rebuttal Figure 9**, middle panel), or by collapsing the *RUNX1* clade to a single representative colony, effectively treating *BCR::ABL1* separately (right hand panel) both give higher central estimates that are more in keeping with the observed clonal sweeps. To avoid overfitting, we have decided to infer all *BCR::ABL1* growth rates using a uniform approach regardless of apparent staggered clonal evolution (e.g in PD57332 or PD51635).

Rebuttal Figure 9 The estimated growth rates for PD57332 treating all branches of the BCR::ABL1 clonal expansion (left), only the RUNX1/BCR::ABL1 branches (middle), and BCR::ABL1 ‘only’ descendent branches (with the RUNX1 mutant clade collapsed to be represented by a single terminal branch) (right). The median and 95% CIs of the growth (“s”) are shown.

Thank you also for the observation that for *RUNX1*, which is a prognostic marker in CML, we observe this mutation sweep to clonal dominance, unlike for other chronic phase patients that have additional mutations within the *BCR::ABL1* clone. Indeed, this pattern of clonal sweeps is also observed in the advanced CML patients. **We include this point regarding dominance of additional driver mutations within the BCR::ABL1 clone in a new results section (commences page 9), as seen in Rebuttal Table 1 and 2 above (now included as Supplementary Table 6). We also include the growth rates for these additional clones in Supplementary Table 3 and Supplementary Table 4.**

- PD51635: From the tree structure for the clone with BCR::ABL1, there appears to be subclonal expansions (but without known drivers) and then Tx drives a refractory clone without those subclonal expansions.

We were intrigued by this observation as well and we have searched for an explanation, however, ultimately, we are not confident of the cause of this staggered pattern of clonal outgrowth. First, we considered whether *BCR::ABL1* was acquired in one of these subclonal expansions, with the earlier branches representing pre-existing clonal

haematopoiesis. However, all colonies in the clade confidently demonstrate *BCR::ABL1* which we have confirmed by both reviewing the variantcalls, but also viewing the individual structural variant data for all the colonies in this clade. Secondly, we considered if there was an additional driver mutation (indel, point mutation, copy number change etc) within a subclonal expansion of this *BCR::ABL1* mutated clade or on the ancestral clade. Again, we cannot find any such events, even with a manual curation and search for such mutations within the genomes of these colonies. The pattern of branching strongly suggests that the growth is not linear in this patient, and therefore, what we infer, at best, is the average growth rate across the period of clonal expansion.

Rebuttal Figure 10: The inferred growth rate for the highlighted sub-clade is more in the range typical of the rest of the cohort. The median and 95% CIs of the growth (“s”) are shown.

To avoid overfitting, we have decided to report all *BCR::ABL1* growth rates using a uniform approach regardless of apparent staggered clonal evolution (eg. in PD57332 or PD51635). However, **we acknowledge this unusual pattern of clonal outgrowth as follows (page 8 para 2):** Of note, the growth rate in PD51635 appeared initially to be slower (as judged by longer inter-coalescent intervals) and then more rapid within a subclone (Fig.2, Supplementary Table 3, 4).

10. It would be nice to also include comparisons using cloneRate and Phylofit for clones other than *BCR::ABL1* such as NF1 embryonic mutation PD51634, RUNX1 subclonal mutation in PD57332, and multiple clones in PD51635. This excellent dataset can also advance knowledge of somatic mosaicism/CHIP in general! Also, making the processed data easily accessible (namely, the mutation and ultrametric trees for each patient) upon publication will benefit the research community and future studies.

Thank you for this valuable recommendation. **We have now added the estimates for the clones other than *BCR::ABL1* using cloneRate, phylofit and MCMC model (Supplementary Table 3 (phylofit estimates) and 4 (comparison between the different methods).** We do not estimate a growth rate for the NF1 embryonic mutation in this study as this is occurring very early in life and therefore, we would need to make assumptions regarding any ongoing growth in the background population of HSCs that might be occurring during this time which would influence our estimates of growth rates. In a previous publication, we attempted to take this into account (Williams et al, Nature 2022⁴) due to the multiple early driver mutations observed - we concluded that very early clonal expansions (eg in *DNMT3A* mutated clones) could hitchhike on background population growth to fix in the population despite low *s*. We express them in terms of *s*, where *s* is much less than 1 then this roughly corresponds to the annualised growth rate *S* e.g. $s=0.1 \Rightarrow S \sim 10\%$. For higher *s*, the corresponding *S* is dramatically higher ($S = \exp(s) - 1$). This allows for comparisons with the estimates of growth rates in clonal haematopoiesis presented in Mitchel et al. 2022 and Fabre et al. 2022.

All data will be made available in line with the Data Sharing Policy at Sanger. Therefore, original BAM files will be available via EGA (<https://ega-archive.org/>). In addition, we had already made available the somatic mutation files and tree structures via the .zip file at initial review but we are uncertain if this made it to the reviewers. These have

now been made live via the github links (<https://github.com/nangalialab/CML>).

Referee #2 (Remarks on code availability):

The link <https://github.com/nangalialab/CML> [github.com] does not seem to work currently.

We are planning on making the link public at the time of publication and had included a .zip file with code, somatic mutations and tree structures for the Reviewers. Our apologies if the Reviewer was unable to access these.

<https://github.com/nangalialab/CML> should now be live.

Referee #3 (Remarks to the Author):

In this manuscript, the evolutionary genomic dynamics of HSC lineages in Philadelphia translocation-positive (PhT+) CML patients have been characterized in nine individuals. The new data generated through WGS of single-cell colonies are valuable for exploring the mutation rate in PhT+ CHPs and the timing of PhT acquisition, but they seem to have lower sequencing depths than some recent studies (<https://doi.org/10.1038/s41586-022-04786-y> and <https://doi.org/10.1038/s41586-021-04312-6>). Anyway, the investigation is biomedically important, as the initiation of CH (leading to CML) by PhT acquisition would suggest that it could serve as a unique driver event, enabling more precise disease prediction and indication of imminent blood cancer.

We thank the Reviewer for their kind comments. With respect to the depth of sequencing, we would like to assure the Reviewer that our depth of whole genome sequencing is comparable to our other similar studies, including the one cited above by Mitchell *et al*, Nature 2022³. In that study, we sequenced to an average depth of 14x coverage, and in our current study we have sequenced to a mean depth of 15.3x coverage. In Williams's *et al*, Nature 2022⁴, where we reconstruct the phylogeny of Ph- myeloproliferative neoplasms, we also sequenced to a similar depth of 16.7x. In all these studies, the wildtype mutation accumulation rates are not statistically significantly different, and are in line with mutation rate reports from other labs (Osario *et al*¹⁴.) and orthogonal techniques (Abascal *et al*.¹⁵) (**Rebuttal Table 3**). In addition, mutations are assigned to the tree in a sequencing depth-sensitive manner using *treemut* (<https://github.com/nangalialab/treemut>). Branch lengths are adjusted for the branch specific SNV detection sensitivity⁷, where the sensitivity of detection of fully clonal SNV variants is directly estimated from the per colony sensitivity for detecting germline heterozygous SNVs together with a multiplicative correction for the clonality (VAF) of the colonies.

Rebuttal Table 3: Estimates of mutation rate over life in haematopoiesis from different cohorts/studies

Study cohort	Reference	Method	Mean sequence depth	Wildtype mutation rate SNV/yr (95% CI)
CML	This study	Colony WGS	15.3x	18 (16.1-20.2)
HSPC	Mitchell et al ³	Colony WGS	14x	16.8 (16.5-17.2)
Ph-ve MPN	Williams et al ⁴	Colony WGS	16.7x	17.0 (14.8-19.2)
HSPC	Osorio et al ¹⁴	Colony WGS	30x	14.2 (6.1-22.4)
Granulocyte	Abascal et al ¹⁵	Nanoseq		19.9 (18.3-21.4)

HSPC, haematopoietic stem and progenitor cells

The manuscript claims that the PhT acquisition, mapped on the shared branch of large clonal hematopoiesis (CH) in every patient, occurred at or near its most recent common ancestor (MRCA) or coalescence. Given the considerable length of these shared branches in the CHs, with numerous mutations also mapping to them, the authors are faced with the challenge of demonstrating that PhT acquisition was the critical event precipitating CH formation. This necessitates an ordering of polymorphisms along a lengthy shared branch—a task rendered impossible by the statistically indistinguishable frequencies of these variants. The current manuscript does not solve this problem. It has not convincingly mapped PhT acquisition onto MRCA, meaning that MRCA's timing only places the latest time bound on PhT acquisition rather than providing information on PhT's actual acquisition timing.

We thank the reviewer for this important point. We wished to be quite careful in the manuscript in stating that we are timing the MRCA of the CML clone and the commencement of clonal expansion. For example, on page 5, where we first discuss the patterns of branching in the phylogenies we state, "These data suggest that the MRCA and commencement of tumour clonal expansion in CML patients are recent events. This contrasts with patterns observed for other driver mutations in both malignant and healthy haematopoiesis from previous studies^{5-7,9}."

The rapid clonal outgrowth of *BCR::ABL1* positive clonal expansion and trajectory to CML, is a key finding of this paper. As the Reviewer states, for any branch on the phylogenetic tree, we cannot infer the relative timing of mutations within that shared branch. We agree that at times we have inadvertently referred to the end of the branch (the MCRA) as the time of *BCR::ABL1* acquisition - we have changed such references to ensure that we are timing the start of the *BCR::ABL1* clonal expansion and not strictly speaking *BCR::ABL1* acquisition. For example (page 8): "Acknowledging the small number of patients in this study, there appeared a plausible trend of younger age of onset correlating with more explosive growth and a shorter duration between the beginning of *BCR::ABL1* clonal expansion and clinical diagnosis (Fig 4a,b)", and on page 10: "Our inferred estimates of the duration between the start of *BCR::ABL1* clonal expansion and disease presentation fall within the ranges previously reported in radiobiological studies of cancer incidences in Japanese survivors of the atomic bombs...".

Nevertheless, we believe it is reasonable to suggest that *BCR::ABL1* might be occurring at the end of the shared branch for the following reasons:

1. CML is pathognomically defined by the *BCR::ABL1* fusion and *BCR::ABL1* is sufficient to induce a CML like phenotype in mice, suggesting that the acquisition of this mutation triggers transformation.^{16,17}
2. In line with other recent genomic studies in chronic phase CML^{11,12}, no other genetic driver mutation is found on any of the shared branches across the 9 phylogenies. Indeed, dN/dS only identifies *DNMT3A* mutations (in wildtype colonies) as an additional recurrent event reaching significance.
3. The incidence of CML within the population is in line with a one-hit hypothesis¹⁸, unlike other multi-hit cancers such as ovarian cancer and colorectal cancers, that have an exponentially increasing incidence with age in keeping with the time taken to acquire multiple hits.
4. Whilst it is possible that *BCR::ABL1* is acquired earlier in the branch, this would then additionally require that (i) the HSC in which *BCR::ABL1* occurs continues to behave entirely as a wild-type HSC both in terms of its mutation acquisition patterns and cell division rates, despite the presence of *BCR::ABL1*, and (ii) a second non-genetic event then triggers clonal expansion in every patient. Citing Occam's razor, this would not be the most parsimonious explanation for what triggered the clonal expansion.
5. *BCR::ABL1* is not reported in the healthy population as stable CHIP clones (as defined by VAF>2%). For a two-hit hypothesis (*BCR::ABL1* acquisition plus an additional non-genetic event), one would expect to see *BCR::ABL1* clonal expansions in healthy individuals. This has not been reported within haematopoietic stem cells. Of note, while there are reports of *BCR::ABL1* RNA transcripts at incredibly low levels within blood¹⁹⁻²², these have not been shown to be in self-renewing cells, or been shown to be maintained over time. In keeping with this, a positive finding of *BCR::ABL1* in the clinic using routine clinical assays is currently diagnostic of CML. In contrast, mutations in many other drivers

(*TET2*, *DNMT3A* etc) are not alone diagnostic for a clinical disease, as they commonly cause asymptomatic clonal haematopoiesis with normal blood counts.

6. In cancer studies, a clonal expansion observed in association with a recurrently mutated driver mutation is assumed to have been caused by that driver mutation, due to the positive selection that these driver mutations confer. In Watson *et al*²³, clonal expansions are identified by the VAF of driver mutations. The rate of growth of the clone is inferred to be the fitness advantage induced by those driver mutations. Similarly, in phylogenetic studies such as Mitchell *et al*³, Williams *et al*⁴, Machado *et al*², etc, clades found to expand downstream of a recurrently mutated driver mutation present on a shared branch, and in the absence of another coexisting driver mutation, are rightfully identifying the fitness of those driver mutations. In these phylogenetic studies, the driver mutation is assumed to be occurring at the end of the shared branch, triggering the clonal expansion. We have made the same reasonable assumptions in this study.

However, the Reviewer is right that we cannot time *BCR::ABL1* to the end of the shared branch from our phylogenies specifically. Therefore, as mentioned above, we have referred to the commencement of “*BCR::ABL1* clonal expansion” rather than “*BCR::ABL1* acquisition”. What we wish to focus on is that the rapid clonal expansion results shown for *BCR::ABL1* positive clades in this study which are starkly different to those observed in association with other drivers in the studies described above. Whilst *BCR::ABL1* may not always have the potential to do this in all individuals, we describe here is the pattern of clonal outgrowth observed for *BCR::ABL1* in CML patients.

Additional considerations cause me to be skeptical.

Selecting CML patients with PhT for WGS sequencing means the universal presence of PhT across the samples does not substantiate the claim that its acquisition directly led to the CH event. Such a conclusion would require further evidence even with an unbiased sample of PhT+ and PhT-CMLs. For instance, should PhT emerge as the sole driver in PhT+ HSC lineages, it could be argued as the primary cause of CH initiation, but still, the longitudinal sampling of patients before they develop would be needed to be sure.

Thank you for this interesting point. Clinically, there is no official diagnosis of PhT-negative CML. A finding of *BCR::ABL1* or PhT (or equivalent translocation resulting in activation of *ABL*, aka PhT) detected at the DNA level by G-banding/karyotype or Fluorescent in-situ hybridization (FISH) is pathognomonic for CML. Historically, some CML cases were taxonomically referred to as *BCR::ABL1* negative, however, such cases are now considered entirely separate disease entities, and most recently some of these cases have been reclassified further as myelodysplastic/myeloproliferative neoplasms [MDS/MPN] with neutrophilia²⁴.

The Reviewer asks if the *BCR::ABL1* emerges as the sole driver in *BCR::ABL1*+ HSC lineages. This is what we have shown in our study, which is in line with other recent genomic studies in chronic phase CML^{11,12}. Our phylogenetic trees show the historical HSC divisions (self-renewing symmetrical HSC divisions) right back to the start of the *BCR::ABL1*+ HSC clonal expansion and we show that the only recurrent genomic driver is *BCR::ABL1* in all 9 individuals. We do show that some individuals have additional driver mutations (e.g., an *in utero NF1* frameshift in PD51634 and a *RUNX1* frameshift mutation in PD57332), however, *BCR::ABL1* is the only common event across all 9 chronic phase patients and the 4 blast crisis/advanced phase CML patients.

If we understand the Reviewer correctly, they may be raising the possibility that *BCR::ABL1* can cause a benign clonal haematopoiesis in the general population, and that by only studying CML patients, we would not observe this alternative downstream consequence of *BCR::ABL1*? In other words, if we only focus on patients with *BCR::ABL1* who are diagnosed with CML, it does not necessarily substantiate the claim that the acquisition of *BCR::ABL1* directly leads to the onset of CML. However, as already discussed above, we wish to highlight several lines of data that suggest *BCR::ABL1* is sufficient to cause CML. To our knowledge, recurrently positive karyotype or FISH for *BCR::ABL1* is not found in healthy individuals.

Furthermore, we have searched for published studies of CHIP (defined as VAF>2% of a recurrently mutated driver gene in an individual without overt haematological blood count abnormalities), and cannot find data to support *BCR::ABL1* is a CH marker. However, it appears that most CH studies have not specifically looked at chromosomal translocations, instead focusing on point mutations, insertions/deletions or chromosomal copy number changes^{23,25-28}.

So to better address this question in a relatively unbiased way, we gained access to a USA based population-based cohort (the “All-of-Us” Research Program) where electronic health records (EHR) data are linked to whole-genome sequencing (WGS, average sequencing depth $\geq 30\times$) data for >200K individuals. Specifically, we aimed to characterise individual carriers of *BCR::ABL1* and their disease profiles to determine if the acquisition of *BCR::ABL1* leads to a CML diagnosis or if *BCR::ABL1* is found in the absence of CML, akin to other drivers of clonal haematopoiesis found in the healthy ageing population.

We searched each sample’s whole genome sequencing (WGS) CRAM file (GRCh38 mapped; average genome-wide coverage $\geq 30\times$) for sequencing read pairs that support a canonical *BCR::ABL1* (or *ABL1::BCR* reciprocal) variant between *BCR* (chr22:23289313-23292813) and *ABL1* (chr9:130674613-130874613). We retained read pairs with correct orientation and good mapping quality (MAPQ score >6). These reads were then annotated using GRASS (Gene Rearrangement AnalySiS, <https://github.com/cancerit/grass>), which takes pairs of genomic coordinates describing potential rearrangement events and predicts fusion consequences. We required the fusion event to be between *BCR* and *ABL1* genes.

After screening 206,173 whole-genome sequenced participants who have EHR data linked (from 245,388 total WGS participants), we only identified 39 individuals (0.019%) who harboured two or more supporting reads of *BCR::ABL1*, consistent with definite presence of PhT. Interestingly, the prevalence of CML in the USA is estimated to be 0.02% (<https://seer.cancer.gov/statfacts/html/cm1l.html>), very similar to the prevalence of *BCR::ABL1* positivity within the All-of-Us cohort, suggesting that there is not a further cohort of individuals with positive *BCR::ABL1* in the absence of CML.

On further analysis, we also identified individuals that only had a single read of *BCR::ABL1* or *ABL1::BCR*. Whilst the confidence around finding a single read is lower, we found 40 such “singleton” individuals in the whole cohort (0.019%, 11 with single *BCR::ABL1* and 29 with single *ABL1::BCR*). We included them in our subsequent analysis.

To identify what proportion of *BCR::ABL1* carriers had known or suspected CML, we extracted their electronic health record data and searched for International Classification of Diseases (ICD) codes related to cancer, blood, or immune diseases. We searched for a mention of CML from the ICD code, but also included codes mentioning ‘abnormal white cell count’, ‘basophilia’, ‘splenomegaly’, ‘unspecified cancer’ or ‘anaemia’.

The majority of individuals with 2 or more reads had either a diagnosis of CML, or other haematological diagnoses suspicious for CML (e.g., splenomegaly or basophilia) or unspecified haematological issue (**Rebuttal Figure 11**). Only 14 individuals out of the 206,173 analysed cohort had 2 or more reads of *BCR::ABL1* and did not have CML relevant status reported, and of these, 3 had sampling after the date of their most recent ICD record, and 3 had no clinical information updates from 1 year after sampling (**Rebuttal Figure 12**). We noted that among all carriers with 2 or more reads, those with a follow-up time of over 2.5 years all have either a reported haematological issue (e.g., anemia) or a diagnosis of CML or pre-CML conditions (e.g., leukocytosis). These data suggest that a readily detectable *BCR::ABL1* clone in the healthy population in the absence of a diagnosis of CML is genuinely rare. When found, a diagnosis of CML is highly likely to follow - and this is in keeping with clinician experience.

Rebuttal Figure 11. CML diagnosis increases with supporting read count.

Rebuttal Figure 112. Each bar in the plot represents a PhT carrier identified in our analysis, with the bar colour indicating the number of reads supporting their carrier status. Individuals diagnosed with CML or a pre-CML state during the follow-up period are marked with an asterisk. The Y-axis shows follow-up time, transformed using an arcsinh transformation (which approximates the natural logarithm while retaining zero-valued observations) for visualisation purposes. We have received an exception to the Data and Statistics Dissemination Policy from the All of Us Resource Access Board to display this data.

Just over half of individuals with a single read of *BCR::ABL1* or *ABL1::BCR* did not have a diagnosis or features suspicious of CML. We wondered if this group of individuals included some genuine CML patients (e.g., those on treatment, or pre-diagnosis), but also those in whom the detection of *BCR::ABL1* was either an artefact or events that occur in mature blood cell populations and not the stem cell compartment.

To address the first possibility - whether these occasional single reads were sequencing artefacts - we screened ~10,000 whole genomes derived from single-cell derived haematopoietic colonies previously sequenced at the Sanger institute^{2-5,29} for the presence of *BCR::ABL1* reads. In such colonies, mutations are present across all cells

within the colony, due to the clonal nature of the sample, and thus bonafide mutations would have a variant allele frequency (VAF) of 50%. Furthermore, colonies are derived from haematopoietic stem and progenitor cells and not mature cells. WGS coverage was 10-15x and no donors were known to have CML. Across ~10,000 colonies, we only identified three reads from three colonies (from 3 separate individuals), which were translocated between our BCR and ABL1 target regions described above. However, these reads were not in the correction orientation to support a BCR::ABL1 or ABL1::BCR fusion. Thus, none of the local samples were identified as containing any productive BCR::ABL1 reads. This suggests that the BCR::ABL1 single reads in the All-of-Us dataset are less likely to be an artefact of sequencing, as otherwise we would have expected to see some positive single reads in the 10,000 WGS dataset from blood progenitors. Of note, all 10,000 local whole genomes were generated from *in vitro* culture of stem and progenitor haematopoietic cells, whereas All-of-Us sequenced whole blood which includes mature cell fractions.

Therefore, it is possible that such ‘singleton’ cases in the “All of us” cohort represent mutated, differentiated blood cells that cannot maintain clones over time. If this was the case, we would not expect such clones to increase in frequency with age, unlike clones within the stem cell compartment that would accrue over time. To explore this hypothesis, we examined the age distributions of those individuals with single-read BCR::ABL1/ABL1::BCR (n=40), those with 2 or more reads (n=39), and compared these to CML incidence in the remaining wild-type population. We first calculated the CML incidence rate (per 100,000) from 2018-2022, during which the currently available data was actively collected. This was done across the entire cohort, regardless of WGS data availability (n=280K), as well as within the WGS subset (n=200K). We then calculated similar statistics for PhT carrier status, stratified by singletons and other carriers. The age groups shown are age at diagnosis for CML incidence and age at sampling for PhT carrier status.

Indeed, we found that ‘singleton’ carriers displayed a more uniform incidence across age groups, while carriers with 2 or more reads, or patients with CML showed increased incidence with age (**Rebuttal Figure 13**). This would be the expected pattern if BCR::ABL1 is being acquired in non-persistent clones at a constant rate during life.

Rebuttal Figure 13 CML incidence and PhT carrier status across age groups. In each panel, the bars represent the average number of new CML cases or PhT carriers per year, aligned with the left y-axis. The dots indicate the incidence rate per 100,000 population, corresponding to the right y-axis. For each age group, summary statistics were calculated by averaging annual statistics from 2018 to 2022, under the assumption that the population structure remained relatively stable during this period.

Overall, our analyses suggest that *BCR:ABL1* does not commonly exist as stable clonal haematopoiesis without CML, or impending CML, in the healthy population. This provides further support for the one-hit CML model confirmed by our phylogenetic analysis.

The finding that PhT+ CHs in CML patients mutate at an accelerated rate fails to associate PhT with the MRCA node specifically. Such a rate change is previously documented, with both healthy (in older age) and diseased (across all ages) individuals exhibiting more CHs and a faster accumulation of mutations (<https://doi.org/10.1093/molbev/msad279> [doi.org]). Therefore, faster mutation rates are linked with the presence of CHs in general, not uniquely with PhT+ CHs. Anyway, such rate changes cannot preferentially place PhT onto the MRCA node, as there is no way to exclude many other variants mapped to long shared branches.

We thank the reviewer for their comment. We would like to clarify that the increase in rate of somatic mutation acquisition that we describe during *BCR::ABL1* associated clonal expansion is magnitudes higher than any previous reports in CH and other MPNs. Indeed, in CH clones^{3,4} and Ph- MPNs⁴, the increase in somatic mutation burden within the clonal expansion is barely significant and reflects the very modest increase in HSC cell division rates (**Rebuttal Figure 14**). This results in slow clonal expansions over decades reflecting the mild to moderate fitness of these driver mutations, confirmed by published studies using both phylogenetic trees^{3,4,29} and evolutionary framework based approaches²³. Of note, the majority of mutations accrue in HSCs due to the passage of time, and only 1-2 of the 17-18 mutations accrued per year are thought to result from errors of cell replication. Thus, with mild increases in cell division rates as observed in Ph-MPN or CH, the overall mutation burden only increases slightly^{3,4}. The marked increase in mutation burden in CML, predominantly due to increases in SBS1 and also contributed to by SBS18, are uniquely reflecting the more rapid cell division and greater S of this tumour. Such dramatic differences in S of the clonal expansions between CH, Ph-MPN and CML, are also clearly visible from the different coalescent patterns of the respective phylogenetic trees, with longer inter-coalescent intervals during clonal expansion in CH and Ph-MPN. These shorten in Ph-MPN only upon the acquisition of additional driver mutations, reflecting more rapid clonal expansion⁴.

To illustrate the point above, we present **Rebuttal Figure 15** describing the S for *JAK2*-mutated MPN from Williams *et al*, Nature 2022⁴ alongside that for the *BCR::ABL1* mutated clades from this study. We have set a reference line at $S = 1$ as this is equivalent to 100% additional growth per year. This plot clearly shows that the majority of *JAK2*-mutated clades have $S < 1$ with a median of 0.52. Conversely the majority *BCR::ABL1* mutated clades have $S > 1$ with a median of 110 - bear in mind this is an over 100-fold increase in fitness over the *JAK2*-mutated clades.

Rebuttal Figure 14. Plot showing S , the median annual growth rate per year, for a variety of *JAK2*-mutated clones from Williams *et al*, Nature 2022, and for CML clones in this study. X axis shows the donor ID from this study and Williams *et al*. 9pUPD, chromosome 9 uniparental disomy; *JAK2:DNMT3A*, represents clonal expansion with both driver mutations upstream of the clonal expansion. The dots are coloured by the study, and the error bars reflect the 95% CI around the median S .

We now add further clones from Mitchell *et al*, Nature 2022³, to further contrast the growth rate of the *BCR::ABL1* mutated clade (median $S = 110$) and demonstrate it to be magnitudes higher compared to other CH clades in healthy individuals (median $S = 0.19$) below (**Rebuttal Figure 15**).

Rebuttal Figure 15. Plot showing S , the median annual growth rate per year, for a variety of clones from MPN patients from Williams *et al*, Nature 2022, for CH clones from healthy individuals from Mitchell *et al*, Nature 2022, and for CML clones in this study. X axis shows the donor and clone ID from these studies. The dots are coloured by the study, and the error bars reflect the 95% CI around the median S . Note the two clones to the far left of the plot that have very low growth rates (PD5847_DNMT3A, PD5182_DNMT3A) - these two clones were acquired in utero.

While we cannot specifically assign *BCR::ABL1* acquisition temporally within the long branch, we can say that the rapid expansion occurs on the background of *BCR::ABL1* positivity and in the absence of any other recurrent known CH/MPN coding drivers. The most parsimonious explanation is that it is *BCR::ABL1* that triggered this growth at the end of the shared trunk. However, as mentioned above, **we now refer only to the timing of the *BCR::ABL1* clonal expansion, and not the *BCR::ABL1* acquisition.**

While the patterns of telomere decay observed during CH-HSC diversification are reasonable, they do not aid in precisely mapping PhT to the MRCA node in the HSC phylogeny.

We agree that we cannot map *BCR::ABL1* along the long branch. The data on telomere attrition in *BCR::ABL1* colonies is not intended to map *BCR::ABL1* to the long ancestral branch. We analysed telomeres and identified marked attrition. This was merely an independent indicator of rapid expansion of the *BCR::ABL1* positive clone, alongside other lines of evidence pointing to rapid growth such as increased SBS1/18 within the branches of the clonal expansion, and the pattern of branching of the clonal expansion that was used to infer growth rates and estimates of fitness of *BCR::ABL1*.

Similarly, the identification of unique mutational signatures (SBS18) in expanding CML clades, although potentially diagnostic of PhT+ CML, does not preferentially map PhT to the MRCA, as compared to other mutations. It's noteworthy that a common SBS9 signature is reported in older age MPN patients in another study (<https://doi.org/10.1093/molbev/msad279> [doi.org]), suggesting that disease-related CHs can exhibit new signatures and such findings are not exclusive to PhT+ CML.

We are uncertain regarding the data mentioned on the presence of SBS9 in older age MPN patients. To our knowledge, this has not been described before. In Williams *et al*, Nature 2022 and Van Egeren *et al* Cell Stem Cell 2022, both of which analyse mutational signatures in MPN single cell derived colonies, no SBS9 is observed. SBS9 is thought to reflect the mutational signature of somatic hypermutation observed in lymphoid cells (<https://cancer.sanger.ac.uk/signatures/sbs/sbs9/>) and this is in line with our experience. The reference mentioned by the Reviewer above (Craig *et al*, Molecular Biology and Evolution 2024), states, “However, we found SBS9 mutational signatures in CHIP lineages of elderly MPN patients only (ages > 80 yr), which showed limited discrimination when using phyloAges.” We have struggled to find the data within that study to support this statement to assess this further. We would not suggest that one uses SBS9 to discriminate between MPN versus healthy individuals as suggested by this study. We currently use national and international clinical guidelines that have standardised the identification and diagnosis of MPNs - using driver mutation presence, abnormal blood counts, thrombotic history and bone marrow morphology^{24,30,31}.

Anyway, the same three mutational signatures are seen across every lineage in all patients’ phylogenies (the aging markers SBSblood and SBS1, and the oxidative damage marker SBS18). The higher proportion of SBS18 observed in CML branches may simply be due to their recent coalescence, meaning that the same number of mutations would have been acquired under this regime in PhT- and PhT+ lineages, but they simply represent a smaller proportion of the total mutational burden in the longer, PhT+ lineages.

We thank the reviewer for raising the possibility that SBS18 could simply be a recent event that occurs equally in all colonies, but appears as a greater proportion in the CML clade due to their shorter branches. We have now investigated this further. There are two reasons why we feel this is not the case. (1) We have now calculated the total absolute numbers of mutations assigned to each signature for individual *BCR::ABL1* and WT colonies. We still find that there is elevated SBS18 in *BCR::ABL1* mutant colonies (**Rebuttal Figure 16**). Paired tests for median numbers of SBS18 vs mutant status at a donor level is not significant (although $p < 0.1$), but in this analysis, we have included pre-CML branches in *BCR::ABL1* positive colonies (which have a WT pattern of mutation acquisition, **Rebuttal Figure 17**, Figure 3b, c, d, e). The trend is stronger and significant when we restrict to the CML clade itself wherein the increased SBS18 occurs (Figure 3b, c, d, e). (2) We also observe increased SBS18 in Early-CML branches (**Rebuttal Figure 17 showing** Figure 3b, c) suggesting these SBS18 induced mutations are not simply recent events.

We absolutely agree that there are no novel mutational processes that are specifically induced by *BCR::ABL1*, despite previous claims that *BCR::ABL1* is mutagenic³², and therefore, we believe our results are an important finding in this regard. We suggest that the increased mutations that accumulate are the result of increased cell division reflecting the rapid expansion of the *BCR::ABL1*+ve clade. In this context, SBS1 increases would be expected. However, it is possible that SBS18 also accumulates in the context of rapid cell division, or during oxidative stress, as increased SBS18 has also been observed in a variety of settings where there is rapid clonal expansion, early in life haematopoiesis⁵, mouse haematopoiesis, especially early in life³³, placenta cells³⁴, colonic crypts³⁵.

Rebuttal Figure 16 Mutational signature contributions of *BCR::ABL1* +ve versus wildtype colonies. Mutations attributed to *BCR::ABL1* colonies are summed along all branches in the lineage of each colony. Y-axis shows the number of mutations per signature.

Rebuttal Figure 17 (Fig.3b,c) Left b. A phylogenetic tree for an example patient PD51632 with the proportion of single base substitution (SBS) signature SBS1, SBSblood and SBS18 contributing to the SNV spectrum annotated on each of the branches. The branches are categorised into five groups as shown on the phylogenetic tree: (i) branches of the *BCR::ABL1*-positive clonal expansion (“CML”, depicted on the tree), (ii) the ancestral branch of *BCR::ABL1*-colonies representing the lineage of origin of the CML (“pre-CML lineage”), (iii) the earliest mutations within the CML (ie., the shared branches within the clonal expansion, “early CML” depicted by the lower shaded purple box), (iv) the branches of *BCR::ABL1*-negative colonies (wild-type, WT, depicted by arrows) and (v) early in life mutations representing the top 100 mutations of the phylogenetic tree, depicted by the upper shaded blue

box . **Right c.** Bar plot represents the proportion of SBS1, SBSBlood and SBS18 contributing to SNV spectra of the example case PD51632 for the different phylogenetic tree categories shown in (b).

Ultimately, the detection of a PhT event in an individual has not been shown to be THE event causing CML CH, so the current results cannot claim to establish “the oncogenic potency of a single BCR::ABL1 fusion and its capacity to drive fast clonal outgrowth to cancer” without further evidence, such as longitudinal sampling of individuals who ultimately develop CML.

It has long been accepted that PhT is pathognomonic for and causes CML. There are of course likely to be other factors, such as the cell of origin (primitive versus more mature HSC), etc. However, in terms of genetic drivers of CML, BCR::ABL1 is accepted to be the single-hit required. This is also evident in the unique response of the tumour to tyrosine kinase inhibition, and the resistance mechanism to therapy that emerges via acquisition of ABL1 kinase domain mutations on the BCR::ABL1 allele. In this study, BCR::ABL1 is the only recurrent genomic event associated with the observed rapid clonal expansion. Whilst it is possible that in every single patient, a second unobserved non-genomic event was acquired in addition to the initial BCR::ABL1 driver, this would not be the most parsimonious explanation especially when the progression from CP CML to AP and BP is marked by additional genomic mutations. Indeed, in other studies, a clonal expansion observed in association with a driver mutation is understandably assumed to be driven by that driver mutation. In Watson *et al*³, clonal expansions are identified by the VAF of driver mutations. The rate of growth of the clone is inferred to be the fitness induced by those driver mutations. Similarly, in Mitchell *et al*³, Williams *et al*⁴, Machado *et al*², etc, clones found to expand downstream of a driver mutation and in the absence of another coexisting driver mutation are rightfully identifying the fitness of those driver mutations.

The rapid clonal expansion results shown for BCR::ABL1+ve clades in this study are starkly different to those observed in association with other drivers (for CH and MPN) in the studies described above, and thus we do believe it is fair to highlight the potency of a single BCR::ABL1. Whilst a BCR::ABL1+ve clone may not always have the potential to do this in all individuals, we describe here the pattern of clonal outgrowth observed here for CML patients. Coupling this data with the All-of-us analysis showing low prevalence of detectable BCR::ABL1 in the population in the absence of CML, lends further support to the notion that BCR::ABL1 has significant oncogenic potency in comparison to other driver mutations. We believe that most clinicians would agree with this conclusion, and that our study provides evidence for the single-hit origin of this disease.

Referee #3 (Remarks on code availability):

The reproducibility of results and the adequacy of computational scripts provided could not be assessed because the somatic HSC variation data were not provided. In the past, authors of Nature manuscripts have provided such information freely for somatic variation datasets (as it is not germline).

We apologise that the Referee did not have these data available at the time of review. We had made an example phylogenetic tree and provided somatic mutations, together with code as a .zip file. Upon publication (and before upon request), all original BAM files will be available via EGA (<https://ega-archive.org/>). Code and mutation files including tree topology have now been made available via the github links (<https://github.com/nangalialab/CML>).

1. Radivoyevitch, T., Hlatky, L., Landaw, J. & Sachs, R. K. Quantitative modeling of chronic myeloid leukemia: insights from radiobiology. *Blood* **119**, 4363–4371 (2012).
2. Machado, H. E. *et al.* Diverse mutational landscapes in human lymphocytes. *Nature* **608**, 724–732 (2022).
3. Mitchell, E. *et al.* Clonal dynamics of haematopoiesis across the human lifespan. *Nature* **606**, 343–350 (2022).
4. Williams, N. *et al.* Life histories of myeloproliferative neoplasms inferred from phylogenies. *Nature* **602**, 162–168 (2022).
5. Spencer Chapman, M. *et al.* Lineage tracing of human development through somatic mutations. *Nature* **595**, 85–90 (2021).

6. Machado, H. E. *et al.* Convergent somatic evolution commences in utero in a germline ribosomopathy. *Nat. Commun.* **14**, 5092 (2023).
7. Salichos, L., Meyerson, W., Warrell, J. & Gerstein, M. Estimating growth patterns and driver effects in tumor evolution from individual samples. *Nat. Commun.* **11**, 732 (2020).
8. Lee, N. D. & Bozic, I. Inferring parameters of cancer evolution in chronic lymphocytic leukemia. *PLOS Comput. Biol.* **18**, e1010677 (2022).
9. Keller, G. *et al.* Telomeres and telomerase in chronic myeloid leukaemia: impact for pathogenesis, disease progression and targeted therapy. *Hematol. Oncol.* **27**, 123–129 (2009).
10. Wenn, K. *et al.* Telomere length at diagnosis of chronic phase chronic myeloid leukemia (CML-CP) identifies a subgroup with favourable prognostic parameters and molecular response according to the ELN criteria after 12 months of treatment with nilotinib. *Leukemia* **29**, 2402–2404 (2015).
11. Branford, S. *et al.* Integrative genomic analysis reveals cancer-associated mutations at diagnosis of CML in patients with high-risk disease. *Blood* **132**, 948–961 (2018).
12. Ochi, Y. *et al.* Clonal evolution and clinical implications of genetic abnormalities in blastic transformation of chronic myeloid leukaemia. *Nat. Commun.* **12**, 2833 (2021).
13. Roche-Lestienne, C. *et al.* RUNX1 DNA-binding mutations and RUNX1-PRDM16 cryptic fusions in BCR-ABL+ leukemias are frequently associated with secondary trisomy 21 and may contribute to clonal evolution and imatinib resistance. *Blood* **111**, 3735–41 (2008).
14. Osorio, F. G. *et al.* Somatic Mutations Reveal Lineage Relationships and Age-Related Mutagenesis in Human Hematopoiesis. *Cell Rep.* **25**, 2308–2316.e4 (2018).
15. Abascal, F. *et al.* Somatic mutation landscapes at single-molecule resolution. *Nature* 1–6 (2021) doi:10.1038/s41586-021-03477-4.
16. Cross, N. C. P. *et al.* European LeukemiaNet laboratory recommendations for the diagnosis and management of chronic myeloid leukemia. *Leukemia* **37**, 2150–2167 (2023).
17. Clarke, C. J. & Holyoake, T. L. Preclinical approaches in chronic myeloid leukemia: from cells to systems. *Exp. Hematol.* **47**, 13–23 (2017).
18. Michor, F., Iwasa, Y. & Nowak, M. A. The age incidence of chronic myeloid leukemia can be explained by a one-mutation model. *Proc. Natl. Acad. Sci.* **103**, 14931–14934 (2006).
19. Bose, S., Deininger, M., Gora-Tybor, J., Goldman, J. M. & Melo, J. V. The Presence of Typical and Atypical BCR-ABL Fusion Genes in Leukocytes of Normal Individuals: Biologic Significance and Implications for the Assessment of Minimal Residual Disease. *Blood* **92**, 3362–3367 (1998).
20. Bäscke, J., Griesinger, F., Trümper, L. & Brittinger, G. Leukemia- and lymphoma-associated genetic aberrations in healthy individuals. *Ann. Hematol.* **81**, 64–75 (2002).
21. Biernaux, C., Loos, M., Sels, A., Huez, G. & Stryckmans, P. Detection of major bcr-abl gene expression at a very low level in blood cells of some healthy individuals. *Blood* **86**, 3118–3122 (1995).
22. Song, J., Mercer, D., Hu, X., Liu, H. & Li, M. M. Common Leukemia- and Lymphoma-Associated Genetic Aberrations in Healthy Individuals. *J. Mol. Diagn. JMD* **13**, 213 (2011).
23. Watson, C. J. *et al.* The evolutionary dynamics and fitness landscape of clonal hematopoiesis. *Science* **367**, 1449–1454 (2020).
24. Khoury, J. D. *et al.* The 5th edition of the World Health Organization Classification of Haematolymphoid Tumours: Myeloid and Histiocytic/Dendritic Neoplasms. *Leukemia* **36**, 1703–1719 (2022).
25. Jaiswal, S. *et al.* Age-related clonal hematopoiesis associated with adverse outcomes. *N. Engl. J. Med.* **371**, 2488–2498 (2014).
26. Xie, M. *et al.* Age-related mutations associated with clonal hematopoietic expansion and malignancies. *Nat. Med.* **20**, 1472–1478 (2014).
27. Stacey, S. N. *et al.* Genetics and epidemiology of mutational barcode-defined clonal hematopoiesis. *Nat. Genet.* 1–11 (2023) doi:10.1038/s41588-023-01555-z.
28. Bernstein, N. *et al.* Analysis of somatic mutations in whole blood from 200,618 individuals identifies pervasive positive selection and novel drivers of clonal hematopoiesis. *Nat. Genet.* **56**, 1147–1155 (2024).
29. Fabre, M. A. *et al.* The longitudinal dynamics and natural history of clonal haematopoiesis. *Nature* **606**, 335–342 (2022).
30. McMullin, M. F. *et al.* A guideline for the diagnosis and management of polycythaemia vera. A British Society for Haematology Guideline. *Br. J. Haematol.* **184**, 176–191 (2019).
31. Harrison, C. N. *et al.* Modification of British Committee for Standards in Haematology diagnostic criteria for essential thrombocythaemia. *Br. J. Haematol.* **167**, 421–423 (2014).
32. Slupianek, A. *et al.* BCR-ABL1 kinase inhibits uracil DNA glycosylase UNG2 to enhance oxidative DNA damage and stimulate genomic instability. *Leukemia* **27**, 629–634 (2013).

33. Kapadia, C. D. *et al.* Clonal dynamics and somatic evolution of haematopoiesis in mouse. *BioRxiv Prepr. Serv. Biol.* 2024.09.17.613129 (2024) doi:10.1101/2024.09.17.613129.
34. Coorens, T. H. H. *et al.* Inherent mosaicism and extensive mutation of human placentas. *Nature* **592**, 80–85 (2021).
35. Cagan, A. *et al.* *Somatic Mutation Rates Scale with Lifespan across Mammals*. 2021.08.19.456982 <https://www.biorxiv.org/content/10.1101/2021.08.19.456982v1> (2021) doi:10.1101/2021.08.19.456982.

SUPPLEMENTARY NOTE 3: Estimating *BCR::ABL1* growth rates

In the absence of longitudinal measurements of the *BCR::ABL1* clone VAF prior to diagnosis, we estimated the rate of growth of *BCR::ABL1* clones using the single cell-derived colony based phylogenetic trees and the pattern of coalescences of the *BCR::ABL1* mutant clade. To minimise the use of phylogenetic information that may be affected by therapy we subset the trees to use only those single cell derived colonies that were grown from cells extracted at the earliest diagnostic or post-diagnostic timepoint. We have 3 donors where the earliest sampling time point is after diagnosis. One of these, PD57333, has no detectable clone. The other two are PD57334 and PD57335. To infer the growth rates we use our previously published method, *Phylofit*^{7,9}, and corroborate the results using a suite of methods available in the R package “*cloneRate*”³¹. Both *Phylofit* and *cloneRate* methods estimate growth based on the phylogeny of the mutant clade. *Phylofit* and *cloneRate*’s “maxLikelihood” require the tree to be time-based ultrametric trees. We infer a time-based tree, including all available sampling time points, using our previously developed R package “*rtreefit*” (<https://github.com/NickWilliamsSanger/rtreefit>)⁷, a Bayesian method that jointly fits the mutant rate, the wild-type rate and the timing of the coalescences under the assumption that the observed per branch mutations are Poisson distributed.

Timing *BCR::ABL1* clonal expansion

The *rtreefit* method allows for the transition between mutant and wild-type rates to occur a fixed fraction down the branch on which the mutation is acquired, which is the shared branch upstream of the *BCR::ABL1* positive clonal expansion (Fig.2, showing the upstream ‘trunk’ of the CML clade, and marked as Pre-CML lineage in Fig.3b). As shown in Fig.3e, the signature analysis suggests *BCR::ABL1* clonal expansion commences towards the very end of this branch given its mutation profile is indistinguishable from that of wild-type cells. We therefore configured *rtreefit* to set the acquisition as occurring at the very end of the branch (*rtreefit::fit_clade* parameter: *xcross*=0.99).

A post-hoc justification for the late positioning of the acquisition on this ancestral branch is also furnished by the resulting growth rate estimates. Simulations indicate that the expected time between acquisition and the MRCA of a large mutant clade is approximately exponentially distributed with a mean of $1/s$ (Figure S1a), indeed this is consistent with a known result for birth only processes⁶⁶. This implies that the acquisition of *BCR::ABL1*, and the start of clonal expansion is likely to have occurred towards the end of the shared trunk. The resulting estimated gap between acquisition and the end of the branch is shown in Figure S1b and with S1c showing that our choice of *xcross*=0.99 is sensible. Given the more rapid acquisition of somatic mutations within the mutant clade, clade specific rates are inferred by *rtreefit* for clonal expansions.

This is strong evidence that the acquisition of *BCR::ABL1* (or the commencement of *BCR::ABL1* clonal expansion) is occurring at the very end of the shared branch upstream of the clonal expansion, particularly in patients where the subsequent clonal expansion rate is very high. In those patients that have intermediate growth rates, *BCR::ABL1*

acquisition (or the commencement of *BCR::ABL1* clonal expansion) or is still likely within ~1 year of the end of this shared branch.

Figure S1: a) Time between driver acquisition (or commencement of *BCR::ABL1* clonal expansion) and timing of the most recent common ancestor (“gap”) of a sampled clade with 100 cells. The error bars show the 95% confidence intervals for the mean gap based on the observed gap in 50 simulations for each value of the growth rate s . **b)** Estimated time between *BCR::ABL1* acquisition (or commencement of *BCR::ABL1* clonal expansion) and the MRCA, based on the Phylofit inferred growth rate. **c)** Estimated expected time between *BCR::ABL1* acquisition (or commencement of *BCR::ABL1* clonal expansion) and MRCA as a proportion of mutant trunk length. The gaps range from 0.004 to 0.035 indicating the acquisition is expected to have taken place very much towards the end of the branch.

Comparison between *cloneRate* and *Phylofit* growth estimates

Johnson *et al*³¹ developed a suite of methods for estimating birth-death models with a constant net growth rate. The estimates and associated confidence intervals are asymptotically correct in the limit of a large n (the number of samples in the sampled mutant clade), whilst n remains much smaller than the underlying population. In practice the authors recommend only using the approaches when $n > 10$.

To validate our results using *Phylofit*, we compared them to the results for two of their methods:

- **birthDeathMCMC**: This uses the exact likelihood equation of the constant parameter birth-death process in a Bayesian model to estimate the birth and death rates. As with all the cloneRate methods the estimation essentially assumes a pure exponential growth process.
- **“maxLikelihood”**: This performs a maximum likelihood fit for the observed coalescence times under a model where the coalescence times are independent and identically distributed with an underlying logistic distribution. The growth rate is essentially estimated as being inversely related to the spread of the coalescence timings. We fit the model using our *rtreefit* time based trees inferred using clade specific rates..
- **“Phylofit”**: This fits a growth rate based on a three-parameter logistic growth curve. We fit the model using the *rtreefit* time-based trees.

The estimated growth rates for all methods are shown in **Figure S2**.

Figure S2 Alternative Growth Rate Estimates for the BCR::ABL1 expansions. The error bars are 95% Equal Tailed Credibility Intervals for Phylofit and birthDeathMCMC and 95% Confidence Intervals for the maxLikelihood method. The results in red incorporate a BCR::ABL1 specific mutation rates and are broadly comparable across the methods, except for PD51635 which has an atypical staggered pattern of clonal diversification indicative perhaps of non-constant growth and PD57335 which has some late coalescences which are perhaps due to TKI induced clonal reduction prior to the first sampling point.

Removing late coalescences from PD57335 restores comparability between *Phylofit* and *cloneRate* and yields results that are consistent with the later time point

PD57335 is an outlier with respect to the *cloneRate* results (Figure S2a,b,c). The *cloneRate* estimates are strongly affected by the presence of late coalescences. Dropping 3 selected samples to remove the 3 late coalescences in this phylogenetic tree has very little effect on *Phylofit* ($s=6.602$ goes to $s=6.596$), but markedly increases the

cloneRate estimates (*birthDeathMCMC*: $s=3.82$ goes to $s=6.10$; *maxLikelihood* $s=3.88$ goes to 6.26) to approximately match the estimate of *Phylofit*. PD57335 is also sampled a year after diagnosis, and growth rates inferred using this time point are relatively consistent across methods ($s=7.78, 7.29$ and 7.39 for *Phylofit*, *birthDeathMCMC* and *maxLikelihood* respectively) and are also quite close to the estimates made excluding the late coalescences in the first time point. Under these circumstances it seems likely that *Phylofit* perhaps better captures the early growth rate that was applicable before treatment.

Benchmarking of *Phylofit* and *cloneRate* for large growth rates

Neither *Phylofit* nor the *cloneRate* methods have previously been benchmarked for their accuracy in inferring the very large growth rates of the magnitude that are evident in some of the *BCR::ABL1* positive expansions. In Johnson et al³¹ they benchmarked *cloneRate*, *birthDeathMCMC* and *Phylofit* and concluded that *cloneRate* was slightly less accurate for small n but that both *cloneRate* and *birthDeathMCMC* are more accurate for large n and that *Phylofit*'s CI's tended to be too narrow. In benchmarking the methods for drivers with large growth rates in the presence of limited saturation of the population size we made similar findings. However, if we follow expansions into an extended period of mutant population saturation which is the scenario *Phylofit* was designed to accommodate, we find that the relative inference performance of *Phylofit* improves. The simulation of clonal expansions are generated using *rsimpop* as described in Williams et al,⁷ where the birth rate of mutant cells differs from the birth rate of wild-type cells by " s " but both cell types have the same death rate which is set as the weighted average of the birth rates so that the expected overall population size remains the same. In the early stages of the expansion when the size of the mutant clone is small the death rate is approximately the birth rate of the wild-type and so the growth rate of the mutant clone is " s ". The simulations were set to terminate when there is a 99.99% probability that a mutant population evolving under a constant birth death process with the specified " s " has reached a population of 100,000.

We benchmarked the three methods using a 1000 simulated trees for each of 6 growth rates regimes ($S=1, 10, 100, 1000, 10000, 100000$) and 7 sampled clade sizes (4, 11, 32, 58, 75 and 91) corresponding to mutant clade sizes and the range of inferred growth rates in our CML cohort. For comparability, the *maxLikelihood* estimates were capped at the upper end of the prior range used for the bayesian methods ($s_{max} = 10$ (equivalent to $S=22025$) for $S=1$ and 10, and $s_{max} = 30$ (equivalent to $S=1.07 \times 10^{13}$) for $S \geq 100$). The benchmarking indicates that all methods are somewhat inaccurate for low n (in our case 4 and 11), and that *birthDeathMCMC* and *Phylofit* are similarly accurate for moderate n , however for large the *cloneRate* methods appear to be less accurate (**Figure S3**). The source of the emerging difference in accuracy for large n is evident in **Figure S4** where the *cloneRate* methods tend to underestimate the growth in this limit - presumably because of the presence of late coalescences induced by the saturation of the growth of the mutant population is not compatible with very rapid pure exponential growth.

Figure S3 Root Mean Square Error in Growth Rate Estimation. This plot shows that the cloneRate methods and Phylofit are similarly accurate in their central estimate of growth. For very low values of n (4 and 11) none of the methods performs well, but birthDeathMCMC and Phylofit are more accurate.

Figure S4 Mean Error in Growth Rate Estimation. This shows that for saturated simulated trajectories the cloneRate methods tend to underestimate “s” whilst Phylofit overestimates “s” but tends towards the correct estimate for large n.

Simulated history of a hypothetical CML case

Here we use a case study of a hypothetical CML case to examine the robustness of the estimation methods of the initial phase growth rate based on trees sampled from a mutant population at various longitudinal timepoints from a mutant population with a reasonably realistic mutant cell population size trajectory as illustrated in **Figure S5a**. **Figures S5b-d** show example sampled trees from the mutant cell population at highlighted timepoints in **Figure S5a**. The figures illustrate that in the presence of unconstrained growth e.g. between time point **a** and **b** all methods give reasonably good inferences, whilst following periods of saturation (**Figure 5c**) and population contraction via apoptosis (**Figure 5d**) Phylofit gives estimates that are reasonably accurate and consistent across trees sampled at the same time point where as the other methods give highly variable estimates depending on whether late coalescences are captured.

Figure S5a. The driver is acquired at a) 5 years and introduced into an HSC population of constant size where the same death rate grows rapidly ($s=11$) until the end of the period. At b) trees are sampled, and then the clone carries on growing and then remains at the HSC carrying capacity (here 100,000 cells) for approximately 3 years. At c) trees are sampled again, and then the donor undergoes TKI reducing the mutant population by apoptosis (increased death rate). Finally as d) trees are sampled again.

Phylofit :s_est=11.1(10.0 - 12.5)
maxLikelihood :s_est=10.8(8.5 - 13.1)
birthDeathMCMC:s_est=10.4(8.4 - 12.7)

Phylofit :s_est=10.5(9.8 - 11.7)
maxLikelihood :s_est=9.4(7.4 - 11.4)
birthDeathMCMC:s_est=9.1(7.2 - 11.3)

Phylofit :s_est=10.7(9.8 - 12.0)
maxLikelihood :s_est=9.9(7.8 - 12.0)
birthDeathMCMC:s_est=9.5(7.6 - 11.7)

Phylofit :s_est=10.8(9.9 - 12.1)
maxLikelihood :s_est=9.7(7.7 - 11.8)
birthDeathMCMC:s_est=9.3(7.4 - 11.5)

Phylofit :s_est=11.5(10.2 - 13.0)
maxLikelihood :s_est=11.2(8.9 - 13.6)
birthDeathMCMC:s_est=10.8(8.6 - 13.3)

Phylofit :s_est=11.0(9.9 - 12.6)
maxLikelihood :s_est=10.0(7.9 - 12.1)
birthDeathMCMC:s_est=9.7(7.6 - 11.9)

Phylofit :s_est=12.3(10.7 - 14.0)
maxLikelihood :s_est=12.7(10.0 - 15.4)
birthDeathMCMC:s_est=12.1(9.6 - 14.9)

Phylofit :s_est=11.2(9.9 - 12.7)
maxLikelihood :s_est=10.7(8.4 - 13.0)
birthDeathMCMC:s_est=10.3(8.3 - 12.6)

Phylofit :s_est=11.4(10.0 - 13.1)
maxLikelihood :s_est=10.6(8.3 - 12.8)
birthDeathMCMC:s_est=10.2(8.1 - 12.4)

Figure S5b. Trees sampled immediately at the end of the period of rapid growth at time point b. The figure shows all methods adequately recover the true growth rate ($s=11$).

Figure S5c. Trees are sampled at time c at the end of a prolonged period where the mutant population is at carrying capacity. For this population Phylofit generally adequately recovers the true growth rate. However, the cloneRate methods tend to markedly underestimate the growth rate in the five cases where late coalescences have been captured.

Figure S5d. Trees sampled at time d at the point where the TKI has reduced the mutant clone by about 80%. For this population Phylofit still generally adequately recovers the true growth rate. However, the cloneRate methods markedly underestimate the growth rate in the eight cases where late coalescences have been captured.

16th January 2025

Dear Michelle,

We are very pleased that the Reviewers appreciated the revisions made and additional analyses. We address the remaining comments below. Referee comments are in blue text. Our responses are in black text, with changes that we have made in red text. Text included from the manuscript is highlighted, with any changes we have made to the manuscript text in red text. Changes in the main manuscript are highlighted in yellow. We have also substantially shortened the manuscript to <4300 as requested.

Referees' comments:

Referee #1 (Remarks to the Author): I thank the authors for the additional work done in response to my comments and also for their expanded explanations of the limitations and interpretations of their results. I understand the limitations and feel that the authors have done as much as they can with available material and resources to improve the manuscript. I have no further suggestions

We thank the Referee for their positive comments.

Referee #2 (Remarks to the Author):

The authors have done an excellent job of addressing the Reviewers' comments and have conducted appropriate further analyses, some of which are now included in the paper and Supplementary. As a result, the manuscript is further strengthened. This is a well conducted study that will be an important contribution to the field. I have a couple final minor comments for the authors based on their changes to address for clarity.

We thank the Referee for their valuable suggestions that have strengthened the analyses in this study.

1. The authors now include thorough benchmarking of the different methods for growth rate estimation in the Supplementary, and there is one point of interest that they could expand upon to strengthen the paper. Figures S5c and S5d show that late coalescences will always cause cloneRate estimates to not align well with estimates from Phylofit. The authors state correctly that this is expected, and indeed removing the late coalescences when applying cloneRate recovers the correct growth estimate. In the real data presented however, estimates were typically found to be quite close using the 3 methods in 7 of 8 clones (only PD57335 does not have overlapping confidence intervals as the authors explain). This implies that consequential late coalescences are actually rare in their dataset (as also seen in trees in Figure 2). Alternatively, the authors simulate the test data in revised Figures S3 and S4 for a longer time period (until saturation with probability 99.99%), which I suspect causes late coalescences to occur with very high probability when using the fixed birth/death rates equal to 1 in the simulations. This scenario may not be reflective of most of the clone data in the paper, as observed above. Thus, these observations likely influence the validity of the following statement that the authors make in the rebuttal: "To better reflect the likely CML clonal dynamics, we have simulated the scenario that the clone growth continues beyond the saturation of the population (Rebuttal Figure 2 P=0.9999), in contrast to previous simulations where the clone growth generally halted towards the outset of population saturation (Rebuttal Figure 2 P=0.9)." If the authors could please expand on these

observations with respect to saturation and turnover, it would help tease out plausible biological implications for the clone growth dynamics.

The reviewer wonders whether our choice of saturation probability of 99.99% tends to produce late coalescences with high probability. We address this directly via simulation and find that whilst late coalescences (e.g. ones that would not be consistent with a corresponding constant parameter birth death process) are likely especially in large trees, they are by no means occurring with “high probability”.

Firstly, here is a reminder of the clades used for inferring S together with a magenta line indicating the time from MRCA to most recent coalescence prior to sampling.

The figure below shows the simulated distribution of time between MRCA and the most recent coalescent for the constant time birth-death process run for the same time period as the sigmoidal P=99.99% simulations. For completeness, we also show the distribution for sigmoidal P=90% that was used in our initial benchmarking. It can be seen that trees sampled from the Sigmoidal (P=99.99%) process have a propensity to have late coalescences that would not be expected under the constant birth death process, but that generally there is considerable overlap between the distributions. It will

also be observed that all but two of the trees (PD51633 and PD57335) have observed MRCA->Most Recent Coalescence that is consistent with all the simulated distributions.

- PD51633: This overlaps the tail end of both distributions (at 99.1% and 98.6% quantile for “PureExp” and “Sigmoidal” respectively), so the last coalescence does appear later than would be expected. This would be consistent with an overestimate of S , and as discussed in the previous rebuttal, this is due to our use of the diagnosis time as an upper bound on the prior for the midpoint of the sigmoidal growth curve.
- PD57335: The observed last coalescence lies beyond both distributions. This final coalescence could possibly be because of the first sample being taken 8 months after diagnosis, or possibly a technical duplication of the colony. There are two additional earlier coalescences (see trees above) that would be much consistent with the Sigmoid (99.99%) trajectory but not with the either Sigmoid(90%) or the constant birth-death model.

Given that we have only 4 trees with more than 50 tips, we feel that the presence of multiple late coalescences in even 1 of those trees justifies our choice of 99.99% saturation as a reasonable threshold that encompasses a wide range of possible sampled trees. It is generally consistent with our observed data, and where it is not consistent then it is more consistent than the previous threshold.

With reference to teasing out the biological consequences, one might, for instance, use approximate Bayesian computation to seek to answer the question of “What value of saturation threshold and saturation population most accurately corresponds to the time between MRCA and diagnosis”. Given the heterogeneous nature of our cohort (in terms of differences in elevation of white count, degree of organomegaly, symptom severity, and even *BCR::ABL1* levels at presentation), modest mutant clade tip counts, small sample size and model assumptions, we prefer not to go down this route. The main objective of the benchmarking analysis, and the hypothetical CML case vignette in the supplementary note, was to demonstrate that *phylofit* gives reasonably accurate estimates of initial growth rate over a range of possible clonal dynamics.

2. Thank you for adding the Supplementary Tables S3 and S4. It would be helpful to define what method was used for column G “estimate_cloneRateMoltime” in a Table S4 caption.

We would like to thank the reviewer for pointing out this omission. We have now added detailed captions on the methods used in Table S4 (as below), as well as to remaining supplementary tables where required to aid readability.

Key:	
Note:	All methods : Estimated small s for all methods. To convert to S in excel do: $\exp(s)-1$.
cloneRateML	Estimated using cloneRate::maxLikelihood on the rtreefit time based tree. The values given are the maximum likelihood estimate and corresponding 95% confidence intervals for the annualised growth rate.
cloneRateMoltime	Estimated using cloneRate::sharedMuts on the molecular time based tree and the supplied mutation rate parameter is the median of the rtreefit based clade specific rate. The values given are the maximum likelihood estimate and corresponding 95% confidence intervals for the annualised growth rate.
cloneRateBirthDeathMCMC	Estimated using cloneRate::birthDeathMCMC on the rtreefit time based tree. The value given for the estimate is the median of the posterior distribution of the annualised growth rate and the bounds are for the corresponding equal tailed 95% credibility intervals.
phylofit	Estimated using phylofit on the rtreefit time based tree. The value given for the estimate is the median of the posterior distribution of the annualised growth rate and the bounds are for the corresponding equal tailed 95% credibility intervals.

- Table S3

Key:	
Note:	The median is defined with respect to the relevant posterior distribution.
	Lower and upper bounds are of the equal tailed 95% credibility interval of the relevant posterior distribution.
S_median	Annualised growth rates estimated using Phylofit (median). The clone grows by a factor of $1+S$ each year.
S_lb	Annualised growth rates estimated using Phylofit (lower bound).
S_ub	Annualised growth rates estimated using Phylofit (mean).
s_median	Instantaneous growth rate estimate of Phylofit (median). $S=\exp(s)-1$

s_lb	Instantaneous growth rate estimate of Phylofit (lower bound).
s_ub	Instantaneous growth rate estimate of Phylofit (upper bound).
doubling_time	Doubling time based on the median growth rate estimate. Doubling time= $1/\log_2(1+S)$.
◇ Includes RUNX1 clade as a representative singleton + the 4 non RUNX1 BCR::ABL1 colonies	

- Table S5

Key:	
Note:	Unless otherwise specified the median and mean are defined with respect to the relevant posterior distribution.
	Lower and upper bounds are of the equal tailed 95% credibility interval of the relevant posterior distribution.
DIAGNOSIS_GAP	Gap between diagnosis and first colony sampling (Years)
age_at_diagnosis	Age at diagnosis(Years)
node	Phylo object child node of BCR::ABL1 trunk
n_samples_in_clade	Number of samples in BCR::ABL1 clade
start_mutcount	Mutation count at start of trunk
end_mutcount	Mutation count at end of trunk. The end of the trunk corresponds to the most recent common ancestor of the BCR::ABL1 mutant colonies.
t_lower_lb95	Age since conception at start of trunk (lower bound) (Years)
t_lower_ub95	Age since conception at start of trunk (upper bound)(Years)
t_lower_median	Age since conception at start of trunk (median)(Years)
t_upper_lb95	Age since conception at end of trunk (lower bound)(Years)
t_upper_median	Age since conception at end of trunk (median)(Years)
t_upper_ub95	Age since conception at end of trunk (upper bound)(Years)
t_first_sample	Age at first sample since conception (=Age+(40*7-14)/365.25) (Years)
t_diagnosis	Age at diagnosis since conception (Years)
latency_median	Gap between diagnosis and most recent common ancestor (median) (Years)
latency_lb	Gap between diagnosis and most recent common ancestor (lower bound) (Years)
latency_ub	Gap between diagnosis and most recent common ancestor (upper bound) (Years)
S_Itt_median	Annualised growth rates estimated using Phylofit (median). The clone grows by a factor of 1+S each year.
S_Itt_mean	Annualised growth rates estimated using Phylofit (mean).
S_Itt_lb	Annualised growth rates estimated using Phylofit (lower bound).
S_Itt_ub	Annualised growth rates estimated using Phylofit (mean).
WT_mean	rtreefit estimated mutation rates of wild type colonies (mean). Autosomal SNVs per year.
BCRABL_mean	rtreefit estimated mutation rates of BCR::ABL1 colonies (mean). Autosomal SNVs per year.
WT_ub	rtreefit estimated mutation rates of wild type colonies (upper bound). Autosomal SNVs per year.
BCRABL_ub	rtreefit estimated mutation rates of BCR::ABL1 colonies upper bound). Autosomal SNVs per year.

WT_lb	rtreefit estimated mutation rates of wild type colonies (mean). Autosomal SNVs per year.
BCRABL_lb	rtreefit estimated mutation rates of BCR::ABL1 colonies (mean). Autosomal SNVs per year.
lambda_diff_mean	Mean of difference between estimated mutation rates of mutant and wild type colonies.
lambda_diff_median	Median of difference between estimated mutation rates of mutant and wild type colonies.
lambda_diff_lb	Lower bound of difference between estimated mutation rates of mutant and wild type colonies.
lambda_diff_ub	Upper bound of difference between estimated mutation rates of mutant and wild type colonies.
C_T_Mutant	Proportion of SNVs that are C>T at CpG in mutant clade
C_T_Trunk	Proportion of SNVs that are C>T at CpG in the mutant trunk
C_T_WildType	Proportion of SNVs that are C>T at CpG in wild type branches
n_C_T_Mutant	Count of SNVs that are C>T at CpG in mutant clade
n_C_T_Trunk	Count of SNVs that are C>T at CpG in the mutant trunk
n_C_T_WildType	Count of SNVs that are C>T at CpG in wild type branches
n_Mutant_total	Count of SNVs that are assigned to the mutant clade
n_Trunk_total	Count of SNVs that are assigned to the mutant trunk
n_WildType_total	Count of SNVs that are assigned to wild type branches
s_median	Instantaneous growth rate estimate of Phylofit (median). $S=\exp(s)-1$
s_lb	Instantaneous growth rate estimate of Phylofit (lower bound).
s_ub	Instantaneous growth rate estimate of Phylofit (upper bound).
doubling_time	Doubling time based on the median growth rate estimate. Doubling time= $1/\log_2(1+S)$.

We also noted that in our first revision, a part of Table S2 had accidentally been replaced with a previous version which we have reverted to the correct originally submitted version. This also necessitated a minor numerical correction (685bp to 687bp) in the main text on page 3 (highlighted in yellow). Additionally in Table S6 “BCR::ABL1 VAF” (in column G) for PD60243a has been corrected to 0.370, as the initial value (0.396) represented the VAF of the reciprocal ABL1::BCR event.

Referee #2 (Remarks on code availability): I confirm that this is a usable resource, I downloaded and ran the code successfully.

Thank you

Reviewer #3 (Remarks to the Author):

The authors' response confirms the previous concern that, using the reported dataset, it is impossible to precisely locate the timing of the BCR::ABL1 fusion (formation of the PhT) and the onset of CML. This is because all the SNVs occurring along the phylogenetic stem branch of an expanding HSC clade would bear a similar frequency, so their ordering could not be deconvoluted. The PhT may have happened at any point, not necessarily near the phylogenetic crown immediately preceding the expansion.

We would like to clarify that the onset of CML can be timed confidently based on the commencement of the *BCR::ABL1* positive clonal expansion. We agree that any mutations shared along a branch (including *BCR::ABL1*) cannot be ordered without additional methodologies. We have included the following statement on page 8, “The relative timing of mutations within a shared branch on a phylogenetic tree cannot be disentangled. While we can accurately estimate the timing of *BCR::ABL1* clonal expansion, we cannot infer when along the shared branch *BCR::ABL1* occurred.”

However, additional circumstantial evidence in favor of the rapid onset of CML after PhT acquisition is now provided in the rebuttal document. They surveyed data from the All of Us resource for the presence of PhT and CML symptoms/markers. They reported that most individuals with PhT (multi-read detections) show evidence of CML or associated symptoms, which could reasonably be interpreted as evidence that once the *BCR::ABL1* fusion occurs, CML follows shortly after. This is a valuable result, and it lends some credence to the primary conclusion of this study.

We are pleased that the Referee finds these additional data of value and we would be happy to add this work to the revised manuscript (as detailed below). Of note, the timing of *BCR::ABL1* clonal expansions as inferred from the phylogenetic trees, is also very similar to the latency to CML inferred from radiobiological studies that showed a peak in CML incidence in Japanese survivors of the atomic bombs (Radivoyevitch *et al*, Blood 2012) . Given that in the latter scenario, the ionising radiation exposure event is thought to induce *BCR::ABL1*, the similarities in subsequent trajectory to CML would also make it plausible that *BCR::ABL1* is occurring at the end of the long shared branch upstream of the rapid clonal expansion in this study’s cohort.

However, it was surprising that the revised manuscript did not directly discuss the original caveats (articulated in the first paragraph above). Also, the new All of Us analyses are neither presented nor discussed in the main text and the Methods section (primary versus supplementary). (The only mention of “All of Us” appears as an author contribution attributed to the two newly added authors.) This needs to be remedied before the manuscript can be considered further. They are very important to critically evaluate the support for the authors’ most impactful conclusions and assumptions in modeling.

In particular, there is a need to discuss population-level frequencies of PhT and CML in the main text and mention All of Us in the Methods to support their claims about timing and the assumptions in their modeling. The figures, detailed methodology, and results in the authors’ rebuttal letter should be added in the main text, with details provided in the new Supplementary Note or Extended Data Figure.

We would be delighted to include these data in the revised manuscript and the discussions. We have included the pertinent findings from *All of Us* analysis in the main text as follows:

1. New results section “*BCR::ABL1* and CML in the *All of Us* cohort” on page 7-8 includes the following results:

The relative timing of mutations within a shared branch on a phylogenetic tree cannot be disentangled. While we can accurately estimate the timing of *BCR::ABL1* clonal expansion, we cannot

infer when along the shared branch *BCR::ABL1* occurred. Circumstantial evidence makes it highly plausible that *BCR::ABL1* occurs at the end of the branch triggering clonal expansion: (i) mutational patterns in the branch are identical to wildtype HSPCs suggesting absence of *BCR::ABL1* until the end of the branch (ii) *BCR::ABL1* is sufficient to induce a CML-like phenotype in mice²⁵, suggesting that its acquisition triggers transformation, (iii) additional recurrent genetic driver mutations were not found on shared branches harbouring *BCR::ABL1*, (iv) pre-existing clonal haematopoiesis (CH) was not observed upstream of *BCR::ABL1*, (v) acquiring *BCR::ABL1* acquisition earlier in the branch would require additional non-genetic events to consistently trigger clonal expansion in every patient, making this explanation less parsimonious and violating Occam's razor.

Nevertheless, by only studying CML patients, the possibility that *BCR::ABL1* may also drive slower, decades long, expansion, such as CH, would not be observed. While *BCR::ABL1* has not been reported in CH, such studies have generally only focussed on point mutations, insertions/deletions or chromosomal copy-number changes^{26,27}. Studies interrogating genomic translocations in healthy blood have used whole blood cDNA and not assessed for the presence of stable HSPC clones²⁸. Therefore, we searched the USA based *All of Us* cohort²⁹ of 206,173 whole-genome sequenced participants with linked electronic health data for evidence of *BCR::ABL1* CH. We identified 39 individuals (0.019%) with two or more supporting reads of *BCR::ABL1* or *ABL1::BCR*, the majority of whom had either a diagnosis of CML, other haematological diagnoses suspicious of CML (e.g., splenomegaly or basophilia), or unspecified haematological issues (Ext.Data 6a). While 14 individuals did not have a CML relevant status reported, of these, 3 had sampling after their most recent clinical record, and 3 had no clinical information beyond 1yr after sampling (Ext.Data 6b). All those with >2.5yrs follow-up had a CML relevant status reported. These data suggest that a readily detectable *BCR::ABL1* clone in the healthy population in the absence of a diagnosis of CML is genuinely rare. Indeed, the prevalence of *BCR::ABL1* in *All of Us* (0.019%) is highly similar to the prevalence of CML in the USA (0.02%)³⁰, suggesting that *BCR::ABL1* does not also drive asymptomatic CH.

Interestingly, we also identified 40 individuals (0.019%) with a single read of *BCR::ABL1* or *ABL1::BCR*. Confidence for these cases is lower and more than half lacked features of CML. We wondered if this group of individuals included genuine CML patients (eg., pre-diagnosis or treated), *BCR::ABL1* artefacts and/or *BCR::ABL1* in non-HSPCs. To address the possibility of sequencing artefacts, we screened ~10,000 HSPC whole genomes from single-cell derived haematopoietic colonies previously sequenced at the Sanger Institute³¹⁻³⁵ for the presence of *BCR::ABL1*. Due to the clonality of genomes, bonafide mutations have a variant allele frequency of 50%. WGS coverage was 10-15x and no donors were known to have CML. Across ~10,000 colonies, only three reads in three different colonies were translocated between *BCR* and *ABL1* target regions but none were in the correction orientation for *BCR::ABL1* or *ABL1::BCR*. This suggests that single *BCR::ABL1* reads in *All of Us* are less likely to be sequencing artefacts, raising the possibility that some such cases represent mutated, differentiated blood cells that cannot maintain clones over time. If this were the case, we would not expect such clones to increase in frequency with age. Indeed, single read *BCR::ABL1* carriers displayed a more uniform incidence across age, while carriers with 2 or more reads and patients with CML showed increased incidence with age (Ext.Data 6c). Overall, our analyses suggest that *BCR::ABL1* does not commonly cause CH, providing further support for the one-hit model of CML model depicted by our phylogenies.

2. **Methods** - we have added the methodology to the Methods section “All of us cohort analysis” (page 24).

All of Us cohort analysis. The *All of Us* Research Program²⁹ is a U.S. population-based cohort that links electronic health records (EHR) data to whole-genome sequencing data for 206,173 participants, with an average genome-wide coverage of 30x. We first searched the CRAM files of these individuals to identify sequencing reads supporting a canonical *BCR::ABL1* (or reciprocal *ABL1::BCR*) variant between *BCR* (chr22:23289313-23292813) and *ABL1* (chr9:130674613-130874613). Reads that were incorrectly oriented or had insufficient mapping quality (MAPQ score ≤ 6) were filtered out. The GRASS software (Gene Rearrangement Analysis System, <https://github.com/cancerit/grass>) was then used to annotate sequencing reads that passed the filters. GRASS takes pairs of genomic coordinates representing potential rearrangement events and predicts the fusion consequences along with their associated genes. For this analysis, we required the fusion event to be specifically between *BCR* and *ABL1*. After identifying *BCR::ABL1* carriers in the cohort, we extracted their EHR data and searched for International Classification of Diseases (ICD) codes related to cancer, blood, or immune diseases. In addition to searching for mentions of chronic myeloid leukemia (CML) in the ICD codes, we included conditions such as ‘abnormal white cell count,’ ‘basophilia,’ ‘splenomegaly,’ ‘unspecified cancer,’ or ‘anaemia.’ Based on these disease entries and domain knowledge, we categorized carriers into four groups: (1) CML mentioned, (2) Abnormal white count (e.g., leucocytosis), basophilia, or splenomegaly mentioned, (3) Other haematological issues (e.g., anaemia) or unspecified cancer, and (4) No relevant disease. We used the blood sampling date and the date of the most recent ICD code in the EHR for each carrier to define the time from sample collection to the last follow-up. When calculating the incidence of CML and *BCR::ABL1* carrier status across age groups, we defined age at diagnosis for CML and age at blood sampling for *BCR::ABL1* carrier status. For each age group, summary statistics were calculated by averaging annual statistics from 2018 to 2022, assuming that the population structure remained relatively stable during this period.

3. We have added to the discussion section “The cancer trajectory to adult CML bucks this trend, with explosive and rapid tumour growth, tumour doubling times reaching 2-3 weeks, and clinical disease presenting as soon as 3-4yrs later. In keeping with this, our data highlight an absence of *BCR::ABL1* CH in the population. Subsequent genomic evolution beyond *BCR::ABL1* also drives disease progression, in line with previous studies^{15,17,44}, but also cell identity during blast phase.” (page 9)
4. We have added Extended Data 6 incorporating all 3 figures from the previous Rebuttal in relation to the *All of Us* data.

Given that we have had to substantially shorten the manuscript to comply with editorial guidelines, we hope that there is now sufficient attention to this important point in the main text.

STATISTICS: When revising your manuscript, you should ensure that any statistical analysis used is sound and that it conforms to [Nature's guidelines \[nature.com\]](https://www.nature.com/articles/d41586-020-00000-0). A collection of articles explaining the basics of statistical analysis and advice on how to best present it can be found [here \[nature.com\]](https://www.nature.com/articles/d41586-020-00000-0).

We confirm.

REPRODUCIBILITY: All of the checklists provided with the current submission (Reporting summary [nature.com], Editorial policy checklist [nature.com], and Code and software checklist [nature.com] (if applicable)) should be updated to reflect the revisions made and submitted with the revised manuscript.

We can confirm that the data on github allows reproducibility of the analyses presented. We are pleased Referee 2 finds this a useable resource.

LENGTH: In print, biological sciences papers do not normally exceed 8 pages on average; the final print length, however, is at the editor's discretion. The typical length of an 8-page article with 5 modest (quarter-page) display items is 4300 words. If a composite figure (with multiple panels) must occupy at least half a page in order for all the elements to be visible, the text length may need to be reduced accordingly to accommodate such figures. Essential but technical details can be moved into the Methods or Supplementary Information (see below).

Our manuscript main text is now 4298 words (including section subtitles), together with 4 figures and 6 extended data figures. This word count includes the additional *All of Us* data in the main manuscript, as requested by Referee 3.

Titles cannot exceed 75 characters (including spaces); they must not contain punctuation.

Our title has been amended to "Timing and trajectory of *BCR::ABL1* driven chronic myeloid leukaemia" (67 characters including spaces).

SUMMARY PARAGRAPH: All Nature papers begin with a fully referenced paragraph, typically no longer than 200 words, aimed at readers in other disciplines.

Our abstract is now 199 words.

MAIN TEXT: If further introductory material is necessary, the main text can begin with up to 500 words of introduction expanding on the background to the work (some overlap with the summary is acceptable), before proceeding to a concise, focused account of the findings, and ending with 1 or 2 short paragraphs of discussion. Sections are separated with subheadings (up to 40 characters including spaces) to aid navigation.

Our introduction is 307 words. Subheadings have been shortened to 40 characters.

REFERENCES: As a guideline, most papers should include no more than 50 main text references; additional references can be cited in (and listed after) the Methods section, as detailed below.

We now have 50 references in the main manuscript. Remaining references refer to Methods and are listed after that section.

FIGURE LEGENDS: These should be listed sequentially after the main text references and not in the figure files; they should not exceed 300 words each. Each legend should begin with a brief title for the whole figure and continue with a short description of each panel and the symbols used. Each figure legend should contain, for each panel where relevant, the following information:

- * the exact sample size (n) for each experimental group/condition, given as a number, not a range;
- * a description of the sample collection allowing the reader to understand whether the samples represent technical or biological replicates (including how many animals, litters, cultures, etc);
- * a statement of how many times the experiment shown was replicated;
- * definitions of statistical methods and measures:
- * very common tests (e.g., t-test, simple Chi-square tests, Wilcoxon and Mann-Whitney tests) can be identified by name only, but more complex techniques should be described in the Methods;
- * whether tests are one-sided or two-sided;
- * whether there are adjustments for multiple comparisons;
- * the statistical test results (e.g., P values);
- * the definition of 'center values' as median or average;
- * the definition of error bars as s.d. or s.e.m.

Any descriptions too long for the figure legend should be included in the Methods section; see here [nature.com] for further explanation.

We have reduced the legends to <300 words each.

METHODS: After the main text figure legends there should be a section entitled "Methods", which provides the full, step-by-step instructions that would allow other researchers to replicate the results. The Methods section will not appear in print but will appear online in the full-text HTML and PDF versions. The Methods section should be written as concisely as possible but should contain all elements necessary to allow interpretation and reproduction of the results. If there are additional references in the Methods section, their numbering should continue from the last reference in the main text, and they should be listed following the Methods section. Specialized methods that require chemical structures, figures or tables, or methods requiring equations, cannot be accommodated in the Methods section of the main text file. If such information is part of the Methods, the entire Methods section must instead be included within a Supplementary Information text file.

We confirm.

MAIN TEXT STATEMENTS: Several statements (which will not appear in print but will appear online in the full-text HTML and PDF) are required after the Methods, and before the Extended Data legends. First, there should be an Acknowledgements section, listing grant/financial support. Next, we require a detailed Author Contribution statement; the specific contributions of each author, particularly in terms of which authors performed which specific experiments, must be listed. This is followed by a Competing Interest statement. Financial or non-financial interests should be noted here, as well as any patents; patent information should include at a minimum what is covered by the patent and who submitted the patent application. Finally, an Additional Information statement should include information regarding reprints and permissions and name the author(s) to whom correspondence and requests for materials should be addressed. For details of "end note" style and an example see here [nature.com].

We confirm.

DATA AND CODE AVAILABILITY STATEMENTS: All original research manuscripts published in Nature Portfolio journals must include a Data availability statement (DAS). This statement must make the conditions of access to the “minimum dataset” that is necessary to interpret, verify and extend the research in the article, transparent to readers. This minimum dataset may be provided through deposition in public community/discipline-specific repositories, custom proprietary repositories for certain types of datasets, or general repositories like Figshare, Zenodo and Dryad. Providing large datasets in supplementary information is strongly discouraged and the preferred approach is to make data available in repositories. More information on Nature Portfolio’s reporting standards and preparing your Data availability statement can be found here [nature.com].

We confirm.